# Mesoscale cortex-wide neural dynamics predict self-initiated actions in mice several seconds prior to movement

Catalin Mitelut[1,2,3,4]*, Yongxu Zhang[5], Yuki Sekino[1,2], Jamie D Boyd[1,2], Federico Bollanos[1,2], Nicholas V Swindale[3], Greg Silasi[6], Shreya Saxena[5†], Timothy H Murphy[1,2]*†

[1]Department of Psychiatry, Kinsmen Laboratory of Neurological Research, University of British Columbia, Vancouver, Canada; [2]Djavad Mowafaghian Centre for Brain Health, University of British Columbia, Vancouver, Canada; [3]Department of Ophthalmology and Visual Sciences, University of British Columbia, Vancouver, Canada; [4]Biozentrum, Centre for Molecular Life Sciences, University of Basel, Basel, Switzerland; [5]Department of Engineering, University of Florida, Gainesville, United States; [6]Department of Cellular and Molecular Medicine, University of Ottawa, Ottawa, Canada

*For correspondence:
mitelutco@gmail.com (CM);
thmurphy@mail.ubc.ca (THM)

†These authors contributed equally to this work

Competing interest: The authors declare that no competing interests exist.

**Abstract** Volition – the sense of control or agency over one's voluntary actions – is widely recognized as the basis of both human subjective experience and natural behavior in nonhuman animals. Several human studies have found peaks in neural activity preceding voluntary actions, for example the readiness potential (RP), and some have shown upcoming actions could be decoded even before awareness. Others propose that random processes underlie and explain pre-movement neural activity. Here, we seek to address these issues by evaluating whether pre-movement neural activity in mice contains structure beyond that present in random neural activity. Implementing a self-initiated water-rewarded lever-pull paradigm in mice while recording widefield [Ca++] neural activity we find that cortical activity changes in variance seconds prior to movement and that upcoming lever pulls could be predicted between 3 and 5 s (or more in some cases) prior to movement. We found inhibition of motor cortex starting at approximately 5 s prior to lever pulls and activation of motor cortex starting at approximately 2 s prior to a random unrewarded left limb movement. We show that mice, like humans, are biased toward commencing self-initiated actions during specific phases of neural activity but that the pre-movement neural code changes over time in some mice and is widely distributed as behavior prediction improved when using all vs. single cortical areas. These findings support the presence of structured multi-second neural dynamics preceding self-initiated action beyond that expected from random processes. Our results also suggest that neural mechanisms underlying self-initiated action could be preserved between mice and humans.

## Editor's evaluation

This study is a valuable work that advances our knowledge of the neural correlates of voluntary action through a wide range of methods. The evidence supporting their conclusion is convincing, and the results will be of interest to a large class of neuroscientists interested in the neural mechanisms underlying self-initiated actions.

## Introduction

Over the past several decades studies of volitional, that is free and voluntary, action in humans using self-initiated (i.e., uncued) behaviors such as flexing a finger or pressing a button have shown that prior to movement there is a gradual increase in scalp electroencephalography (EEG) signal over pre- and supplementary motor area (pre-SMA and SMA, respectively; *Ball et al., 1999*; *Cunnington et al., 2002*). This increase in activity is known as the 'readiness potential' (RP; *Kornhuber and Deecke, 1964*; *Deecke et al., 1976*; *Deecke and Kornhuber, 1978*; *Libet et al., 1983*; *Shibasaki and Hallett, 2006*) and has received increasing attention with some interpretations that it is evidence that voluntary decisions might be made prior to awareness with several studies replicating and extending the original work (*Haggard and Eimer, 1999*; *Schlegel et al., 2013*; *Sirigu et al., 2004*; *Alexander et al., 2016*). Additionally, single neuron physiology studies have also shown a significant increase (or decrease) in the firing rate of single neurons in SMA and pre-SMA (as well as anterior cingulate cortex) prior to movement (*Fried et al., 2011*). In parallel, human functional magnetic resonance imaging (fMRI) studies have shown that upcoming behaviors could be decoded up to several seconds prior to movement (*Soon et al., 2008*; *Soon et al., 2013*; *Bode et al., 2011*; *Colas and Hsieh, 2014*). The role of pre-movement neural activity in voluntary behavior is the subject of active debates on human decision making including free will (*Jahanshahi and Hallett, 2003*; *Lang, 2003*; *Shibasaki and Hallett, 2006*; *Haggard, 2008*; *Klemm, 2010*; *Custers and Aarts, 2010*; *Schurger et al., 2012*; *Deecke, 2012*; *Guggisberg and Mottaz, 2013*; *Bode et al., 2014*; *Maoz et al., 2015*; *Lavazza, 2016*; *Schurger et al., 2016*). These debates on the neural genesis of voluntary action are further complicated as other studies have shown volitional actions are more likely to occur during certain phases of breathing (e.g., exhalation; *Park et al., 2020*), or phases in cumulative neural activity (i.e., the crest in slow cortical potential – SCP; *Schmidt et al., 2016*) which have no immediately obvious connections to volitional intent, awareness, neural noise, or external stimuli or cues.

Although some have called for better-designed self-initiated behavior studies (e.g., *Mudrik et al., 2020*), it remains challenging to implement them in humans especially in cue-free paradigms. First, neuroanatomically precise high-temporal and spatial precision recordings from many cortical areas are rare in humans (though some limited studies exist, e.g., *Fried et al., 2011*). Second, obtaining statistically sufficient numbers of trials (i.e., tracking behaviors for days or weeks; see *Bode et al., 2014* for a discussion) of higher-value salient actions (e.g., important decisions that are made naturally outside of laboratory environments) is not yet possible in humans. Additionally, human laboratory protocols for volitional studies (e.g., the subject being told to act freely in a study) may result in instructed – rather than free behavior – and there are concerns about whether human subjects can act randomly (*Lages et al., 2013*) or otherwise carry out balanced behaviors (e.g., randomly pressing left vs. right button) in voluntary behavior paradigms (*Bode et al., 2014*). An alternative approach to studying self-initiated action is to characterize the neural correlates of voluntary action and removing the requirement for reporting intent or awareness altogether (similar to some human studies, e.g., *Soon et al., 2008*; *Bode et al., 2011*). This avoids some of the challenges of human paradigms and makes it possible to implement nonhuman animal models where more ethologically valuable actions could be available (i.e., food or water seeking behaviors) and higher-resolution intracranial neural recordings can be made during hundreds or thousands of trials. There is evidence to support this direction as several nonhuman studies have identified structure – or increases – in pre-movement neural activity in nonhuman primates (*Romo and Schultz, 1986*; *Romo and Schultz, 1990*; *Coe et al., 2002*; *Lee and Assad, 2003*; *Maimon and Assad, 2006*; *Ding and Hikosaka, 2006*), rodents (*Hyland, 1998*; *Isomura et al., 2013*; *Murakami et al., 2014*), crayfish (*Kagaya and Takahata, 2010*), and zebrafish (*Lin et al., 2020*). However, none of these studies were designed to – nor report – the neural structure of either voluntary or self-initiated actions and they do not evaluate the predictive relationship between pre-movement cortical neural activity and self-initiated behaviors.

Here, we report results obtained from a self-initiated behavior paradigm targeting the decoding of future body movement and rewarded actions from neural activity in mice. Using a self-initiated task and widefield [Ca++] cortical imaging (*Silasi et al., 2016*; *Vanni and Murphy, 2014*), we tracked both water-rewarded lever-pull behavior of water deprived mice and spontaneous body movements. We gathered hundreds to thousands of self-initiated actions over months of recordings and collected neural activity from several cortical areas. We find that both self-initiated water-rewarded lever pulls and spontaneous body movements could be decoded above chance a few to several seconds prior to

movement initiation from neural activity. We show the self-initiated movement neural code is distributed across multiple cortical areas and additionally replicate and extend several findings from human studies. Our study is in line with accounts of pre-movement neural activity having temporal and spatial structure beyond that present in random neural dynamics and supports a causal role between pre-movement neural activity and action that is on the scale of several seconds prior to action.

## Results

Several studies of voluntary actions, such as finger or wrist movements, in humans have identified an increase in scalp EEG signal over SMA and pre-SMA – known as the RP – occurring 0.5–1.5 s prior to movement and in some cases awareness of movement (*Figure 1a, b*; *Kornhuber and Deecke, 1964*; *Kornhuber and Deecke, 1965*; *Libet et al., 1983*; *Ball et al., 1999*; *Cunnington et al., 2002*).

### Self-initiated movements in mice are preceded by stereotyped neural activity changes several seconds prior to movement

We developed an analogous self-initiated behavior paradigm in six mice (M1–M6) to characterize pre-movement neural activity while recording widefield [Ca++] activity from cortex (*Figure 1c, d*, *Figure 1—figure supplements 1 and 2*; see also Methods; see also *Table 1*). Mice were headfixed and trained to perform a self-initiated lever pull to receive a water reward without sensory cues or stimuli. Four of six mice learned the lever lockout period of 3 s (*Figure 1—figure supplement 2* shows peaks at 3 s in the inter-lever-pull intervals in mice M3–M6) and four of six mice learned to pull increasingly more often toward the end of each ~20-min session (*Figure 1—figure supplement 4*, mice). Mice tended to decrease their body movements prior to a lever pull and we did not find evidence of stereotyped behaviors prior to lever pull (*Figure 1—figure supplement 5*; see also Methods on detecting stereotyped movements). Similar to the RP in humans, self-initiated behaviors in mice are preceded by an increasingly stereotyped average widefield [Ca++] signal up to 5 s or earlier in several areas including motor and limb cortex (*Figure 1e*, *Figure 1—figure supplement 3*). This common dynamical pattern was observed in all sessions and animals but not when considering random segments of neural activity (*Figure 1—figure supplement 3*).

### Self-initiated movements in mice can be decoded seconds prior to action from preceding neural activity

In addition to the human RP, several human fMRI studies have shown that voluntary behavior could also be decoded up to several seconds prior to movement, usually a few percent above chance (*Figure 1f, g*; *Soon et al., 2008*; *Soon et al., 2013*; *Bode et al., 2011*). To compare with human studies, we trained support vector machines (SVMs) using trials within each session to decode upcoming rewarded lever pulls (*Figure 1h*) or spontaneous limb movements (*Figure 1i, j*). We decoded, that is classified, (1) neural activity preceding a self-initiated action (e.g., rewarded lever pull) vs. (2) neural activity representing random periods of behavior similar to two class voluntary choice decoding carried out in humans (e.g., *Soon et al., 2008*; note: we defined random activity as continuous segments of neural data that were centered at least 3 s outside of lever-pull times; see Methods). Within each session we used sliding windows of 1 s of neural activity as input to the SVMs. Additionally, we trained markerless pose estimation methods to track the spontaneous limb movements of mice (see Methods). To disambiguate the effects of multiple sequential lever pulls, we considered only lever pulls that were preceded by at least 15 s of no-lever-pull activity (see further results below and Methods on lever lockout analysis). Upcoming rewarded lever pulls could be decoded several seconds prior to movement with decoding accuracy curves improving closer to the lever-pull time (examples in *Figure 1h*). Similarly, spontaneous limb movements were also decodable above chance a few seconds prior to movement (examples in *Figure 1i, j*).

In sum, neural activity preceding self-initiated lever pulls in mice is preceded by multi-second stereotyped changes in widefield [Ca++] cortical activity similar to the human RP. Similar to human fMRI results, upcoming lever pulls or spontaneous limb movements could be decoded from preceding neural activity with similar decoding accuracy previously reported in humans (e.g., *Soon et al., 2008*;

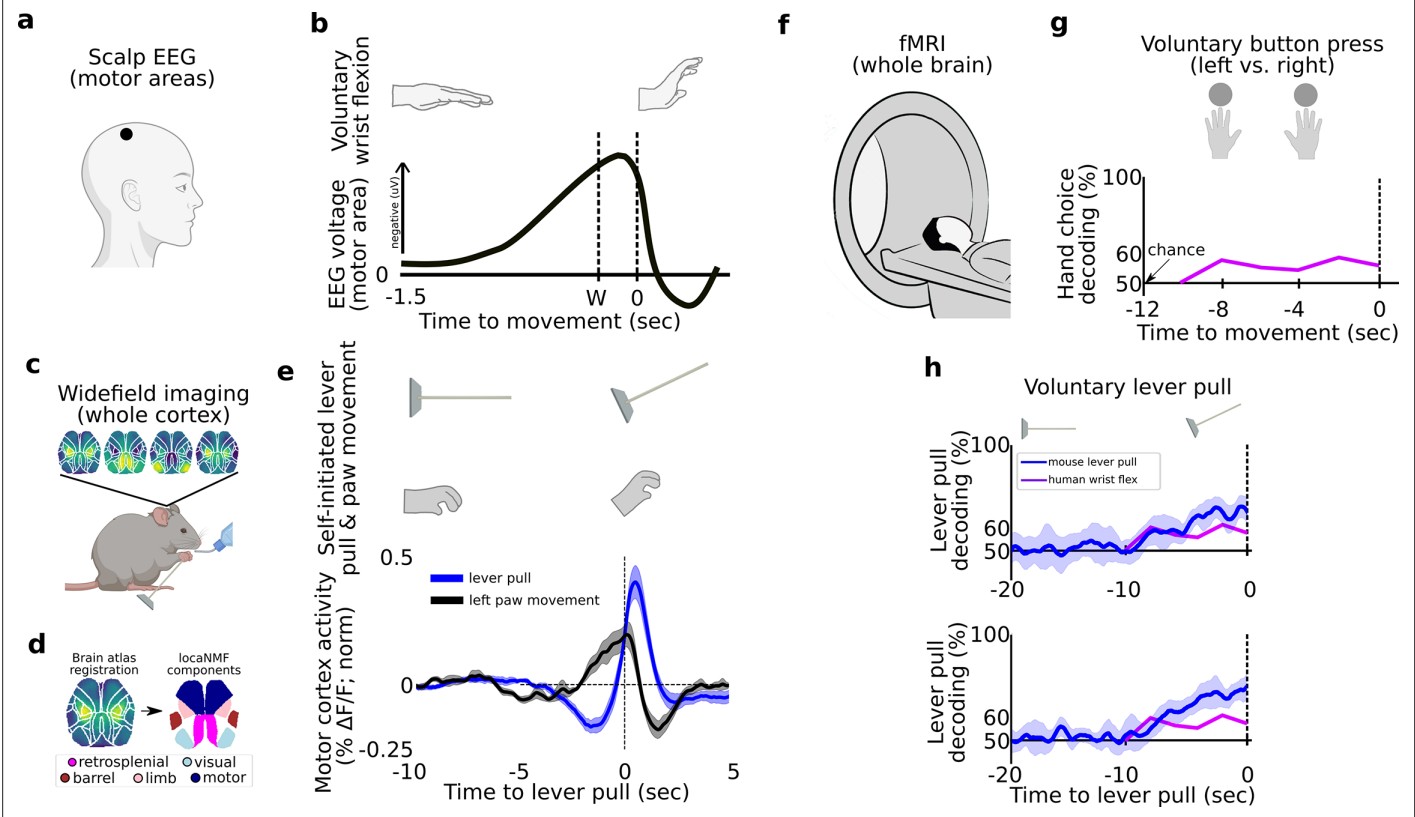

**Figure 1.** Tracking and decoding self-initiated behaviors from widefield neural activity in mice . Detecting and decoding upcoming self-initiated mouse behaviors via widefield calcium activity. (**a**) Human voluntary behavior studies using scalp electroencephalography (EEG) target motor cortex and related areas. (**b**) Human voluntary wrist flexion studies recording EEG from motor areas (as in (**a**)) reveal a change in neural activity 0.5–1.5 s prior to behavior initiation (*t* = 0 s) or even awareness (*t* = *W*) in some studies (note: the *y*-axis of the plot is inverted so the readiness potential [RP] is negative). (**c**) Mice learn to voluntarily self-initiate a lever pull to receive water reward while widefield [Ca++] activity is captured at 30 Hz. (**d**) Allen Brain Atlas and locaNMF decomposition of neural activity into neuroanatomical areas (see also Methods). (**e**) The average motor cortex widefield calcium (neural) activity (solid blue line) becomes increasingly stereotyped prior to a self-initiated lever pull (*t* = 0 s; data shown are from 21 s locked out trials; dashed line represents Hilbert transform of oscillatory signal). (**f**) Human studies seeking to decode voluntary choice relying on functional magnetic resonance imaging (fMRI) during voluntary behaviors. (**g**) Decoding accuracy for left- vs. right-hand voluntary button presses (solid purple line) is a few percent above chance several seconds prior to movement initiation and peaks at approximately 10% above chance at movement time. (**h**) Two examples of decoding accuracy for rewarded lever pull vs. random states in mice (solid blue lines; shading is the standard deviation of 10-fold cross-validation accuracy) showing increases seconds prior to movement and peaking >30% above chance at movement time (data shown for two examples of long earliest decoding times (EDTs) obtained from mice M2 and M3 using 15 s locked out low-pass filtered trials; see Main text). (**b**) has been adapted from Box 1 in *Haggard, 2008*. (**g**) Has been adapted from Figure 2 in *Soon et al., 2008*.

The online version of this article includes the following figure supplement(s) for figure 1:

**Figure supplement 1.** Mouse self-initiated lever-pull paradigm.

**Figure supplement 2.** Mouse longitudinal inter-lever-pull interval distributions.

**Figure supplement 3.** Widefield [Ca++] dynamics during random vs. rewarded lever pulls.

**Figure supplement 4.** The number of lever pulls changes across sessions.

**Figure supplement 5.** Body movement sequence analysis.

**Figure supplement 6.** Characterizing quiescent periods prior to rewarded lever pulls.

**Figure supplement 7.** Characterizing pre-lever-pull neural activity as a function of body movement.

*Bode et al., 2011*). Our findings suggest that, like humans, mice may also engage pre-movement neural dynamics spanning several seconds prior to self-initiated action.

We sought to systematically evaluate decoding accuracy for upcoming behaviors across mice, by evaluating multiple cortical anatomical areas and for different movements (e.g., water-rewarded lever pulls or spontaneous limb movements; *Figure 2*).

**Table 1.** Lever-pull statistics.

| Animal ID | # of behavior sessions* | # of video sessions* | Median # rewarded pulls/session | Median # unrewarded pulls/session |
|-----------|------------------------|---------------------|-------------------------------|----------------------------------|
| M1 | 69 | 30 | 16 | 14 |
| M2 | 42 | 12 | 34 | 14 |
| M3 | 42 | 11 | 42 | 135 |
| M4 | 46 | 10 | 33 | 111 |
| M5 | 42 | 11 | 38 | 116 |
| M6 | 109 | 70 | 45 | 35 |

*Total numbers reported include also rejected sessions due to insufficient trials (see also Methods).

## Decoding future rewarded lever pulls seconds prior to movement

We next focused on decoding water-rewarded lever pulls: that is, lever pulls that reached a minimum lever angle and were not preceded by a previous lever pull for at least 3 s (*Figure 2a*; *Figure 2—figure supplement 1*; see also Figure 4 and Methods). Going back in time from $t = 0$ s, we defined the earliest decoding time (EDT) as the last point in time at which the SVM accuracy was statistically higher than chance (*Figure 2b*, 10-fold cross-validation pval <0.05, Student 1 sample $t$-test corrected for multiple hypotheses using the Benjamini–Hochberg method; see Methods). The decoding accuracy was better than chance seconds prior to movement and gradually increased closer to the lever pull (i.e., $t = 0$ s). EDTs ranged from 0 s (i.e., lever pull was not predicted) to more than 13 s in some sessions (see *Figure 2c* for example EDTs). We also found a correlation between the number of trials within a session and EDT suggesting that EDTs could in principle be even lower (i.e., earlier decoding in time) than reported in our study (*Figure 2c* linear fit; note each EDT was computed from a single session; see also Methods). To evaluate this correlation, trials from sequential sessions were concatenated to obtain at least 200 cumulative trials resulting in improvement in EDT (*Figure 2d, e*). Pooled sessions EDTs were lower for all mice (*Figure 2f* – left panel; pvals <0.01 for all animals; single session averages in seconds: M1: −2.91; M2: −2.87; M3: −4.75; M4: −3.55; M5: −4.32; M6: −6.91; concatenated averages in seconds: M1: −4.84; M2: −5.47; M3: −7.58; M4: −5.76; M5: −6.99; M6: −6.92).

Given the strong dependence of EDT on the # of trials, we sought to re-evaluate EDTs using only trials that were not preceded by another lever pull (either rewarded or non-rewarded). We find that oscillations observed in the neural data are likely enforced by repetitive and stereotyped recent lever pulls and that EDT analysis requires exclusion of trial that occur too soon after a previous lever pull (here we chose a lockout of 15 s; note: this approach significantly decreased the number of trials available for analysis as mice only rarely went without pulling the lever for 15 s; we thus pooled trials from across sessions into a minimum of 50 to a maximum of 200 trial hybrid sessions; see Methods). We found that after lockout the neural data had a single negative (i.e., inhibitory) phase preceding self-initiated rewarded lever pulls that comenced ~5 s prior (similar to *Figure 1e*; see *Figure 2—figure supplement 2*). We additionally found that EDTs decoded from lockout trials were shorter (*Figure 2g*: average EDT in seconds: mouse 1 (M1): −1.93; M2: −3.14; M3: −2.27; M4: −1.87; M5: −1.64; M6: −2.49). However, low-pass filtering the neural time series (at 0.3 Hz) (as a type of feature engineering based on power analysis results in Figure 7) resulted in EDTs more similar to the initial results (*Figure 2*; average EDTs of causal filtered neural data in seconds: M1: −3.5; M2: −4.85; M3: −6.95; M4: −4.31; M5: −3.0; M6: 3.7). The improvement in EDT was qualitatively observable in decoding accuracy curves (*Figure 2i*) and was present even for non-lock out trials (see *Figure 2—figure supplement 4*; see also Methods).

The initial loss of EDT (without the filtering step) suggests that sequential lever pulls might have a causal role in lengthening EDTs by generating stereotyped neural time series which represents preceding – not just the current – rewarded lever pulls. However, we also found that pooling trials from sessions far apart in time (days or weeks) as required by the lockout method also shortened EDT values (i.e., closer to 0 s; *Figure 2—figure supplement 3*). This suggests that higher data variance (due to learning, [Ca] bleaching, implant degradation, etc.) might also have a causal role in shortening

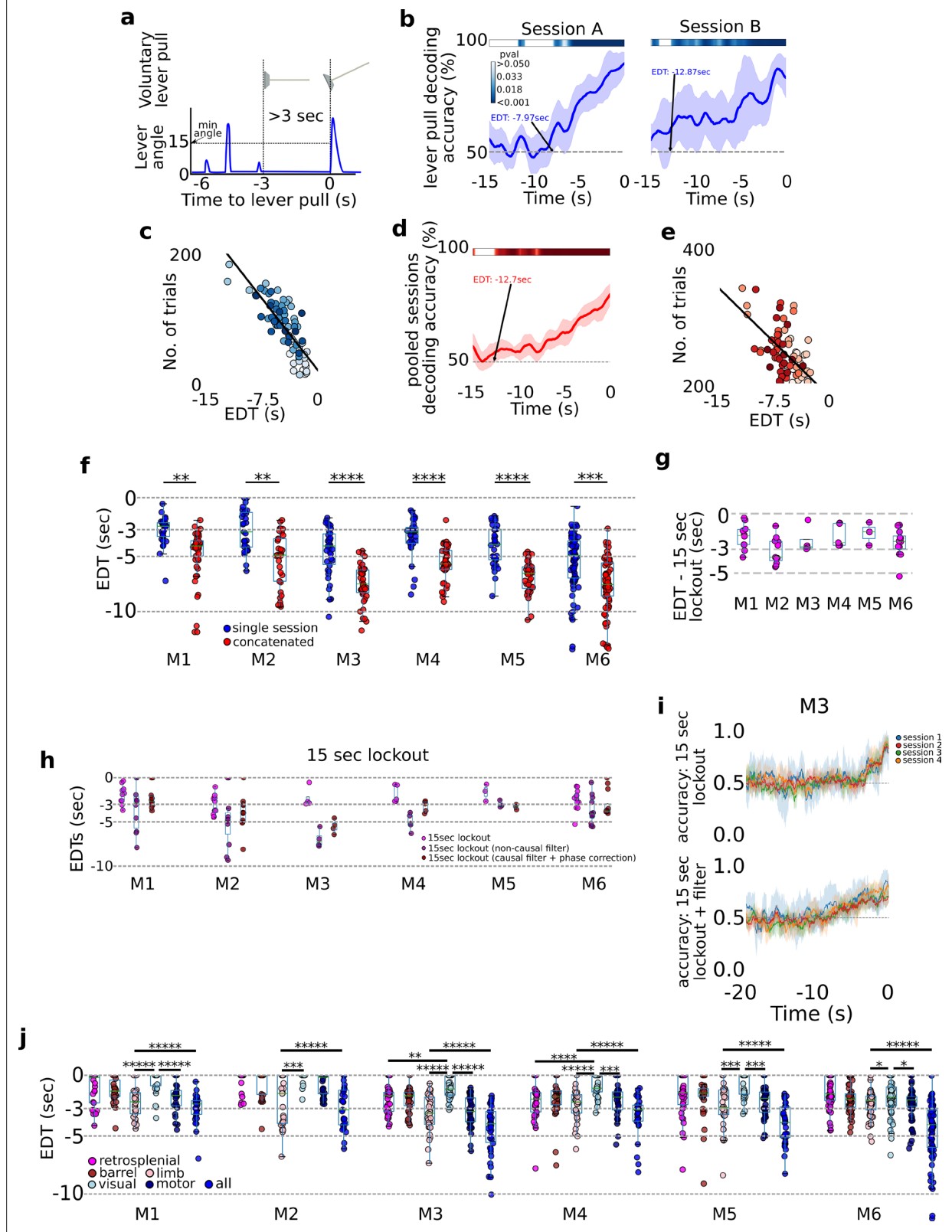

**Figure 2.** A cortex-wide distributed multi-second neural code underlies self-initiated actions . Decoding self-initiated lever pulls using cortical neural activity. (**a**) Decoding self-initiated water-rewarded lever pulls using a minimum 3-s lockout window and a minimum lever angle threshold (see also Methods). (**b**) Support vector machine (SVM) decoding accuracy curves of two different sessions (mouse M4) reveal increased decoding accuracy near lever-pull time and an earliest decoding time (EDT) of several seconds (curves represent average accuracy and shaded colored regions represent

*Figure 2 continued*

standard deviation over 10-fold cross-validation; top colored bars represent p values of Student *t*-test with a Benjamini–Hockberg correction for multiple hypotheses; see Methods for more details; note sessions shown were atypical and were selected to illustrate decoding curves for early EDTs when decoding from all trials, that is without locking out previous lever pulls. (**c**) EDTs from all sessions (mouse M6) show a strong correlation between EDT and the number of trials within a session (lighter colors indicate earlier sessions in the experiment). (**d**) Same as (**b**) but for an example from concatenated, that is multi-session, analysis. (**e**) EDTs for concatenated sessions (M6) also show a correlation between EDT and the number of trials present in the session and the EDT (lighter shading representing earlier sessions in training). (**f**) EDT distributions across all mice for single session trials (blue) vs. multi-session trials (red) reveals a significant lengthening (i.e., further from lever-pull time) in EDTs for multi-sessions across all animals (cyan box plots show 25th percentile, median, and 75th percentile; comparisons between single vs. concatenated sessions were carried out using two-sample KS test with asterisks indicating: *<0.05; **<0.01; ***<0.001; ****<0.0001; *****<0.00001). (**g**) Same as (**f**) but for 15-s lockout trials concatenated across multiple sessions. (**h**) Recomputed EDTs for filtered calcium traces. (**i**) Examples of decoding accuracy curves for 15-s lockout trials using filtered vs. non-filtered neural time series (see also Methods).

The online version of this article includes the following figure supplement(s) for figure 2:

**Figure supplement 1.** Tracking self-initiated body movements.

**Figure supplement 2.** Median neural activity prior to lever pull.

**Figure supplement 3.** Subsampling non-lockout trials longitudinally leads to an increase in earliest decoding times (EDTs).

**Figure supplement 4.** Earliest decoding times (EDTs) from low-pass filtered data do not show increased values.

EDTs. Overall, these findings show that self-initiated water-rewarded lever pulls in mice can have neural correlates that are present up to several seconds prior to lever pull and can be decoded several seconds prior to level pulls, but that such analysis must appropriately take into account previous behaviors and the effects of data variance over longitudinal studies.

## Preparation of upcoming lever pulls is widely distributed across the cortex

We next evaluated decoding of upcoming lever pulls using individual cortical areas rather than the entire dorsal cortex (*Figure 2i*). Anatomically informed components were obtained using LocaNMF (*Saxena et al., 2020*; see Methods) and EDTs were computed for bilateral activity from: retrosplenial, somatosensory-barrel, somatosensory-limb, visual, and motor cortex. Somatosensory-limb cortex was generally the most informative of upcoming lever pulls (i.e., lowest mean EDTs across all mice) followed by motor cortex; visual cortex-based decoding had the highest EDTs (i.e., closest to lever-pull time $t = 0$ s). More importantly, using all regions for decoding yielded lower EDTs than using somatosensory-limb cortex alone (two sample KS test comparing limb-cortex vs. all neural regions EDTS; *D*-statistic for all mice: 1.0; p values: M1: 0.0; M2: 5.55e−16; M3: 2.90e−22; M4: 2.77e−19; M5:4.52e−21; M6: 2.22e−16). These findings are consistent with a single neuron study in humans that showed pooling neurons yielded increased decoding accuracy of upcoming voluntary action when compared to single neuron decoding alone (*Fried et al., 2011*) but our results yield earlier decoding times than previously shown in humans.

## EDTs of non-locked out paw movements and licking events are similar to those for lever pulls

Most human studies on self-initiated voluntary behavior employ simple behaviors such as the flexing of a finger or pressing of a button with a specific hand (e.g., *Libet et al., 1983*; *Soon et al., 2008*). Accordingly, we also sought to determine whether mouse spontaneous paw movements (not just those related to water-rewarded lever pulls) could be decoded from preceding neural activity (see Methods for description of body movement tracking methods). Briefly, we defined a self-initiated body movement as the time when the body part increased its velocity to more than 1× the standard deviation of all movement within the session (we also implemented a 3-s non-movement lockout period as in human studies and as in the preceding section). SVMs were trained as for rewarded lever-pull times but using the body movement initiation time (i.e., $t = 0$). As for self-initiated lever pulls, a strong correlation was present between the number of trials within a session and the EDT suggesting that with higher number of body movements (e.g., longer sessions) EDTs could be even lower (Figure

2, figure supplement 4). Across all animals, upcoming body movements could be predicted above chance a few seconds prior to movement in the vast majority of sessions and in some cases more than 10 s prior to movement (**Figure 2l**). Importantly, with a few exceptions, licking or paw movements EDT distributions were not statistically different from lever pull time ETDs, however, due to the high correlation between paw movements and licking to lever-pull times – we do not view them as completely independent analyses. Importantly, as we show below (see Figure 8) when considering only body movements that are isolated from lever pulls (i.e., not preceded by a lever pull in the previous 15 s, or following 5 s) and also not preceded by other body movements for at least 5 s, we find that [Ca] averages show an increase in motor cortex (and other areas) and that EDTs are near 0 s as the signal is too noisy to enable decoding (see also Methods and Discussion).

In sum, upcoming self-initiated behaviors in mice can be decoded above chance several seconds prior to movement for all animals with a strong dependence of the EDT on the number of trials present in each session. While sequential stereotyped pulls have the effect of lengthening EDTs, longitudinal changes in the neural recordings have the effect of shortening EDTs. Single anatomical area analysis revealed that somatosensory-limb cortex contained the most information about upcoming movements but that decoding information was distributed across multiple regions of cortex.

Studies of human voluntary behavior have shown that SCPs, that is slowly changing voltages measured usually via EEG <1 Hz, might be involved in modulating voluntary behavior (**Jo et al., 2013**; **Schmidt et al., 2016**). In particular, voluntary behavior was found more likely to occur (on the order of ~10%) when the SCP phase over motor areas was near the crest. Additionally, there is some evidence to support that the SCP is related to awareness or consciousness and may play a causal role in internal state driven action (**He and Raichle, 2009**; **Northoff, 2017**). Given these findings we sought to determine whether self-initiated lever pulls in mice co-occurred with specific phases of widefield [Ca++] activity (**Figure 3**).

## Lever pulls occur during narrowly distributed phases of neural activity

As shown above (see **Figure 1**, **Figure 1—figure supplement 3**) within single cortical areas, the neural dynamics preceding rewarded lever pulls become increasingly stereotyped closer to $t = 0$ s (**Figure 3a** – top for an example of left forelimb neural activity). Fitting sinusoids to the last 5-s period prior to movement in each trial yielded sinusoidal fits with very similar phases at $t = 0$ s (**Figure 3a** – bottom). Neither of these stereotyped dynamics were present in random segments on neural activity (**Figure 3—figure supplement 1**). The phase distribution for a single session and cortical area was narrowly distributed with most phases falling in a <90° wide window in many sessions (and <45° for some sessions) (see **Figure 3b** for example distribution from phases in **Figure 3a**; see also **Figure 3—figure supplement 2** for examples from all cortical areas). In contrast, random segments of neural activity had widely distributed phases (**Figure 3c**, **Figure 3—figure supplement 2**). Computing the $t = 0$ s phases for all trials across all sessions also revealed narrowly distributed phases (**Figure 3d**) in contrast to random segments of neural activity which yielded an approximately uniform distribution (**Figure 3e**). Similar human studies where the most likely SCP phase during voluntary action was the crest phase with ~30% probability followed by the rising phase (**Figure 3f** for an adapted example), phases in mice were also biased to these two locations (i.e., crest or rising phase). In contrast to human results, we found an even higher bias as some mice had more than a 50% probability of initiating action in the crest phase – while others preferred the rising phase (see **Figure 3g** for two examples). The phases for all mice, sessions, and trials showed that virtually all mice and cortical areas had narrowly distributed phases (**Figure 3h**) significantly different than random segments of neural activity (**Figure 3i**; Rayleigh test for uniformity $<<1E^{-5}$ for all animals and areas). The diversity of phase preferences was present not only between mice but also within mice as the phase bias could be significantly different between cortical areas (e.g., **Figure 3h**, mouse M1 limb vs. motor cortex phase differences). Lastly, the median inter-area phase correlation varied with some mice having highly correlated phases across areas whereas others having mostly low correlations (**Figure 3j**; see also Methods).

In sum, the phases of neural activity at lever-pull initiation from all areas were significantly stereotyped, consistent with human findings (**Jo et al., 2013**; **Schmidt et al., 2016**). In contrast to human findings we found a higher bias of phases: that is, behaviors were even more likely to occur during a specific phase in mice than in humans. These findings support the presence of biases in

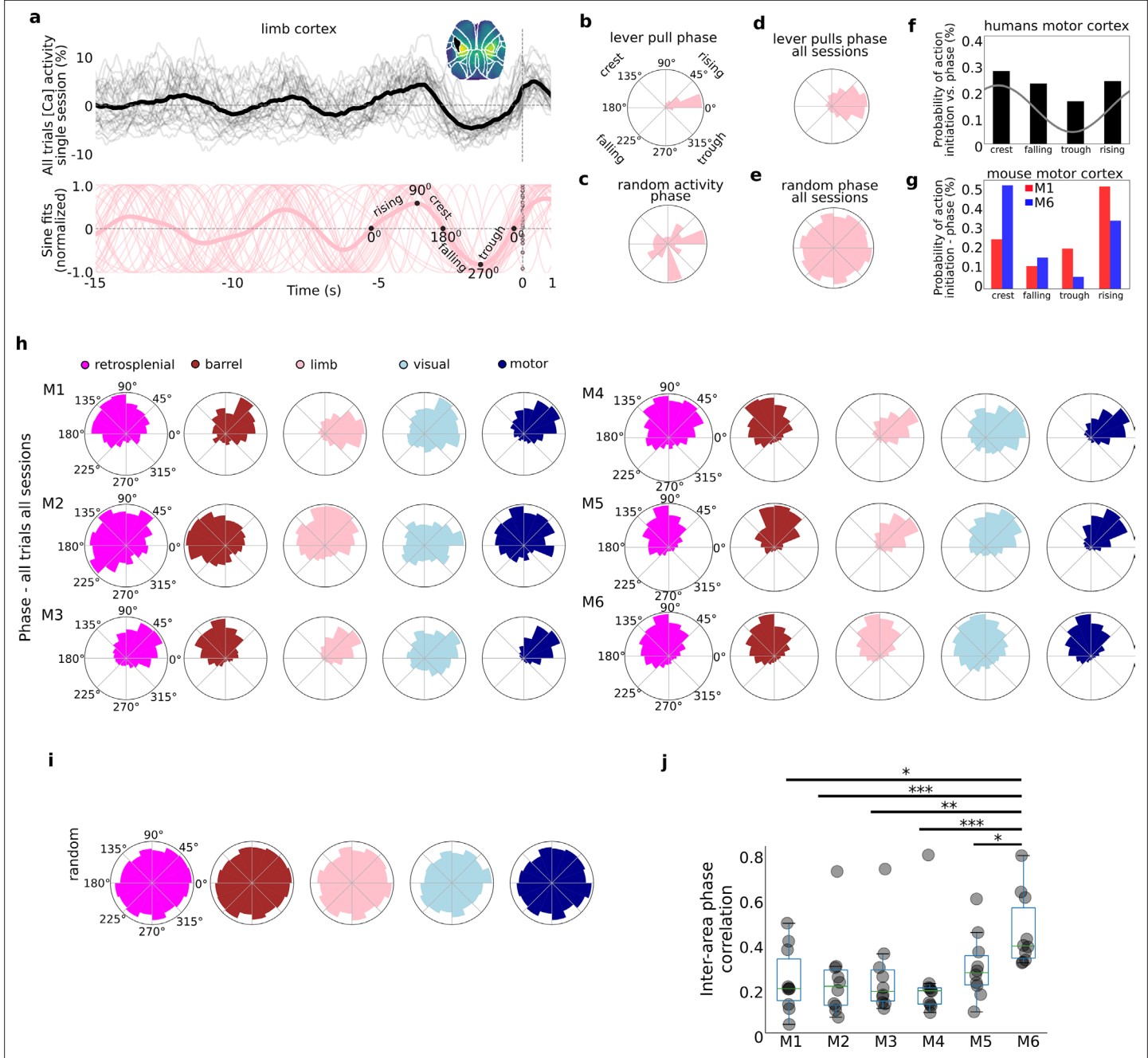

**Figure 3.** Self-initiated lever pulls occur during narrowly distributed slow-oscillation phases. Self-initiated behaviors occur during specific phases of slow-oscillations. (**a**) Top: single-trial neural activity (gray curves) from 33 trials in a single session (M4) for somatosensory-upper left limb cortex contain oscillations that become increasingly stereotyped closer to lever-pull time ($t = 0$ s; thick black curve is session average; inset shows anatomical area selected); Bottom: single-trial sinusoidal fits (thin pink curves) to neural activity (in Top) and phases (scatter dots on the $t = 0$ s line; thick pink curve is session average). (**b**) Polar plot of results in (**a**) showing the distribution of sinusoidal fit phases at $t = 0$ s. (**c**) Same as (**b**) but for random periods of neural activity (i.e., not locked to any behavior). (**d**) Same as (**b**) but for all sessions in mouse M4. (**e**) Same as (**d**) but for random periods of neural activity across all sessions in mouse M4. (**f**) Probability of voluntary action in humans during various phases of the slow cortical potential (SCP). (**g**) Same as (**f**) but widefield [Ca++] from the motor cortex in mice M1 and M6. (**h**) Phase distributions across all mice, sessions, and areas. (**i**) SCP phase distributions for random segments of neural activity for all areas in mouse M4. (**j**) Single-trial pairwise correlation between all cortical areas (e.g., limb vs. motor, limb vs. retrosplenial, etc.). (**f**) Has been adapted from Figure 2 from **Schmidt et al., 2016**. (pval * same as in **Figure 2—figure supplement 3**).

© 2016, Elsevier. Figre 3f is reproduced from Figure 2 from **Schmidt et al., 2016**, with permission from Elsevier (copyright year 2016, copyright holder Elsevier). It is not covered by the CC-BY 4.0 licence and further reproduction of this panel would need permission from the copyright holder

The online version of this article includes the following figure supplement(s) for figure 3:

*Figure 3 continued on next page*

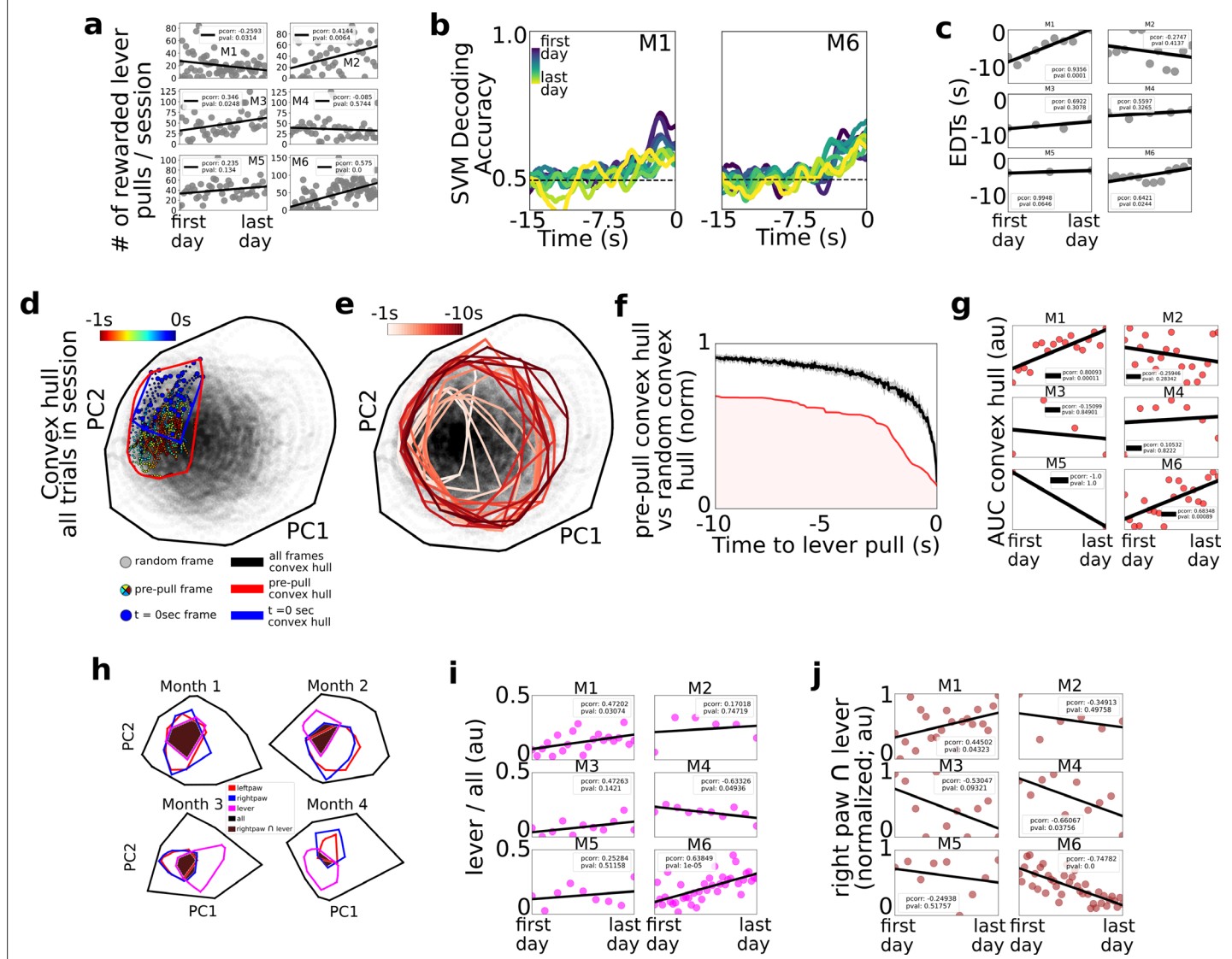

**Figure supplement 1.** Random controls for slow oscillation sinusoidal fits.

**Figure supplement 2.** Phases for lever pulls vs. random segments across major cortical areas.

**Figure 4.** Changes in decoding and neural dynamics over weeks of task performance. Tracking changes in decoding performance and spontaneous neural dynamics over time. (**a**) The number of rewarded lever pulls per session in all mice over the duration of the experiment. (**b**) Support vector machine (SVM) decoding accuracy curves from M1 and M2. (**c**) Earliest decoding times (EDTs) (black dots) for all mice (using 15-s lockout trials only black lines: linear fits; inset: Pearson correlation coefficient; note: the EDTs were computed on non-lockout data). (**d**) Convex hull of neural activity preceding lever pulls from 30 trials from a single session in mouse M3. The colored scatter points represent neural activity at various frames in the −1 to 0 s period and the polygons represent the convex hull of neural activity for all data (black), $t = 0$ s (blue) and $t = −1$ s to $t = 0$ s (red; see also legend for details). (**e**) Same as (**d**) but for 1 s segments ranging from −10 s to $t = 0$ s. (**f**) Convex hull volume (red line; shading represents area under the ratio curve [AUC]) of pre-pulls vs. random segments (black line; shading represents 10-fold sampling standard deviation; see also Methods). (**g**) AUC (scatter points) of ratio curves (as in (**f**)) for all animals and sessions. (**h**) Convex hull of 1 s of neural activity preceding body movements (red and blue polygons), lever pull (magenta polygons), and all neural data (black polygons) at different longitudinal time points in the experiment for mouse M6. Brown shaded regions represent the overlap between the right paw movement initiations and lever pull. (**i**) Ratio of convex hull of lever pull to all neural activity (magenta scatter points), linear fit (black line), and Pearson correlation value (inset). (**j**) Intersection of right paw and lever-pull initiation space (brown scatter points), linear fit (black lines), and Pearson correlation (inset).

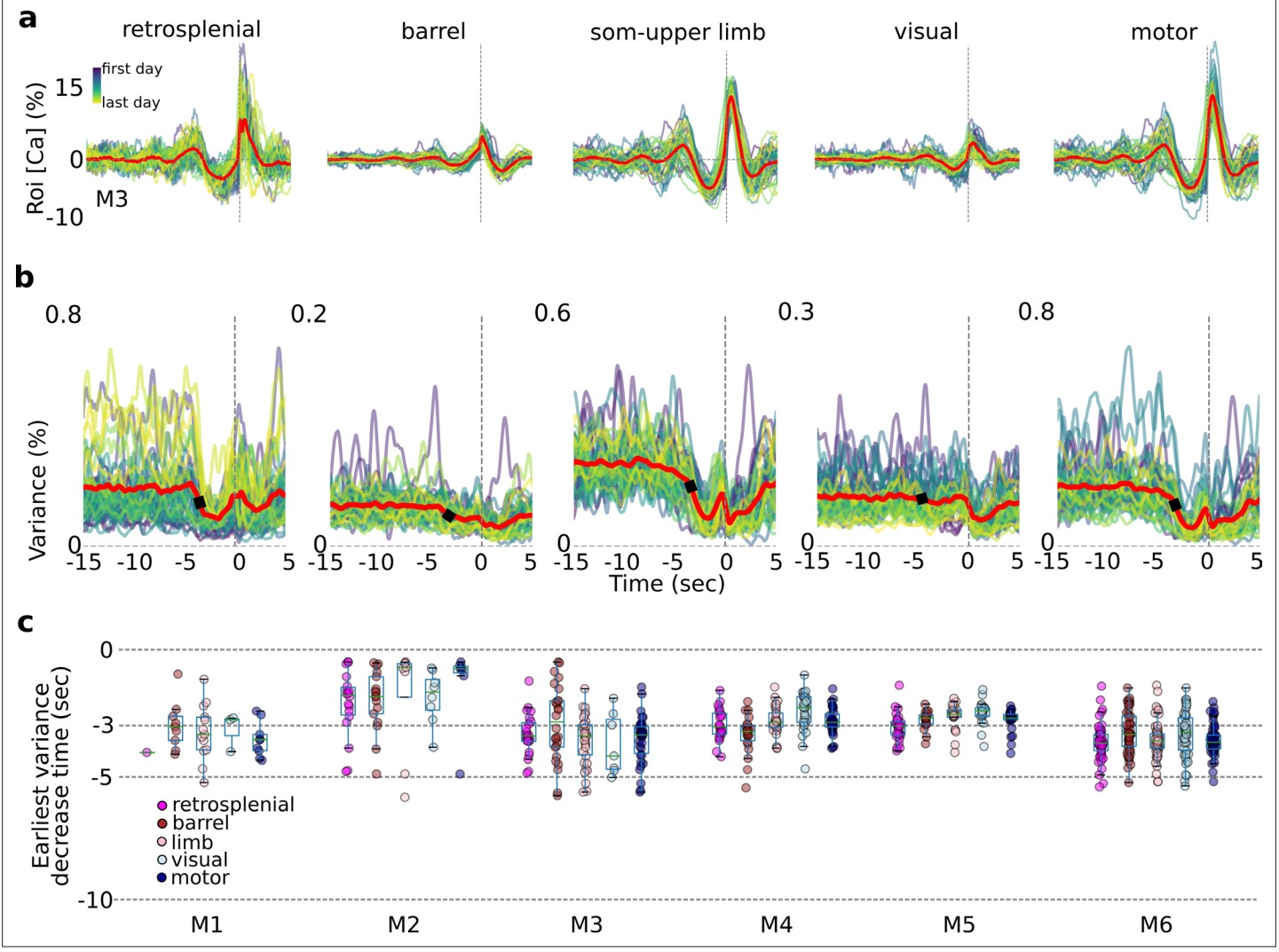

**Figure 5.** Single-trial variance changes seconds prior to self-initiated lever pulls. Internal-state evaluations begin several seconds prior to self-initiated lever pulls. (**a**) Average session neural activity from cortical areas (colored curves) and average over all sessions (red curve) from mouse M3. (**b**) Variance of data in (**a**) and the earliest variance decrease time (EVDT; black dots represent EVDT of all session averages, i.e., red curves in (**a**)). (**c**) EVDTs for all animals and sessions.

neuroanatomical dynamics preceding self-initiated behavior preparation while confirming inter-animal differences in both anatomy and dynamics observed in other findings (see also *Figures 4 and 5*).

We next sought to determine whether learning or mere longitudinal performance of the task changed the decoding accuracy or cortical neural dynamics during voluntary behavior initiations (*Figure 4*).

## Longitudinal cortical network dynamics support shortening of EDTs

Considering the number of rewarded lever pulls per session, we found that between the first and last days of the experiment three of the mice (M2, M3, and M6) increased the number of rewarded lever pulls while one additional mouse (M5) also had a positive trend (with pval of 0.13); one of the mice (M1) decreased its number of pulls per day and the remaining mouse did not have statistically significant changes (*Figure 4a*; Pearson correlation values provided in figure insets; see Methods). We labeled the four mice with either a strong or a trend in positive correlation over time as the 'performer' group (M2, M3, M5, and M6) and the remaining mice (M1 and M2) as 'non-performers' as they did not increase their pulls over time. Given the potential confounds identified in *Figure 2*

between sequential lever pulls and decoding time, further analysis in this section focused primarily on 15-s lockout trials only (see *Figure 2*). We found that SVM decoding accuracy curves over the weeks or months of behavior revealed potential trends over time (*Figure 4b* for examples from two mice). In particular, we found that EDTs shortened (i.e., were closer to the lever-pull time) over time in two mice (M1 and M6); a similar trend was present in another mouse (M5; pval 0.06); while the remaining three mice did not show statistically significant trends. Although only statistically significant in two mice, shortening of EDT decoding time trends may be explained by automaticity findings in other studies: that is, that following learning and repetitive behavior performance, the control of behavior is transferred from cortical structures (which we had access to during widefield [Ca] imaging) to subcortical structures (that we could not access in our paradigm; another explanation could be that implants slowly degraded) (see e.g. *Ashby et al., 2010*; see also Discussion).

## Lever-pull neural activity space increases its cortical representation over time in some mice

We implemented a convex-hull-based analysis to capture how the neural activity *space* prior to lever pulls changed over weeks or months of performance (*Figure 4d–g*; see Methods). For each session, the neural activity *convex hull* at lever-pull time (i.e., $t = 0$ s) was defined as the hyper-volume that enclosed the $t = 0$ s neural activity vectors relative to all the neural activity vectors in the session (note: as convex hull analysis is sensitive to outliers a 10% *K*-nearest-neighbor triage was implemented prior to evaluation; see Methods). The convex hull could be visualized in two dimensions using principal component analysis (PCA) as the area enclosing the $t = 0$ s neural activity vectors for lever pulls in that session (*Figure 4d* – blue dots; see also Methods). Both the convex hull of the $t = 0$ s and $t = -1$ s period prior to lever pull occupied a small subspace within the entire neural space of all the activity in the session (*Figure 4d* – colored dots; see Methods). Going backwards in time, the convex hull space gradually increased further away from $t = 0$ s (*Figure 4e*). This suggests that neural activity looks more like spontaneous (i.e., random) activity further in time from lever pulls and becomes more stereotyped toward $t = 0$ s. The ratio of the pre-pull convex hull to the hull of the entire session was smaller than the hull computed from random trigger times (*Figure 4f*).

Lastly, we evaluated the area under the ratio curve (AUC) longitudinally to evaluate whether there are systematic changes in the neural activity convex hull over weeks of behavior performance (*Figure 4g*). We carried out this analysis using only 15-s lockout data grouped in sessions of up to 200 trials (similar to carried out above to exclude any possible trends arising from increased intra-session lever pulls or effects of sequential lever pulls; see Methods). We found that, as in the EDT longitudinal trends, two mice (M1 and M6) that had decreasing EDTs (i.e., poorer decoding over time) also had an increased similarity (i.e., increased AUC values) between lever-pull dynamics and random neural states. Considering only these two mice (as they were the only statistically significant results), one explanation may be that cortical dynamics may return to pre-lever-pull learning patterns and look increasingly the same as random neural states (occurring near or far from behaviors) because subcortical circuits increasingly facilitate and 'take over' self-initiated behavior preparation during automaticity processes.

## Neural activity space of right paw movements and lever pulls change systematically over time in some mice

Given the findings above we sought to further evaluate systematic changes between lever-pull and random neural activity (*Figure 4h–j*). We recomputed convex hulls for the 1-s period prior to lever pull, 1-s period prior to left and right paw movements and found that the size of the lever-pull convex hulls and its overlap with the rest of the behaviors changed over time (*Figure 4h*). While five of the six mice had increasing convex hulls, the trends were statistically significant in only two of the mice, which again, were mice M1 and M6 (although mouse M3 also showed a similar trend, p value: 0.14) (*Figure 4i*). Interestingly, mouse M4 showed a decrease in overlap of lever dynamics with all dynamics. These mixed results suggest that different mouse-specific mesoscale neural representations may be involved during learning and performing of a task.

We also found mixed results with respect to the intersection between the right paw convex hull, that is the paw used to pull the lever, and the lever-pull convex hulls: the overlap decreased with time in two of the mice (M4 and M6) and showed a similar trend in another mouse (M3; pval 0.09); and it increased with time in mouse M1. These results suggest that in some mice (M4, M6, and possibly M3) lever-pull neural dynamics increasingly specialize or differentiate from non-lever-pull right paw movements neural dynamics (despite the right paw being used for the lever-pull task). In contrast, in one mouse the similarity between right paw movements and lever pulls increased (e.g., mouse M1), but this could be explained by this mouse being a behavior outlier as the only mouse with decreased number of rewarded lever pulls over time.

In sum, EDTs shortened longitudinally in some mice suggesting neural dynamics underlying self-initiated behavior might be transferred from cortical to subcortical circuits decreasing the power of cortical-based decoding methods. The convex hull of the neural activity prior to self-initiated lever pulls also increased over time in some mice with a potential explanation that cortical dynamics return to pre-learning similarity over time. Lastly, right paw and lever dynamics appeared to become increasingly dissimilar in a few mice, with one mouse showing the opposite trend. These findings suggest that learning or mere longitudinal performance of a task restructures the neural dynamics underlying self-initiated action but that the effects could be subject specific, drawing attention to the need for intra-animal analyses (rather than cohort) in future studies.

Over the past few decades, one of the most robust findings in stimulus cued decision making studies has been that stimulus onset decreases neural variability in a wide range of paradigms (see *Churchland et al., 2010* for a summary). This decrease in variance, also evaluated as the fano factor (i.e., the ratio of variance to mean of neural activity) has been interpreted to suggest that incoming information (i.e., stimuli) 'stabilizes' the state of cortical activity (e.g., decreases the variance of membrane potential fluctuations, spiking variance, or correlated spiking variability) and potentially supports the accumulation of internal memory evidence (*Ratcliff, 1978*; *Ratcliff and McKoon, 2008*). Here, we sought to determine whether neural activity preceding self-initiated lever pulls exhibited a change in variance prior to the decision to pull, potentially reflecting the commencement of evaluation of 'internal-state' evidence and/or the preparation of a skilled action (*Figure 5*).

## Single-trial variance decreases seconds prior to self-initiated action

As shown above for single trials (*Figure 3a*), the average trial activity within a session becomes increasingly stereotyped near *t* = 0 s. This stereotypy can be observed in all major cortical areas even across weeks or months of behavior (*Figure 5a*). Interestingly, the variance was also stereotyped with a significant change (decrease in five mice; increase in one mouse: M2) several seconds prior to the movement-related areas (e.g., motor, somatosensory, and retrosplenial) but less so in areas not directly related to movement preparation (e.g., visual cortex; *Figure 5b*). We defined the earliest variance decrease time (EVDT; see Methods) for each session as the time at which the variance decreased (or increased for mouse M2) by two times the standard deviation from a random period of time (*Figure 5b* – black dots; see Methods). Computing the EVDTs for all sessions and mice revealed that in most areas and animals the neural activity variance began to decrease a few to several seconds prior to lever-pull initiation. While these measurements were noisy (see Methods), in mice with significant numbers of detected EVDTs (i.e., mice M3, M4, M5, and M6) the average EVDT for all areas was around −3 s or earlier. Such decreases in variance seconds prior to behavior initiation may represent the times at which internal state evaluations and motor preparation commences (see also Discussion; we note that this analysis was not possible using lockout trials due to significantly higher intra-session variance caused by pooling data from across sessions days and weeks apart; given EDTs ranging from approximately −3 to −7 s for lockout data we anticipate that given sufficient numbers of 15-s lockout trials we would obtain similar or shorter EVDT times as in non-lockout data).

In sum, we found that in most mice and sessions, variance across all areas (excluding visual cortex) began to change several seconds prior to lever-pull time. These results are consistent with and support our prior findings and could be interpreted to suggest that internal-state driven behaviors are underpinned by neural processes similar to those observed in stimulus driven decision making studies (and as observed in *Murakami et al., 2014* in neural activity preceding a self-paced task).

An outstanding question in voluntary behavior neuroethology is how confounds, such as random body movements occurring in the period prior to a targeted action, affect behavior preparation and – in our paradigm – the decoding of future behaviors. For example, in animals that perseverate and pull the lever frequently it is not known whether decoding methods leverage dynamics from multiple lever pulls or just the lever pull occurring at $t = 0$ s. Accordingly, we sought to evaluate the effect of preceding lever pulls and/or body movements on the decoding of future lever pulls.

## Rewarded lever pulls form a small portion of all self-initiated actions

Across the duration of the study mice performed between 1454 (M1) and 6999 rewarded lever pulls (M6) (*Figure 6a*). The ratio of self-initiated rewarded and non-rewarded lever pulls, licking events, and left and right paw spontaneous movements for all sessions revealed that lever pulls constituted only a small portion of behaviors (*Figure 6b*). Across all animals the proportion of rewarded lever pulls compared to all other movements ranged from 0.03 (M1) to 0.07 (M2) (*Figure 6c*). Even though only selected body parts were tracked (i.e., paws and tongue), these results suggest the vast majority of the time mice are engaging in carrying out many other spontaneous behaviors as reported in other studies (e.g., *Musall et al., 2019*).

## Removing pre-action confounds by implementing a post hoc lockout period

To evaluate the effect of intervening body movements in decoding future actions, we evaluated decoding upcoming rewarded lever pulls or spontaneous body movements preceded by periods of quiescence of varying durations (*Figure 6d–h*). All trials across all sessions were pooled (similar to the concatenated analysis in *Figure 2*) and decoding was carried out after 'locking-out' previous (1) lever pulls (i.e., rewarded and non-rewarded), (2) licking events, or (3) left paw movements (note: we selected left paw movements as the right paw was used to pull the lever and excluding such movements from analysis would remove most of the rewarded lever-pull trials). The number of rewarded lever pulls preceded by periods of non-body movements or lever pulls decreased approximately exponentially with increasing lockout duration (*Figure 6d*). This decrease in available trials for analysis was present in all mice (*Figure 6e*). Importantly, locking out licking and lever paw movements beyond 3 s yielded insufficient numbers of trials for decoding analysis (see shaded region *Figure 6e*).

## Lever-pull EDTs are not affected by prior licking events

EDTs were recomputed by locking out (i.e., removing) lever pulls that were preceded by licking events in the previous 0, 1, 2, and 3 s (note: 0-s bin removed lever-pull trials that occurred exactly with a licking event based on our video recording resolution of 15FPS). We found that for all animals, the intra-animal EDT distributions were not statistically different from each other (*Figure 6f*).

## Evaluating lever-pull EDTs vs. lever-pull lockout duration

We also recomputed EDTs for each animal and session after enforcing periods of 3–15 s of lockout (in increments of 3 s) (*Figure 6g*). EDTs for two mice (M1 and M2) stayed the same, while for four mice they shortened with increasing lever-pull lockout duration (M3–M6) (*Figure 6g*). This trend was confirmed by examining the mean EDT at each lockout time point which showed in nearly all mice (excluding M2) a strong trend for the mean EDT to shorten with increasing lockout (*Figure 6h*). In interpreting these results it is important to note that there were substantially fewer consecutive or same-session lever-pull trials when implementing increasingly longer lockouts. Thus, surviving trials used for analysis came from sessions that were increasingly further apart (e.g., multiple days or even a week). Pooling trials from separate days or weeks provides an additional source of noise due to changes in animal behavior, [Ca++] indicator properties, and longitudinal network changes observed in our cohorts (see *Figure 4*). As evidence for this, we found that subsampling the number of non-lockout trials to match

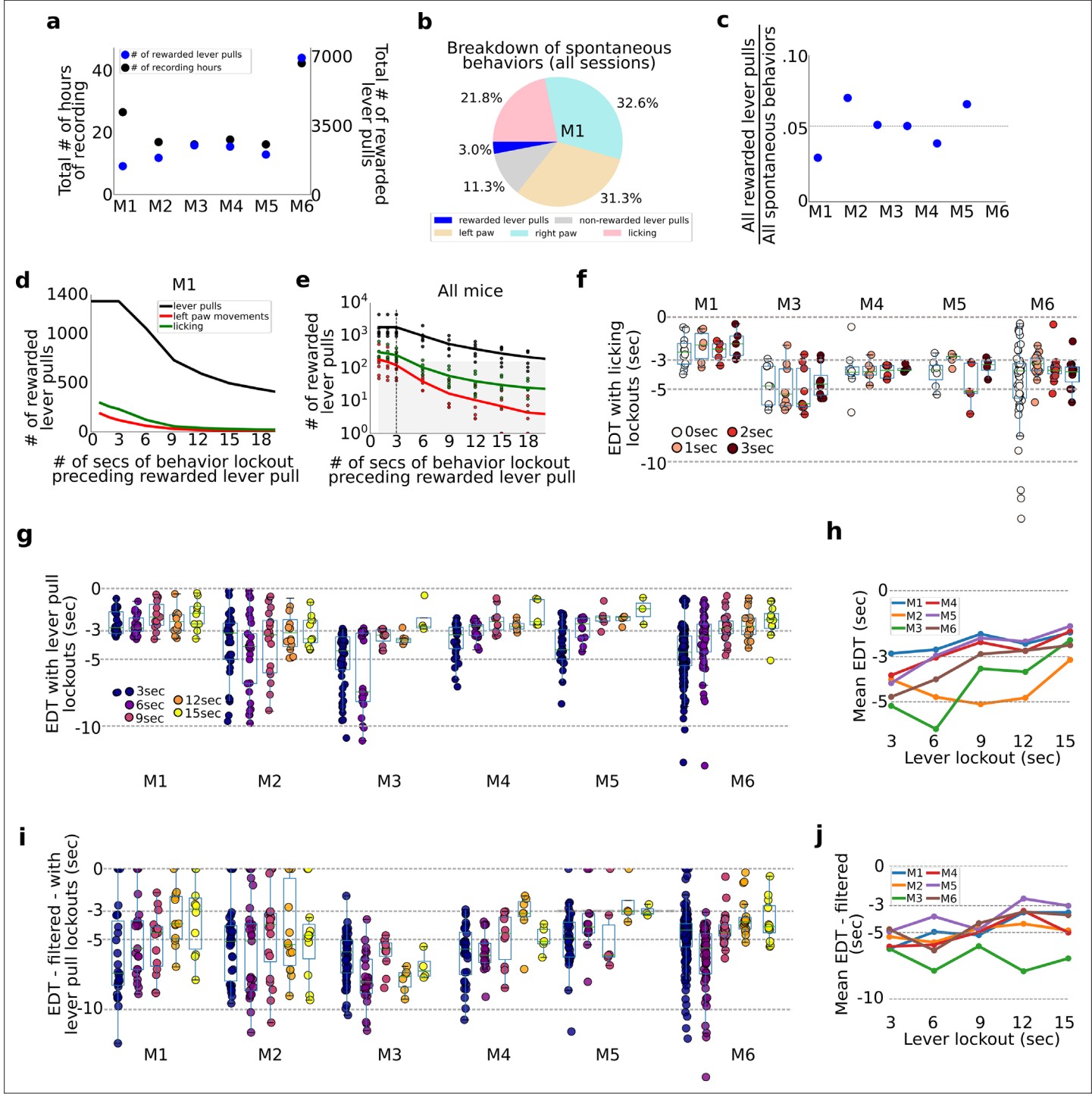

**Figure 6.** Evaluation of pre-ever pull movements and behavior lockout on earliest decoding times (EDTs). Tracking and evaluating the effects of previous self-initiated movements on the decoding of rewarded lever pulls. (**a**) Total number of recording hours and number of lever pulls for each mouse (note: each recording session was approx. 22 min long). (**b**) Percentage of rewarded lever pulls, non-rewarded lever pulls, and left paw, right paw, and licking movements performed by mouse M1 across all sessions. (**c**) Proportion of rewarded lever pulls relative to all other body movements for all animals. (**d**) The number of rewarded lever pulls as a function of 'locking out' previous lever pulls, licking events or left paw movements (note: locking out means excluding any rewarded lever-pull trial that was preceded by a movement in the previous *n*-seconds; see also Main text and Methods). (**e**) Same as (**d**) but for all animals and sessions (shaded region indicates lockout conditions under which less than 100 trials were present across the entire study and decoding was not carried out). (**f**) EDTs for rewarded lever pulls conditioned on licking event locking out periods of 0–3 s (for clarity, the 0-s time point excluded any rewarded lever pull that occurred precisely at the same time as a licking event, i.e. to the resolution of our 15FPS video). (**g**)

*Figure 6 continued on next page*

*Figure 6 continued*

Same as (**f**) but conditioned on excluding previous lever pulls (3-s time point excluded any rewarded lever pulls that occurred exactly 3 s after a previous rewarded or unrewarded lever pull). (**h**) Mean EDTs from (**g**) as a function of lever-pull lockout period. (**i**) Same as (**g**) but following low-pass filtering of the locaNMF components. (**j**) Same as (**h**) but for data in panel (**g**).

the number of trials following 15-s lockout had the effect of shortening most EDTs (*Figure 2—figure supplement 3*).

We also recomputed locked-out EDTs for each animal (as in (**g**)) but following low-pass filtering the neural time series (filter set to 0.3 Hz; see Methods) as described above (*Figure 6i*). We found that average EDTs detected were longer than using non-filtered data (*Figure 6j*; e.g., 15-s lockout data means: M1: −3.50; M2: −4.85; M3: −6.95; M4: −5.01; M5: −3.0; M6: −3.70).

Taken all these factors into account, the results suggest that while EDTs shortened when using only lockout trials, the cause of the increase is due to: (1) removing stereotyped sequential lever pulls which could artificially bias the neural signal; and (2) increased variance in the longitudinal data caused by [Ca++] state changes, systematic neural network restructuring due to longitudinal performance and other unknown factors (see also Discussion).

In sum, we find that despite carrying out thousands of rewarded lever pulls, such pulls constituted a small percentage of overall spontaneous body movements (likely much less if we consider other body movements we did not track). Even when pooling all trials from each animal (resulting in thousands of trials), when locking out previous licking events or left paw movements, exponentially fewer rewarded lever pulls were available with increasing locking out period. EDTs computed when excluding previous lever pulls for a period of up to 15 s prior shortened EDTs in most animals, and average EDTs for band-passed neural data ranged from approximately −3 to −7 s. We conclude that it is critical to exclude sequential lever pulls to the computation of EDT as well as track and control for longitudinal changes in neural dynamics caused by learning or implant-related causes.

While the role of SCP in modulating voluntary behavior is suggested in some human studies (*Schmidt et al., 2016*), less is known about the specific frequencies involved in self-initiated action especially in mice. We thus sought to further characterize the frequency, power, and longitudinal characteristics of slow oscillations in neural activity occurring prior to lever pulls (*Figure 7*).

## Limb and motor cortex oscillations have the highest power during pre-movement neural activity

We first evaluated the power of neural activity in a session (i.e., average of neural data from all trials from −15 to 0 s) and observed that high amplitude oscillations were present in some areas (e.g., limb cortex) but were much weaker in other areas (e.g., visual cortex) (*Figure 7a*). This is consistent with our self-initiated behavior task as no sensory stimuli or cues were used. This difference was consistent across all sessions with limb cortex oscillations being up to 10 times larger than those in visual cortex (*Figure 7b* examples from mouse M4).

## Frequency and power in the average pre-movement neural activity

We evaluated the peak frequency of session averages (i.e., we computed the power-spectrum-density of the lever-pull trial average for each session; ses also Methods). Across all mice the vast majority of session averages had power peaks falling between 0.2 and 0.6 Hz (*Figure 7c*). This suggests that slow oscillations dominated the pre-movement neural activity consistent with our findings that self-initiated action preparation unfolds on time scales of several seconds (and consistent with the time course of our [Ca++] indicator). Turning to longitudinal trends, few statistically significant trends were present with only three mice showing correlations of peak frequency and time (*Figure 7c*; mouse M1: strong increase in peak frequencies in limb and motor cortex; mouse M2: strong decrease in peak frequencies in limb and motor cortex; and mouse M5 had an increase in peak frequency in retrosplenial cortex). The peak frequency power also exhibited differences between animals and also longitudinally (*Figure 7d*). For example, mouse M1 showed significant drops in power in limb, motor, and retrosplenial cortex, while other mice showed increases in power in limb or motor cortex (M2: limb; M3:

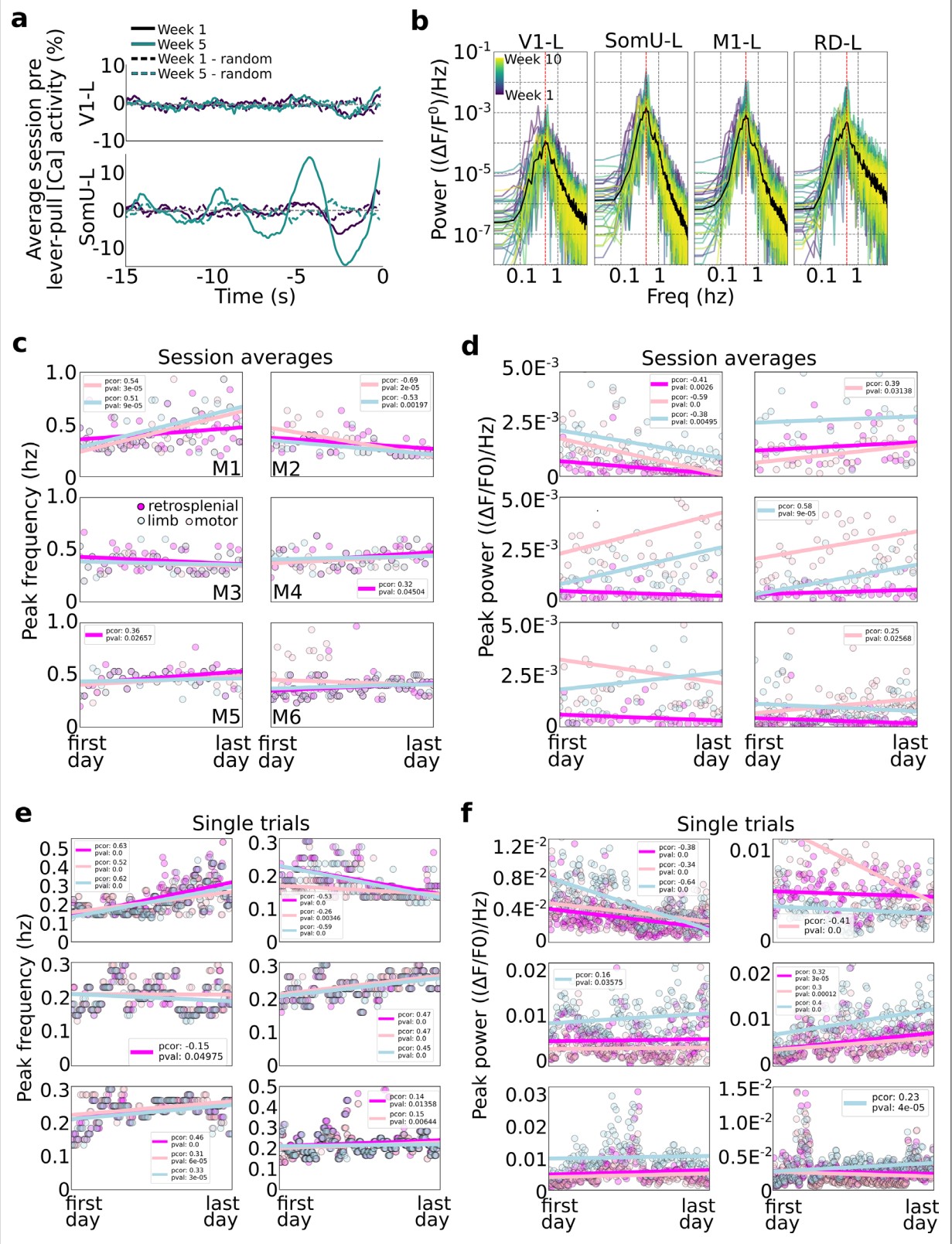

**Figure 7.** Slow oscillations dominate pre-self-initiated behavior neural dynamics. Slow wave oscillations underlie self-initiated behavior in mice. (**a**) Examples of single session averages (dark continuous curves) from two time points and random segments (dark dashed curves) from V1-left and somatosensory-upper limb left reveal the presence of oscillations. (**b**) Power spectra of all session averages (colored curves) and average across all sessions (black curves) from mouse M4 in four cortical areas (dashed vertical lines indicate peak of average). (**c**) Peak frequency power of each session

*Figure 7 continued on next page*

*Figure 7 continued*

trial average for retrosplenial, motor, and limb cortex (colored scatter points). (**d**) Same as plot in (**c**) but for peak power for all animals and sessions. (**e**) Same peak frequency analysis as in (**c**) but for single lever-pull trials (instead of session averages). (**f**) Same peak power analysis as in (**d**) for single trials.

motor; M4: motor; M6: limb). In sum, longitudinal changes in peak frequencies and power were minor and differed across animals.

## Frequency power in single-trial pre-movement neural activity

We carried out a similar analysis as above but on an individual trial basis (*Figure 7e, f*). With respect to peak frequencies, distributions of frequencies from ~0.1 to ~0.5 Hz were observed, similar to session averages. In contrast with session averages, single-trial analysis showed more statistically significant trends (p values <0.05) in most animals and areas considered (retrosplenial, limb, and motor cortex): three animals had mid to strong-level increases in peak frequency (in all areas) with time (M1, M4, and M5); one animal (M6) had slight increases in frequency power in retrosplenial and limb cortex; and one animal has mid to strong decreases in peak frequency over time (M2) (mouse M3 had a −0.01 Pearson correlation value with time in limb cortex). Peak power trends were less common, with only two mice showing strong correlations in the three areas longitudinally (M1 decreases in power over time; M4 increases in power over time) with the remaining mice having changes only in a single area (mouse M2 showed strong decrease in power in limb cortex; and mice M3 and M6 showed a slight increase in motor cortex power over time).

In sum, during self-initiated behavior preparation power in both session averages and individual trials was strongest in the 0.1–0.7Hz. Some animals showed systematic changes in peak frequency suggesting that learning and/or longitudinal performance may change the underlying oscillatory structure of neural activity of self-initiated behavior preparation. These findings suggest a complex picture with different mice potentially engaging different learning mechanisms and areas that should be considered when evaluating in-session and longitudinal performance and decoding upcoming behaviors.

Given the results in *Figures 1–7*, namely, that prior to rewarded lever pulls there are systematic changes in cortex, for example inhibition of motor activity, on the order of several seconds prior to movement we sought to relate our results to human EEG studies of volitional action. In human volitional studies the motor cortex EEG signal preceding spontaneous body movements (e.g., flicking a wrist, pressing a button) generally shows an increase in activity commencing between −2 and −1 s prior to the body movement. To compare our results to human studies we evaluated the [Ca] activity preceding isolated random left forelimb movements, that is movements that were not related to lever pulls nor preceded by other movements for several seconds.

## Identifying left paw movements isolated from other movements

We first computed the locations of all lever pulls relative to every left paw movement in windows of 20 s with 15 s of pre-paw movement and 5 s of post-paw movement (*Figure 8a*). Across all video recorded sessions we identified between 13,757 and 124,397 left paw movement bouts in individual mice (i.e., times where the left paw moved; see Methods; number of movements in all mice: M1: 54016; M2: 16241; M3: 18186; M4: 20487; M5: 13757; M6: 124397). We then ranked every left paw movement bout by the longest period of non-lever-pull activity, that is we ranked paw movement bouts by the amount of lever pulls occurring in the previous 15 s or following 5 s (*Figure 8a*). We next extracted the trials with a complete lever-pull lockout (i.e., no lever pulls 15 s before or 5 s after) and added the locations of left and right paw movements and licking events (*Figure 8b*). We then realigned the surviving bouts by the longest period of body movement quiescence, that is we reranked the remaining paw movement again by quiescence (*Figure 8c*). Finally, we looked for left paw movements that were not preceded by any movements for at least 5 s (*Figure 8d*). This approach revealed between 96 and 557 left paw movements that had at least 5 s of no preceding body movements and were completely isolated from lever pulls (number of movements per mouse: M1: 416, M2: 96, M3: 133, M4: 167, M5: 177, M6: 557).

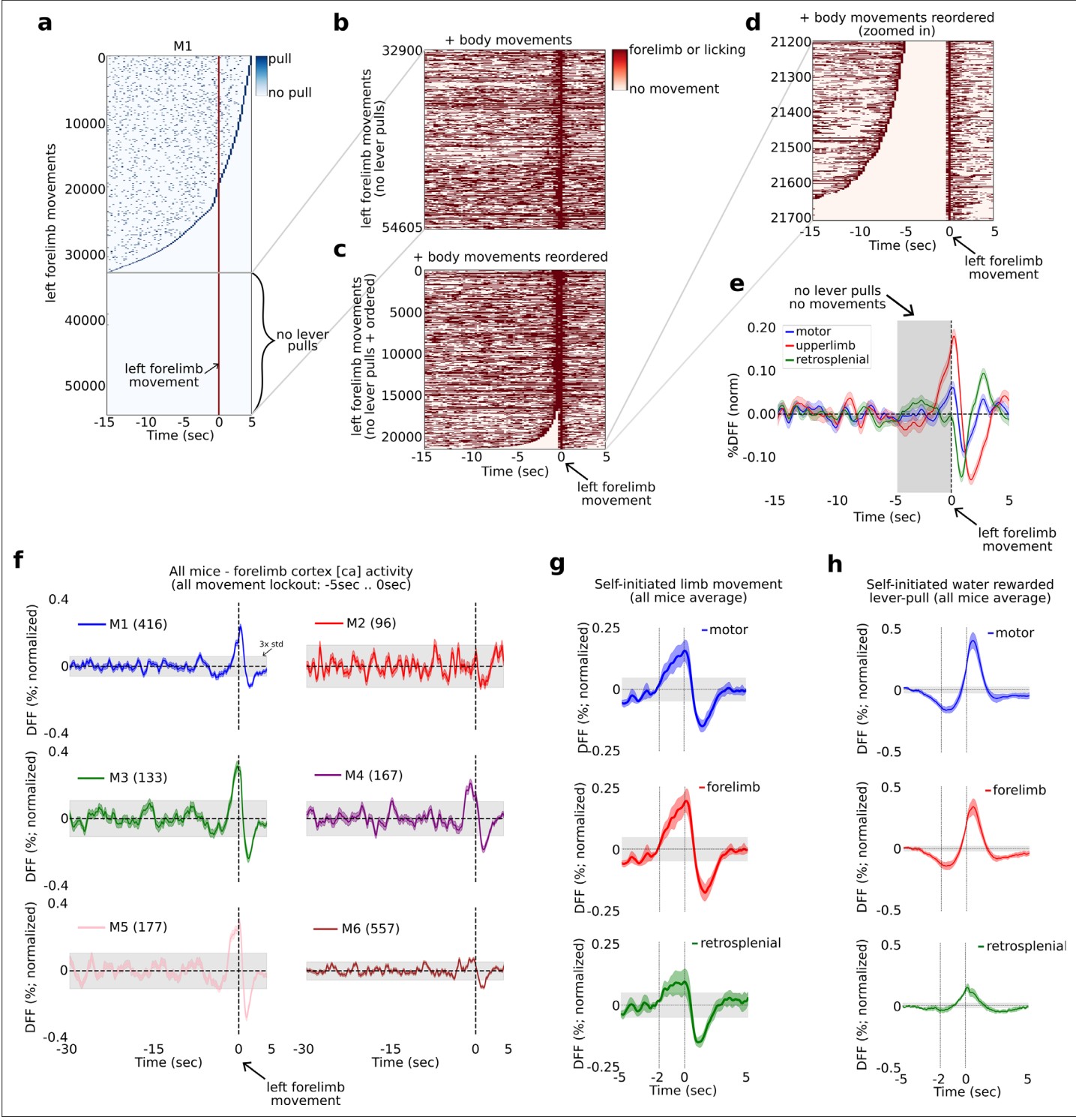

**Figure 8.** The widefield [Ca] activity correlates of self-initiated limb movement vs. goal-oriented actions. The widefield [Ca] activity correlates of random vs. goal-oriented self-initiated actions. (**a**) Raster plot showing the location of all lever pulls in mouse M1 across all sessions (blue rasters) relative to spontaneous left paw movements (red line at $t$ = 0 s) ordered by the duration of pre-paw-movement lever-pull quiescence. (**b**) The completely lever-locked left forelimb movements from (**a**) with added left and right forelimb movements and licking events (red rasters). (**c**) Same as (**b**) but ordered by the duration of quiescence prior to the left forelimb movement. (**d**) The bottom events from (**c**) showing the location of left and right forelimb movements and licking events (red rasters). (**e**) The average neural dynamics in motor cortex (blue), forelimb cortex (red), and retrosplenial cortex (green). (**f**) The average forelimb cortex activity (color shading represents standard error) for each mouse during periods as in (**d**) and (**e**). (**g**) The average neural activity (color shading represents standard error) from all mice averages as in (**f**) for motor, forelimb and retrosplenial cortex. (**h**) Same as (**g**) but

*Figure 8 continued on next page*

*Figure 8 continued*

for water-rewarded lever-pull activity. (Note: gray shading in all plots represents 3× standard deviation of the average neural signal between −30 and −5 s prior to movement or lever pull; see also Methods.)

### Neural activity increases 1–2 s prior to left-limb movement

We next computed the average neural dynamic for the right hemisphere motor, upper-limb, and retro-splenial cortex activity and found that both motor and upper-limb cortex exhibited a slow rising time course commencing at approximately −2 s prior to paw movement, with the upper-limb cortex signal being larger than motor cortex (*Figure 8e* for example from mouse M1). We computed the upper-limb cortex average signal for all animals and found that in four mice (M1, M3, M4, and M5), this signal showed a significant increase prior to movement (i.e., rising above 3× the standard deviation of the preceding neural activity) commencing approximately −2 to −1 s relative to paw movement time (*Figure 8f*). Mouse M2 had an average [Ca] signal which was noisy, likely due to the lower number of trials. Mouse M6 also had a noisy signal that showed an increase above 3× std prior to movement, but not as pronounced as the other four mice. We note that mouse M6 had more than 5 months of recording and the pooled trials likely reflected significantly more variance due to pooling bouts from longitudinal data (see *Figure 2—figure supplement 3* and Methods for a detailed discussion on the challenges of pooling widefield [Ca] data across long periods of time). We also sought to decode upcoming behavior-locked out paw movements but found that SVM-based methods did no better than chance (not shown). This was likely due to higher dynamics occurring during non-goal-oriented paw movements, but we also could not rule out the effect of pooling only a few hundred trials from tens of thousands of trials across many months of recording (see also Discussion).

Computing the average neural signals in motor, upper-limb, and retrosplenial cortex from the means in all mice we found that both motor and upper-limb cortex showed a significant increase in the average neural signal beginning at approximately 2 s prior to paw movement (*Figure 8g*). In contrast, the average neural signal in these areas prior to water-rewarded (locked-out) lever pulls showed a decrease commencing as early as −5 s prior to lever pull (*Figure 8h*).

In sum, we find that prior to non-lever pull related and isolated left paw movements, the [Ca] neural activity in mouse motor cortex begins to increase at approximately −2 to −1 s prior to paw movement – consistent with human EEG studies on the RP dynamics occurring prior to spontaneous finger or hand movement. In contrast, self-initiated water-rewarded lever pulls contain an inhibitory signal that starts earlier, at approximately −5 to −4 s prior to lever pull and contains stereotyped neural patterns that can be decoded to predict upcoming level pull timing (see *Figures 1, 2, and 6*).

## Discussion

Since the 1960s, several human neuroscience studies seeking to identify the neural correlates and genesis of self-initiated, voluntary action have found that increases in neural activity in SMA and pre-SMA precede both voluntary movement and even awareness of the intent to act (*Kornhuber and Deecke, 1965*; *Libet et al., 1983*). Most studies found only small differences (~150 ms) between the intent to act and voluntary action initiation; however, these findings remain controversial and deter-mining the precise arrival of subjective intent and the effect of reporting it is a complex topic (see e.g. *Wegner, 2002*; *Dijksterhuis et al., 2006*; *Tusche et al., 2010*; *Sinnott-Armstrong and Nadel, 2010*; *Dijksterhuis and Aarts, 2010*). Removing the reporting of intent from voluntary behavior paradigms and focusing solely on the relationship between neural activity and self-initiated action avoids some of the controversies and focuses the debate on the study of objective variables (e.g., timing of behavior initiation, neural activity in specific areas) – and enables the use nonhuman animal models for self-initiated and voluntary action research.

### Self-initiated actions in mice are prepared seconds prior to movement and are biased to occur during specific phases of slow oscillations

Using a self-initiated behavior paradigm in mice to relate pre-movement neural activity to the initi-ation of behavior enabled us to collect a high number of behavior trials across weeks and months

of recording and with higher neuroanatomical resolution recordings than EEG and higher temporal resolution than fMRI. Self-initiated behaviors in mice were preceded by a decrease in widefield [Ca] neural activity starting 3–5 s prior to behavior time, similar to (but longer) than the EEG RP signals in humans (*Figure 1*, see also *Kornhuber and Deecke, 1965*; *Libet et al., 1983*). We further found that decoding of upcoming behavior approximately 3–7 s prior to movement was possible, a finding consistent with findings using fMRI in humans (*Soon et al., 2008*; *Soon et al., 2013*; *Bode et al., 2011*; see *Figures 1 and 2*). Self-initiated behaviors were even more biased toward specific phases of neural activity than in humans (*Figure 3*). This suggests that in mice, oscillation phase could be even more determinative of action initiation timing than in humans. Overall, these results link human voluntary action studies with rodent self-initiated behavior studies and suggest mice as an adequate model for studying the neural correlates of self-initiated action.

## Behavior preparation signals are distributed across the cortex

While the vast majority of voluntary action studies in humans focused on SMA (and pre-SMA), we found that decoding future behaviors using motor (or limb cortex) was inferior to decoding using all cortical activity (*Figure 2*). This suggests that pre-voluntary movement neural signals are widely distributed across the dorsal cortex but also that more temporally and spatially precise neural recordings may be required for a complete characterization of pre-movement neural activity in humans (and nonhumans). Coupled with a neural recording modality that more precisely reports local neural activity (i.e., widefield [Ca++] cortex) our findings suggest that voluntary action studies in humans may benefit from subcranial neural signals from multiple areas.

## The role of motor cortex inhibition prior to lever pulls

Our paradigm shows that, on average, motor cortex is inhibited beginning approximately −5 s prior to lever pulls. In some mice and sessions, we found EDTs as low as −7 s which suggest the inhibitory (and other dynamics) commence even earlier. In parallel, we found that mice decreased, but not stopped altogether, random body movements during the period of −5 to 0 s. While our task did not require mice to cease body movements prior to lever pulls, it is possible that preparation of future pulls requires some mesoscale inhibition of motor cortex which necessarily leads to decreases in body movements. This hypothesis of delays in behavior on the order of seconds has some support from studies which show human subjects are more likely to initiate actions during exhalation periods which are generally 3 s in duration (*Park et al., 2020*), although mice breath faster than humans. Another interpretation is that mice consciously inhibit motor cortex in preparation of an action toward a water reward. (We use the term consciously, as we view intentional inhibition as achievable via unconscious systems.) For example, for every lever-pull mice consciously withhold pulling the lever prior to movement as they learn to 'count' several seconds prior to carrying out an attempt at the lever pull. In our opinion, this 'counting' might occur consciously or subconsciously and it would be challenging to establish in humans let alone in mice where we lack subjective report. In addition, such a strategy might lead to higher variances in the duration of inhibitory bouts – yet the neural dynamics of 15-s lever-pull lockout trials were sufficiently stereotyped to yield EDTs on the order of multiple seconds. Additionally, one mouse (M1) in our study did not internalize appear to learn the 3-s lockout rule and did not persevere at lever pulls at approximately 3-s intervals, yet had decodable EDTs. However, we cannot rule this possibility out with our dataset and paradigm. Such an interpretation could also call into question whether volitional actions toward a highly salient goal can be studied at all in the lab under self-initiated or self-paced paradigms. That is, it would raise the question of whether the study of volitional action toward high-value (rather than low-value) goals can only be carried out in natural settings where the opportunity for high-value decisions arises naturally from internal states rather than as part of an instructed paradigm.

A simpler explanation for our results is that the inhibitory dynamics are an integral part of preparation of a future action toward a salient goal and that they are present prior to action toward a salient goal in other paradigms. The duration and extent of such dynamics may be different than our paradigm where we required a 3-s lockout, but could be detected in both rodents and humans acting volitionally toward a highly salient goal.

## Comparison of our study with cued action studies

In our experience, mice following external cues such as a light or sound triggers initiate their behaviors immediately and decoding the timing of future behavior on the scale of seconds is not possible. However, it is possible that in cue-triggered experiments – expert mice exhibit motor cortex inhibitory signals while they are waiting for a cue (e.g., see *Schurger et al., 2012*). However, such neural states would be conditioned on the eventual arrival of an external cue toward a future action and would not constitute self-initiated volitional action. Another possible paradigm would be to train mice to withhold action following a cue for several seconds prior to movement. In such a paradigm the neural correlates of action inhibition would be the focus rather than preparation of future action. Lastly, optogenetic approaches to activate or inhibit motor and forelimb cortex of mice after they have acquired the task could also be implemented, especially if mice learn a self-initiated – rather than cued task (the latter being more often the case). Overall, we view these and other similar paradigms as useful complementary studies to our work and suggest them for future studies.

## Longitudinal performance of a self-initiated task restructures the neural dynamics underlying action and spontaneous neural activity

It has been suggested that noise-driven stochastic models (i.e., leaky stochastic accumulators) can explain the RP as a result of averaging backwards in time over multiple stochastically determined behavior initiations (*Schurger et al., 2012*). In contrast, a recent human study aimed at testing part of this hypothesis found that the RP amplitude increased with learning suggesting that the RP represents planning and learning rather than stochastic structures alone (*Travers et al., 2021*). Consistent with the later study, we also found that decoding times for upcoming water-rewarded lever pulls in some mice changed, in particular they became shorter (*Figure 4*). We also found structural changes in the neural dynamics in some mice as both the lever-pull neural space and its overlap with the left paw changed systematically with time. These results suggest that learning, or merely longitudinal performance, of a high-value behavior (e.g., water seeking in water deprived mice) increases stereotypy even in self-initiated actions. We suggest that automaticity-like processes could be involved with movement preparation becoming less dependent on cortex and more dependent on subcortical structures. These large-scale cortical changes in brain activity have been observed in humans, for example, while learning a brain–machine interface task (e.g., *Wander et al., 2013*) but aside from a very recent EEG study (*Travers et al., 2021*) are not well described. The mesoscale mechanisms for such changes may involve increased representation of lever-pull preparation dynamics in the overall spontaneous dynamics (e.g., occurring more frequently) or increase differentiation between lever-pull preparatory neural activity and other behaviors. These results add to the evidence that self-initiated actions are supported by learned neural-dynamical structures and that those can change on longer time scales.

## Inter-animal variability

We found inter-animal differences in several results: different decoding times, anatomical areas involved as well as longitudinal dynamics (e.g., differences in longitudinal decoding trends *Figure 4*; or preferred phase angles at behavior time different *Figure 3*; different trends in frequency peak and power *Figure 7*). These results suggest that neural dynamics and strategies for initiating a self-initiated action could be specific to individual subjects and that pooling over multiple subjects may remove novel or important differences. In other words, single subject analysis may be critical to further advancing the debate on the dynamics of neural activity underlying self-initiated behaviors in both humans and nonhumans.

## Internal state evidence accumulation commences seconds prior to movement

We found that both lever-pull decoding accuracy increases nearly monotonically with approaching action (e.g., *Figure 4b*) and that intra-session variance decreases several seconds prior to movement within limb, motor, and retrosplenial cortex. We suggest these findings constitute further 'indirect evidence that evidence-accumulation' (*Bode et al., 2014*) is occurring even in the absence of explicit stimuli – likely based on evaluations of internal states and models. This supports the hypothesis that internal-state driven self-initiated actions could be potentially modeled by commonly used perceptual

and cognitive decision making models (*Gold and Shadlen, 2007*; *Heekeren et al., 2008*; *Murakami et al., 2014*; *Murakami et al., 2017*).

## The effect of confounding movements on the decoding of future rewarded action

Our study does not directly address the effects of stimuli or other perturbations on the neural dynamics preceding self-initiated decisions, for example, as in some human choice paradigms that consider decision choice in the presence of novel information (e.g., *Maoz et al., 2019*). Additionally, there is evidence that micro-movements can contribute to ongoing neural activity between behavior bouts (*Stringer et al., 2019*; *Musall et al., 2019*). While it is possible that in preparing for a self-initiated rewarded behavior mice undergo a series of physical movements, we did not see any evidence for this in our recordings. In particular, we found a tendency for mice to decrease their body movements in the 3–5 s prior to self-initiated action and we did not find evidence for sequences of paw or licking movements. We suggest future studies focusing on the micro-movements underlying self-initiated action could address this issue using high-frame rate and high-resolution video recordings.

## Limitations

Our study focused on characterizing *neural dynamics* and *timing* of an ethologically valuable movement rather than identifying *intent* or *awareness* of upcoming movements. As such, we do not directly address the role of subjective 'intent' as in some human EEG studies (e.g., *Libet et al., 1983*) or the role of reasons or deliberation on decision making. (Note: as mentioned above, the effect of reporting intent and the use of reasons or deliberation in voluntary actions are the subject of ongoing debates; see, for example, *Dijksterhuis et al., 2006*; *Dijksterhuis and Aarts, 2010*; *Vierkant et al., 2019*; *Wegner, 2002*.) Our study was also not aimed at disambiguating between the timing of intent awareness and movement. Uncued voluntary action studies in humans generally find the difference between the timing of subjective intent and movement to be small (e.g., 150 ms; *Libet et al., 1983*) or even negligible, orders of magnitude smaller than the EDTs in our results. Second, our decoding times showed a strong dependence on the *number of trials* suggesting that additional trials would change (most likely improve) our decoding results (though we note that this effect may merely reflect sequences of lever pulls). Although it is challenging to keep animals motivated across many trials within a single ~20-min session, decreasing reward size might have increased the number of self-initiated lever-pull behaviors and reduced the dependence we observed. Third, we sought to remove pre-movement confounds from our results by 'locking out' previous lever pulls or body movements (*Figure 6*). A more direct approach where animals are specifically trained to remain quiescent prior to an action may yield more trials and easier to interpret results. However, it is practically challenging to train mice to withhold behaviors for significant periods of time (e.g., >>3 s) while also performing a task for a valuable reward. Fourth, we found strong correlations between lever pulls and body movements in all mice (not shown). However, we did not take into account the temporal location of lever-pull activity when decoding the body movement (*Figure 2*); for example, we decoded upcoming left paw movement initiations without accounting for – or removing – lever-pull initiations co-occurring with such paw movements. It is obvious that many of the spontaneous paw movements also coincided with lever pulls and thus the EDTs for paw movements were not an independent measure from the EDTs of rewarded lever pulls. However, we chose to remain agnostic and not separate body movement initiations into those coinciding with lever pulls and those that occurred many seconds away from rewarded lever pulls (this also had the effect of preserving a higher number of body-movement trials for decoding). Despite not separating the data, we did find a significant difference in lever pull and right paw dynamics longitudinally (*Figure 4j*) suggesting that further separation may have only increased this difference. We acknowledge that it would have been interesting to divide the behaviors and carry out separate analysis, and leave this direction for future projects.

## Conclusion

Over the past few decades, rodent models of sensory systems and decision making have become increasingly common (e.g., visual evidence accumulation; *Najafi and Churchland, 2018*; *Odoemene et al., 2018*; *Aguillon-Rodriguez et al., 2021*). The findings presented here suggest that mice could also be an appropriate model for neuroscience investigations into self-initiated action. While

characterizing the dynamics underlying self-initiated behavior in rodents may advance our understanding of self-initiated action in nonhumans as well, it could also advance our understanding of developmental and psychiatric disorders that have behavioral symptoms such as avolition in depression (e.g., lack of will to move; *Brakowski et al., 2017*) and behavior repetition observed in obsessive–compulsive disorders (*Lysaker et al., 2018*). Our findings suggest that the neural mechanisms underlying self-initiated action preparation and performance could be preserved in part, or in whole, between humans and mice and that studies of self-initiated action in humans would benefit from mouse models and the vast libraries of behavior, genetic, and neural recording methodologies available.

## Materials and methods

### Mice

Mouse protocols were approved by the University of British Columbia Animal Care Committee and followed the Canadian Council on Animal Care and Use guidelines (protocols A13-0336 and A14-0266). Six GCaMP6 transgenic male mice (Ai93 and Ai94; *Madisen et al., 2015*) were used. For the study the mice names were defined as M1–M6 and had the following genotypes: M1–M5: Ai94; M6: Ai93.

### Lever-pull task

Mice were kept on a restricted water schedule as previously described (see *Silasi et al., 2016*). Briefly, mice were implanted with a head post and head fixed with their bodies partially resting in a 28-mm diameter Plexiglass tube. A 1-cm cutout from the right side of the tube floor accommodated a monitored lever that was positioned at the same height as the tube. At the start of each session a water spout was set near the mouth of the mice, ensuring the mice could obtain dispensed water drops by licking. In order to receive a water reward mice were required to pull the lever past a threshold and then hold the lever without pulling it to the maximum value. Correct lever pulls – 'rewarded lever pulls' – were tracked in real time to provide water reward. Following a reward a lockout period of 3 s was implemented during which mice could pull the lever but would not be rewarded irrespective of performance. This task required learning the minimum threshold, the duration of the hold and the refractory period of the lockout. We note that the duration of the hold was gradually increased by 0.1-s increments in mice that learned to perform the task well. We selected 3 s as longer lockout values limited the consistent acquisition of the lever-pull task. We recorded widefield calcium activity across each recording session (1330 s, i.e., ~22 min long) over many days (42–109 days, average = 58.3 ± 24.6 standard deviation). Mice had longitudinal trends with some increasing the # of lever pulls over time and others decreasing. (See *Figure 4a*; M1–M6: Pearson correlations: −0.25, 0.39, 0.65, −0.37, 0.18, 0.68; p values: 0.048, 0.020, 1.095e−5, 0.288, 6.320e−15; note: because mice were not habituated we discarded the first week of training in this computation to better capture longitudinal trends rather than habituation idiosyncrasies.)

### Widefield calcium imaging

Widefield calcium imaging was carried out as described previously (*Xiao et al., 2017*; *Silasi et al., 2016*). Briefly, mice with a chronically implanted transcranial window were head fixed under a macroscope. Images were captured at 30 Hz with 8 × 8 pixel binning, producing a resolution of 68 µm/pixel (*Vanni and Murphy, 2014*). To visualize the cortex, the surface of the brain was illuminated with green light (but not during image acquisition). Calcium indicators were excited with blue-light-emitting diodes (Luxeon, 470 nm) with bandpass filters (467–499 nm). Emission fluorescence was filtered using a 510–550 nm bandpass filter or collected in a multi-band mode as described below. For single wavelength green epifluorescence, we collected 12-bit images at varying time resolution (33 ms; i.e., 30 Hz) using XCAP imaging software. In order to reduce file size and minimize the power of excitation light used, we typically bin camera pixels (8 × 8) thus producing a resolution of 68 µm/pixel. Hemodynamic correction was not available as we only used single wavelength excitation. Based on previous experiments using similar imaging conditions (*Vanni and Murphy, 2014*; *Silasi et al., 2016*; *Xiao et al., 2017*) in control GFP expressing mice we would not expect significant contributions from hemodynamic signals under the conditions we employed. These imaging parameters have been used previously for voltage-sensitive dye imaging (*Mohajerani et al., 2013*) as well as anesthetized

GCaMP3 imaging of spontaneous activity in mouse cortex (*Vanni and Murphy, 2014*) and awake GCaMP6 imaging in mouse cortex with chronic window (*Silasi et al., 2016*).

### Behavioral recordings

Behavior was recorded using a Windows OS camera at 15 frames per second. Video recordings were saved in the native .wmv format and converted at the same resolution to .mp4 format for post-processing steps. Each video recording session lasted approximately 22 min and contained ~20,000 frames.

### $\Delta F/F_0$ computation

$\Delta F/F_0$ computation was carried either via bandpass filtering (0.1–6.0 Hz) or as previously described (*Xiao et al., 2017*). Briefly, $F_0$ was computed as the average pixel activity in the window preceding the analysis window. For example, for analyses of neural activity within a ±3 s window following a behavior, $F_0$ was computed based on the previous 3 s of neural activity, that is the −6 to −3 s window. We found no statistical differences in our results between using sliding window $\Delta F/F_0$ or bandpass filtering and for our analysis we relied only on bandpass filtered data.

### Registration to Allen Institute dorsal cortex map

We used a 2D projection of the Allen Institute dorsal cortex atlas, similar to *Musall et al., 2019*, *Saxena et al., 2020*, and *Couto et al., 2021* to agnostically identify ROIs without the need for stimulus driven or other neuroanatomical markers. We rigidly aligned the widefield data to a 2D projection of the Allen Common Coordinate Framework v3 (CCF) (*Oh et al., 2014*) as in *Musall et al., 2019*, *Saxena et al., 2020*, and *Couto et al., 2021*, using four anatomical landmarks: the left, center, and right points where anterior cortex meets the olfactory bulbs and the medial point at the base of retrosplenial cortex. The ROIs identified for each animal and session were individually inspected to qualitatively match expected activation of somatosensory cortex during lever-pull trials in each session.

### Analysis

All analysis was carried out using custom python code developed as part of an electrophysiology and optical physiology toolkit available online (https://github.com/catubc/widefield; *Mitelut, 2022*; copy archived at swh:1:rev:726ecd42f035f17af9cc7e4f274b3b55a3ef6908). Methods for computing event triggered analysis for widefield imaging have been previously published (*Xiao et al., 2017*) and are also available online (https://github.com/catubc/sta_maps; *Mitelut, 2017*).

### Unsupervised behavior annotation and body movement computation

Seven features were identified for tracking: center of left paw, center of right paw, the underside of the jaw, the tip of the nose, the underside of the right ear, the tongue, and the midpoint of the lever. DeepLabCcut (DLC v. 2.1.8; *Mathis et al., 2018*) was used to label these features in 60 frames per video for three videos in each animal. The DLC predictions were inspected and smoothing was applied to correct missing or error frames (using a 30 frame window sliding mean or Savitsky-Golay 31 frame filter using third degree polynomial; note: mouse M2 did not have good tongue tracking and this feature was excluded from analysis). Body movement initiations were computed as the first time point at which the velocity was larger than three times the standard deviation of velocity over all periods. We then excluded movement initiations which were preceded by another initiation (of the same body part) in the previous 3 s of time.

### Principal component analysis

PCA was applied to neural activity time series to decrease the dimensionality and denoise the data. For each session we first converted the filtered [Ca++] neural activity from pre lever neural recordings from time −15 to +15 s into a series of [n_frames, width_pixels * height_pixels]. These data were then run through principle complement analysis linear dimensionality reduction using the python sklearn package to obtain a pca model (available here). We next selected the number of principal components required to reconstruct the data to ≥95% variance explained precision. Lastly, we applied the PCA model (i.e., denoised) to both the lever-pull neural data and control data.

## SVM classification – decoding single sessions

We used SVM classification to decode neural activity preceding an action vs. random periods of time using methods similar to those used in humans with fMRI data (*Soon et al., 2008*). Briefly, for each session and each rewarded lever pull or body movement initiation we extracted segments of neural activity 30 s long centered on the time of the action (i.e., −15 to +15 s following the action). Controls were selected similarly but the time of the action (i.e., $t = 0$ s) was randomized to fall anywhere in the session except a ±3 s window around an action. For clarity, controls could contain neural activity from rewarded lever pulls or body initiations; we found this to be a more conservative method than to manually select only non-movement periods as controls. We next denoised both the behavior data and the control data using PCA (see description above). We then built SVM classifiers using as input 1-s-wide windows (30 frames @ 30 FPS) of data from both the behavior (i.e., class #1) and the random controls (i.e., class #2). The input to each SVM classifier was a 2D array [n_trials, n_frames * n_PCs]. For example, in a session where >95% of the data dimensionality was captured by 10 PCs, the input the the SVM classifier was: [n_trials, 30*10] = [n_trials, 300]. We similarly computed the control array and the SVM classifier were trained on two classes (i.e., lever pull vs. control). We tested additional sized windows (i.e., single frame = 30 ms, or 150 frames = 5 s) but did not see significant improvements. We used sigmoidal SVM kernels as they showed a slight improvement in SVM accuracy over linear kernels (see https://scikit-learn.org/). We carried out 10-fold cross-validation using a split of 0.9:0.1 train:validate. The output of the SVM classification for a 1 s window was assigned to the value of the last time point in the stack (e.g., the accuracy computed from decoding the −15 to −14 s time window was assigned to the −14 s time bin). We carried out this SVM classification for each time point in the −15 to +15 s window. For clarity, 870 SVM classifiers (−15 to +14 s = 29 s * 30 fps = 870 frames) were trained for each validation point (i.e., 10-fold cross-validation).

## SVM classification – decoding concatenated sessions

We additionally trained SVM classifiers on concatenated sessions to increase the number of trials available. Sessions were concatenated across sequential behavior days to reach a minimum of 200 trials. Intra-session PCA was first applied to each session to denoise the data locally using a minimum of 95% reconstruction accuracy as above. The denoised time series were then concatenated and fit to a PCA model (available here) using randomly sampled (3%) of the data from the concatenated stack. The multi-session PCA matrix was then used to denoise the individual sessions and we kept a fixed 20 principal components to reconstruct the concatenated datasets. The remaining steps (SVM training and decoding) were carried out as for the single session approach described above.

## SVM classification – decoding locked out concatenated sessions

We trained SVM classifiers on trials that were locked out (i.e., not preceded by a previous rewarded or unrewarded lever pull) by several seconds (see *Figure 6*). Sessions were generated by pooling a minimum of 50 to a maximum of 200 locked-out trials. We chose these values to be qualitatively similar to the analysis done for non-lockout trials in which we used single session trials (ranging from 30 to >100 trials per session) and concatenated trials (which contained trials from several sequential sessions reaching a minimum of 200 trials). We used a 50 trial-sliding window to select trials for each concatenated session. For example, for mouse M1, for a lockout of 15 s, a total of 503 trials survived out of all rewarded lever-pull trials. We pooled these 503 trials in groups of 200 using 50 trial windows resulting in 10 groups of (overlapping) data to be processed (i.e., 0–200, 50–250 … 450–503). This method is essentially similar to the concatenated datasets where for each session we added trials from subsequent sessions until we reached at least 200 trials (i.e., that method also yielded overlapping trials across time). While this method does yield overlapping trial sessions, our goal was to show EDTs for such synthetic sessions. Using non-overlapping data (i.e., the sliding window was set to 200 trials) yields similar results.

## Computation of EDT

We sought to use a method that detected the first time point in the cross-validated SVM accuracy curves that was above chance at a statistically significant level (i.e., Student $t$-test p value <0.05). We denoted this time as the EDT for each session. We obtained the EDT for each session using several steps. First, for each session, we computed the 10-fold cross-validated accuracy curves using 30-time

step (i.e., 1 s) windows as described above but we additionally filtered the accuracy curves using a 30-time step moving average to further decrease the effects of noise on the prediction curves. Next, we obtained the significance at each time point by computing a one-sample $t$-test between the SVM cross-validation accuracy values (i.e., 10 values) and a population mean of 0.5 (i.e., chance) using the python scipy stats package. We next applied a Benjamini–Hochberg correction for multiple tests using the python statsmodel package. Finally, starting at $t = 0$ s, we moved backwards in time until we found the last time point that was statistically significant (i.e., p value <0.05 as computed above). This last step had the effect of imposing a constraint which required all decoding accuracy distribution times following the EDT to be statistically significant – thus excluding random stochastic fluctuations in the accuracy curves which could result in very low EDTs that are not reasonable or meaningful. The effects of this last constraint could be observed in *Figure 2g, h* where the EDT is higher (i.e., closer to $t = 0$ s) than other time points that are statistically significant (see top colored bars for statistical significance and note that there are some isolated times that show statistical significance). We denote 'shortened' EDT time to indicate that the EDT time to the lever-pull time (i.e., $t = 0$ s) decreased (i.e., decoding become poorer) and 'lengthened' EDT time to indicate that the EDT time to lever-pull time increased (i.e., decoding was better).

## Statistical tests for single ROI vs. all neural areas

Two-sample KS tests were carried out between EDTs obtained by using limb cortex neural activity vs. EDTs obtained using all areas. This analysis was done on non-lever lockout data (i.e., all rewarded lever trials) as it is a cross-feature analysis not a cross-session analysis and we were interested in the relative effects between areas not absolute EDT times. This approach enabled us to obtain a significantly higher number of sessions (as opposed to using locked out data only).

## LocaNMF

LocaNMF was applied to the data from each widefield session as in *Saxena et al., 2020*. Briefly, we applied semi non-negative matrix factorization (sNMF) to the denoised data, while encouraging localization in the spatial components to the Atlas regions. This results in spatial components that are aligned to the different Atlas regions, thus allowing us to further analyze the corresponding temporal activity in each region. The locaNMF parameters used were as follows: maxrank = 1; min_pixels = 200; loc_thresh = 75; r2_thresh = 0.96; nonnegative_temporal = False; maxiter_hals = 20; maxiter_lambda = 150; lambda_step = 2.25; lambda_init = 1e−1.

## Power spectra

LlocaNMF temporal component spectra were computed using the python scipy signal package.

## Low band-pass filtering of time series

We used both causal and non-causal filters to evaluate the effects of filtering on EDTs. For non-causal filtering we used the scipy.signal.filtfilt(), and for causal filtering we used scipy.signal.lfilter() and corrected by shifting the filtered time series by an amount of fps x 1/filter_frequency = 30 fps × 1/0.3 Hz = 100 time steps. Both filtered time series were qualitatively similar in shape, amplitude, and phase.

## Convex-hull analysis

For each session, the neural activity 'convex hull' at lever-pull time (i.e., $t = 0$ s) was defined as the hyper-volume that enclosed the $t = 0$ s neural activity vectors. As convex hull analysis is sensitive to outliers a 10% $K$-nearest-neighbor triage was implemented prior to evaluation. The convex hull of lever-pull (i.e., $t = 0$ s) neural activity can be visualized in two dimensions by carrying out PCA on all the neural activity from the session and then computing the convex contour enclosing the $t = 0$ s neural activity vectors (see *Figure 4d* – blue dots). For computing the ratios of the pre-pull convex hull to the random neural data we chose random periods of time uniformly from the session. The normalized area under the ratio was computed by first computing the convex hull of neural dynamics from 10 to 0 s prior to a pull, normalizing at every computation by the total area of the convex hull for all neural activity occurring in the session. We then computed the area under this curve and divided it by a

similar computed curve by this time starting at randomized times (i.e random times at least 3 s away from a rewarded lever pull).

## Pre-movement region-based ROI phase computation

We computed the phase of a neural activity of single trial for each region's temporal components ROI by fitting a sine function to the period of −5 to 0 s prior to lever pull. We next computed the phase of each trial as the intersection of the sine fit with the $t = 0$ s line.

## Sinusoidal fits to single-trial data

Sinusoids were fit using scipy curve fit function to single-trial neural data from each area based on the last 5 s preceding the lever pull (i.e., −5 to 0 s).

## Earliest variance decrease time

We defined the EVDT for each session as the time at which the variance in a 1-s sliding window decreased by 2× the standard deviation of the variance computed in the window −30 to −15 s prior to the lever pull. We found the requirement for all variance values in a 1-s sliding window to fall below the threshold as necessary to deal with noise or fluctuations in variance. We also required the EVDT to fall between −6 and 0.5 s prior to the pull or it was discarded from analysis resulting in many sessions being discarded from analysis, especially for mice M1 and M2. More robust methods for detecting the variance decrease are likely possible, but were not explored. We also note that in Mouse M2, the variance change prior to movement was positive (i.e., variance increased) and thus we used the absolute of the difference to compute the EVDT (instead of only considering decreases). We note that as the time courses could be quite noisy, this method is sensitive to thresholds being set, for example 2 vs. 4× std can yield somewhat different distributions. We also note that we used a Savitsky-Golay filter (31 samples) to furter smooth the data in order to get consistent results from visual cortex data.

## Detecting stereotyped movements

We sought to detect the presence of stereotyped movement patterns (e.g., left paw followed by right paw followed by licking) using PCA applied to binarized behavior time series (not shown in results). We first obtained DeepLabCut time series locations for the left paw, right paw, and tongue as a 2D vector [n_time_points, n_dimensions], where n_time_points: number of imaging frames in a session (usually 1300 s × 15 fps) and n_dimesions: x and y video coordinates (in pixels). We next generated a Boolean array with zeros representing no movement and ones representing body movements. We then extracted segments of 20 s long around each lever pull (i.e., from −10 to +0 s relative to lever pull) resulting in arrays of dimension [10 × 15 fps, 2] = [150, 2]. We then flattened and stacked the arrays (trial wise) and computed PCA on the resulting data (i.e., input into pca [n_trials, n_time_points x dimension]). In an additional step, we also concatenated across body parts resulting in single-trial dimensions of [n_time_steps, n_dimensions x n_body_parts]. Neither approach (single body part of concatenated body parts) yielded multiple discernable clusters in the first two PCs.

## Quiescence period analysis

Analysis in *Figure 1*, *Figure 1—figure supplements 6 and 7* and in *Figure 8* required the computation of bouts of activity around a lever pull or limb movement. Movements (of limb or lever) were detected as explained above and binned in segments of 0.250 s. Given the lower temporal resolution of our videos, when computing lockouts we allowed for the first or last bin of a lockout period to contain movements (e.g., the first bin in a 15-s lockout period or the very last bin prior to movement, i.e., the −0.250 s bin).

## Processing and analysis of data for Figure 1, Figure 1—figure supplement 7

Pooling data across weeks or months of longitudinal recordings adds substantial noise and other confounds to our datasets (see *Figure 2—figure supplement 3*, and Main manuscript for *Figure 6* for a detailed explanation). For the analysis here we pooled lever lockout trials (as in *Figure 6*) but were further limited to only those sessions where video was available. This reduced our overall trials by approximately 50% (or more) in all mice compared to *Figure 6* analysis. The trials were further

split into three groups representing least to most body movements. Additionally, trials pooled into these subgroups were ordered by movement amount – as opposed to ordered by day of acquisition time. This also has the effect of increasing the variance of data means due to time between individual trials in the subgroup. Given these issues, our analysis in this panel was limited to characterizing [Ca] time courses only (i.e., it was not possible to also carry out EDT decoding on trials contained in these reduced datasets – as the time series were too noisy).

## Acknowledgements

This work was supported by Canadian Institutes of Health Research (CIHR) Operating Grant MOP-15360 and National Science and Engineering Research Council of Canada 178702; Canadian Institutes of Health Research (CIHR) Operating Grant MOP-12675 and Foundation Grant FDN-143209 to THM. We thank Pumin Wang, Cindy Jiang for surgical assistance; Jeff LeDue for technical assistance. We also thank Kenny Kay, Xuexin Wei, and Allen Chan for comments on the initial manuscript.

## Additional information

### Funding

| Funder | Grant reference number | Author |
|--------|------------------------|--------|
| Canadian Institutes of Health Research | MOP-15360 | Catalin Mitelut<br>Yongxu Zhang<br>Yuki Sekino<br>Jamie D Boyd<br>Federico Bollanos<br>Nicholas V Swindale<br>Greg Silasi<br>Timothy H Murphy |
| Canadian Institutes of Health Research | MOP-12675 | Catalin Mitelut |
| Canadian Institutes of Health Research | FDN-143209 | Timothy H Murphy |
| National Science and Engineering Research Council of Canada | 178702 | Catalin Mitelut<br>Yongxu Zhang<br>Yuki Sekino<br>Jamie D Boyd<br>Federico Bollanos<br>Nicholas V Swindale<br>Greg Silasi<br>Timothy H Murphy |

The funders had no role in study design, data collection, and interpretation, or the decision to submit the work for publication.

### Author contributions

Catalin Mitelut, Conceptualization, Resources, Data curation, Software, Formal analysis, Supervision, Validation, Investigation, Visualization, Methodology, Writing - original draft, Project administration, Writing – review and editing; Yongxu Zhang, Software, Formal analysis; Yuki Sekino, Conceptualization, Investigation, Methodology; Jamie D Boyd, Conceptualization, Data curation, Software, Methodology; Federico Bollanos, Conceptualization, Data curation, Investigation; Nicholas V Swindale, Funding acquisition, Writing – review and editing; Greg Silasi, Conceptualization, Data curation, Supervision, Investigation, Methodology, Writing – review and editing; Shreya Saxena, Software, Formal analysis, Supervision, Writing – review and editing; Timothy H Murphy, Conceptualization, Resources, Data curation, Supervision, Funding acquisition, Methodology, Project administration, Writing – review and editing

### Author ORCIDs

Catalin Mitelut ![ORCID] http://orcid.org/0000-0003-0471-9816
Yuki Sekino ![ORCID] http://orcid.org/0000-0003-2038-274X

Nicholas V Swindale (ID) http://orcid.org/0000-0002-7106-5114
Timothy H Murphy (ID) http://orcid.org/0000-0002-0093-4490

### Ethics

Mouse protocols were approved by the University of British Columbia Animal Care Committee and followed the Canadian Council on Animal Care and Use guidelines (protocols A13-0336 and A14-0266).

### Decision letter and Author response

Decision letter https://doi.org/10.7554/eLife.76506.sa1
Author response https://doi.org/10.7554/eLife.76506.sa2

## Additional files

### Supplementary files

• Transparent reporting form

### Data availability

Code for generating all figures is provided here: https://github.com/catubc/elife_self_init_paper, (copy archived at swh:1:rev:2a6d97d1afdc611e827d952dfa3c7d2fecb4ec33). Datasets are provided on Dryad under the information below: Mitelut, Catalin (2022), Mesoscale cortex-wide neural dynamics predict self-initiated actions in mice several seconds prior to movement, Dryad, Dataset, https://doi.org/10.5061/dryad.ttdz08m0z.

The following dataset was generated:

| Author(s) | Year | Dataset title | Dataset URL | Database and Identifier |
|---|---|---|---|---|
| Mitelut C | 2022 | Mesoscale cortex-wide neural dynamics predict self-initiated actions in mice several seconds prior to movement | https://dx.doi.org/10.5061/dryad.ttdz08m0z | Dryad Digital Repository, 10.5061/dryad.ttdz08m0z |

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
