## [Editor Report]

This study is a valuable work that advances our knowledge of the neural correlates of voluntary action through a wide range of methods. The evidence supporting their conclusion is convincing, and the results will be of interest to a large class of neuroscientists interested in the neural mechanisms underlying self-initiated actions.

---

## [Decision Letter]

**Decision letter after peer review:**

Thank you for submitting your article "Mesoscale cortex-wide neural dynamics predict self-initiated actions in mice several seconds prior to movement" for consideration by *eLife*. Your article has been reviewed by 3 peer reviewers, and the evaluation has been overseen by a Reviewing Editor and Christian Büchel as the Senior Editor. The reviewers have opted to remain anonymous.

Essential revisions:

1) There was confusion amongst the reviewers as to the precise definitions of what the authors called "voluntary" movements. Providing a more precise definition (and the appropriate controls, as described below) will be important.

2) One of the goals of the authors was to study the neural mechanisms underlying voluntary movements. While they acknowledge (in the discussion) that they do not have evidence that actions are "intentional", they make the assumption that mice do "form the intent to act near the lever pull time". To back up this assumption, the authors should at least present some evidence that the action of interest (i.e., the rewarded lever-pull) is not just a random jerky movement that happens to be rewarded once in a while. In fact, mice seemed to pull the lever very frequently and impulsively (the majority of inter-pull intervals were way below 3 s in Supplementary Figure 1.2) even for the last sessions of the training. Therefore, it is not readily apparent that mice apply any control to their lever-pull actions. Providing evidence that the action is goal-directed is important if the goal of the paper is to study neural signatures of the intention to act. A somewhat compelling analysis could be to compare rewarded lever-pulls with "spontaneous" movements, provided that these two types of movement can be convincingly characterized as goal-directed vs. incidental. In contrast, throughout the manuscript, the neural activity aligned to rewarded lever-pull events (which are assumed to be "voluntary" actions) is compared to the neural activity aligned to random times during the task (whether or not it involved movements), which may not be the most convincing control.

3) The learning trajectory of mice is also not well characterized (e.g. changes in inter-pull intervals are not quantified, nor the relative increase in rewarded actions across training sessions, etc.). Yet, several claims in the paper are directly based on the fact that mice have learned to pull the lever after 3 s interval to receive water rewards (which relates to point 1). In particular, one important assumption in the paper is that as mice learn, the lever-pull movements become more stereotyped, but this has not been shown explicitly. It would be helpful, for example, to see how analog traces of lever-pulling change throughout the learning stages and how the variance of the movement across trials decreases in late sessions.

4) The central claim of the paper is that rewarded lever-pulls can be predicted from pre-movement neural activity several seconds (even up to 10 s) prior to the action. However, obvious motor confounds and other alternative explanations have not been convincingly ruled out. In fact, the action of lever pulling may require a series of complex movements (like changing posture, extending the forelimb, reaching the lever, grabbing the lever, etc.). The authors themselves mentioned that they found strong correlations between lever pulls and body movements in all mice, but the data is not used nor shown in the paper. The motor commands preceding but related to lever-pull could unfold at least a few hundreds of milliseconds prior to the detection of lever-pull in the task, and thus be reflected in the neural activity that is predictive of the lever pull. Moreover, if this series of movements is highly stereotyped, and in turn leads to stereotyped neural activity (like the slow oscillations observed before the lever-pulls), it could explain why the detection of lever-pulling actions always occurs at a given phase of the neural oscillation. Such observations that stereotyped movements occur way before the lever-pull detection could partially rule out the fully "cognitive" explanation proposed in the paper, but would concur with recent findings that showed that ramping neural activity can be, for the most part, explained by movement-related activity (Musall et al., 2019).

5) Toward the end of the result section (Figure 6), the authors briefly begin to address the issue about whether pre-movement activity can really be considered movement free. Here, "lockouts", i.e. periods where other movements (like licking, or previous lever-pulls) did not occur, were introduced in the analysis. The lockouts altered the earliest-decoding-time (EDT) of the lever-pull (in some mice EDT was even divided by half: from -4 s to -2 s). However, the effects of "micro-movements" like facial movements or changes in body posture may not be taken into account with the lockout approach. Such micro-movements have been shown to explain a large variance of the neural activity (see Stringer et al. 2019 and Musall et al. 2019). Therefore, to fully control for movement confounds, the effect of high dimensional/micro-movements extracted from video recordings should be removed from the neural activity. These analyses could yield a much shorter EDT (e.g., -0.15 s), more consistent with previous reports.

6) SVM decoding accuracy in Figures 1h, 2b, 2d, etc. shows in some cases that the earliest decoding time can precede a lever pull by 10-15 seconds. But how are we to know that SVM predictions in a given 1-second window are predicting an upcoming movement 10-15 seconds in the future, versus a separate pulling event that occurs within the SVM window on some subset of trials in a session? The increase in SVM accuracy as the movement approaches may be (as in A) due to changes in probability and temporal coupling in pulling events. The data are aligned to lever pull n, hence maximizing SVM accuracy at that time point. It is unclear whether the authors can address such concerns with the existing data. One analysis that would be telling would be something akin to GLM. For example, in Figure 1e, are the oscillations due to multiple movements occurring in succession? The authors could fit a Gaussian to the last peak (from about -2 sec to 2 sec) in Figure 1e. Then the authors can take this Gaussian and convolve it with each lever pull event, and then add all these together into a composite trace that represents what neural activity should look like if it is only determined by lever pull events that follow some Gaussian distribution. Figure 1e could be reconstructed from this idealized timeseries as if it were the actual raw data. Should this control also show oscillations with increasing amplitude, this suggests that the neural activity in Figures 1e (and the SVM analyses by extension) is due to multiple lever pulls – not volitional activity that predicts an upcoming lever pull. Similarly, as a control, the authors should also re-plot neural activity in Figures 1e, 5a, etc., using the longer post-hoc lockouts employed in Figure 6. That is, the authors should re-plot activity but exclude all trials where a previous lever pull occurred within x seconds, where x is 6, 9, 12, etc. as in Figure 6g.

7) Another related concern seems to be the convex hull analysis in Figures 4d-g. For example, in Figure 4e it is shown that the convex hull decreases in size as the movement (t=0 sec) approaches. The authors conclude that neural activity looks more random prior to the lever pull. However, given the concerns above, it seems this "randomness" could actually be due to other lever pulls (and other spontaneous behaviors) that are not exactly temporally aligned, and not certain to occur in each sample. Thus, we would also ask the authors to repeat this analysis using longer post-hoc lockouts to prevent contamination from earlier lever pulls.

8) It is excellent that the authors considered the issue described in Point (1), in their control analysis in Figure 6. But the reviewers disagree with their conclusions. Figures 6g and 6h clearly show that EDT drops considerably as the lever lockout window increases. For example, the 9, 12, and 15 second lever lockout periods appear to mostly eliminate all sessions with EDTs greater than 5 seconds. This is a drastic reduction in the time horizon (10-15 seconds) argued by the authors given their earlier results.

The authors might argue that these results are mixed, given that they were not statistically significant in M1 and M2 (Figure 6g). But this is misleading. In M1, there are only 2 or so EDTs greater than 5 seconds in the 3 second lockout condition. There are none in the 6 second lockout condition. Thus, in this mouse, the lack of statistical significance is simply because there are little to no EDTs to eliminate by gradually increasing the lever lockout period. This analysis is not useful in a mouse that exhibited little to no EDTs in the 10-15 second range argued by the authors. For M2, it is surprising that no statistically significant effect was detected given that all EDTs greater than 5 seconds are eliminated in the 12 and 15 second lockout conditions. There is a clear trend here despite the statistical results, which calls into question the statistical technique used to assess the relationships in Figure 6g. Could the authors report their statistical technique? As far as we can tell this is not described in the paper. Furthermore, is there a way the authors could analyze outlying mice behavior (M1 and M2) to understand why their results differ? M1 in particular may be very telling, because they have very few "large" EDTs to begin with as shown in Figure 6g? Can this be attributed to some behavior that M1 did (or did not do) as compared with other mice?

9) Overall, it is surprising that the observed clear and large reductions in EDTs in Figures 6g and 6h are dismissed or downplayed by the authors at several points. For example, the authors state "Taking all these factors into account, the results suggest that while EDTs increased slightly in some animals, the cause of the increase could not be disambiguated between [Ca++] state changes, animal behavior, systematic neural network restructuring due to longitudinal performance…". The title of this section reads "Lever pull EDTs are similar or slightly higher with increasing lever-pull lockout duration". The Discussion again states that "Self-caused movement confounds have only minor effects on the decoding of future rewarded action." First, the increase in EDTs is substantial (for example, in Figure 6h, the mean EDT across mice seems to increase from about -4.5 to about -2.5 sec, almost a 50% change) and calls into question the primary results. And second, while we can appreciate that many factors can worsen EDT detection as the lockout period increases (e.g., fewer sessions that occurred further apart), it is not sufficient to explain away the clear and concerning trends by listing several limitations. Especially when it seems that there may be ways to test this. For example, the authors could downsample lever pull events in the 3 sec and 6 sec lockout groups, etc., in order to match the sample size across the 5 conditions, and then repeat their analysis. Additionally, if their concern is that sessions are further apart in the longer lockout conditions, they could resample trials in the smaller lockout conditions in such a way to match the "timing" of samples across the 5 conditions, and then repeat their analysis.

*Reviewer #1 (Recommendations for the authors):*

Recommendations for the author

1. Our main concern with the authors' study is the lack of quiescence prior to lever pulling events. The inter-lever-pull distributions in Supplementary Figure 1.2 show that many pulls will occur within the 15-20 second window prior to a given lever pulling event. This is problematic. Suppose we analyze pull n. How are we to know that neural activity prior to lever pull n relates to that pull, when lever pulls n-3, n-2, n-1, etc., occur within the analysis window?

It seems very possible that several key results may be altered by this confound, as outlined below:

A. The elongated and oscillatory neural activity profiles in Figure 1e, Figure 3a, Figure 5a, and Figure 7a seem to us to reflect a sequence of lever pulling movements. For example, in Figure 1e, there are oscillations in Ca++ activity, with 4-5 peaks. The authors' attribute the change in peak amplitude to an increase in stereotypy as the movement approaches. An alternate hypothesis here is that the change in amplitude is related to the probability and alignment of successive pulling events. For example, there are two large peaks because two pulling events are likely to occur together. The smaller earlier peaks reflect that a third, fourth, and fifth pulling event can occur, but with a smaller likelihood and a less temporally precise coupling to pull n.

B. SVM decoding accuracy in Figures 1h, 2b, 2d, etc. shows in some cases that the earliest decoding time can precede a lever pull by 10-15 seconds. But again, how are we to know that SVM predictions in a given 1-second window are predicting an upcoming movement 10-15 seconds in the future, versus a separate pulling event that occurs within the SVM window on some subset of trials in a session? The increase in SVM accuracy as the movement approaches may be (as in A) due to changes in probability and temporal coupling in pulling events. The data are aligned to lever pull n, hence maximizing SVM accuracy at that time point.

It is unclear whether the authors can address such concerns with the existing data. One analysis that would be telling would be something akin to GLM. For example, in Figure 1e, are the oscillations due to multiple movements occurring in succession? The authors could fit a Gaussian to the last peak (from about -2 sec to 2 sec) in Figure 1e. Then the authors can take this Gaussian and convolve it with each lever pull event, and then add all these together into a composite trace that represents what neural activity should look like if it is only determined by lever pull events that follow some Gaussian distribution. Figure 1e could be reconstructed from this idealized timeseries as if it were the actual raw data. Should this control also show oscillations with increasing amplitude, this suggests that the neural activity in Figures 1e (and the SVM analyses by extension) is due to multiple lever pulls – not volitional activity that predicts an upcoming lever pull.

Similarly, as a control, the authors should also re-plot neural activity in Figures 1e, 5a, etc., using the longer post-hoc lockouts employed in Figure 6. That is, the authors should re-plot activity but exclude all trials where a previous lever pull occurred within x seconds, where x is 6, 9, 12, etc. as in Figure 6g.

Finally, another related concern seems to be the convex hull analysis in Figures 4d-g. For example, in Figure 4e it is shown that the convex hull decreases in size as the movement (t=0 sec) approaches. The authors conclude that neural activity looks more random prior to the lever pull. However, given the concerns above, it seems this "randomness" could actually be due to other lever pulls (and other spontaneous behaviors) that are not exactly temporally aligned, and not certain to occur in each sample. Thus, we would also ask the authors to repeat this analysis using longer post-hoc lockouts to prevent contamination from earlier lever pulls.

2. It is excellent that the authors considered the issue described in Point (1), in their control analysis in Figure 6. But we disagree with their conclusions. Figures 6g and 6h clearly show that EDT drops considerably as the lever lockout window increases. For example, the 9, 12, and 15 second lever lockout periods appear to mostly eliminate all sessions with EDTs greater than 5 seconds. This is a drastic reduction in the time horizon (10-15 seconds) argued by the authors given their earlier results.

The authors might argue that these results are mixed, given that they were not statistically significant in M1 and M2 (Figure 6g). But this is misleading. In M1, there are only 2 or so EDTs greater than 5 seconds in the 3 second lockout condition. There are none in the 6 second lockout condition. Thus, in this mouse, the lack of statistical significance is simply because there are little to no EDTs to eliminate by gradually increasing the lever lockout period. This analysis is not useful in a mouse that exhibited little to no EDTs in the 10-15 second range argued by the authors. For M2, it is surprising that no statistically significant effect was detected given that all EDTs greater than 5 seconds are eliminated in the 12 and 15 second lockout conditions. There is a clear trend here despite the statistical results, which calls into question the statistical technique used to assess the relationships in Figure 6g. Could the authors report their statistical technique? As far as we can tell this is not described in the paper. Furthermore, is there a way the authors could analyze outlying mice behavior (M1 and M2) to understand why their results differ? M1 in particular may be very telling, because they have very few "large" EDTs to begin with as shown in Figure 6g? Can this be attributed to some behavior that M1 did (or did not do) as compared with other mice?

Overall, it is surprising to us that the observed clear and large reductions in EDTs in Figures 6g and 6h are dismissed or downplayed by the authors at several points. For example, the authors state "Taking all these factors into account, the results suggest that while EDTs increased slightly in some animals, the cause of the increase could not be disambiguated between [Ca++] state changes, animal behavior, systematic neural network restructuring due to longitudinal performance…". The title of this section reads "Lever pull EDTs are similar or slightly higher with increasing lever-pull lockout duration". The Discussion again states that "Self-caused movement confounds have only minor effects on the decoding of future rewarded action."

First, the increase in EDTs is substantial (for example, in Figure 6h, the mean EDT across mice seems to increase from about -4.5 to about -2.5 sec, almost a 50% change) and calls into question the primary results. And second, while we can appreciate that many factors can worsen EDT detection as the lockout period increases (e.g., fewer sessions that occurred further apart), it is not sufficient to explain away the clear and concerning trends by listing several limitations. Especially when it seems that there may be ways to test this. For example, the authors could downsample lever pull events in the 3 sec and 6 sec lockout groups, etc., in order to match the sample size across the 5 conditions, and then repeat their analysis. Additionally, if their concern is that sessions are further apart in the longer lockout conditions, they could resample trials in the smaller lockout conditions in such a way to match the "timing" of samples across the 5 conditions, and then repeat their analysis.

3. There is another point to check here, related to these matters. If we understand correctly, the authors used a post-hoc lockout of 3 seconds in their primary analyses (e.g., Figures 2b and 2d). Other lockout periods are tested in the controls in Figure 6. We are curious what constitutes a lever pull in these post-hoc lockout periods? In other words, there is a threshold beyond which the lever needs to be pulled to be rewarded (e.g., the 15{degree sign} angle illustrated in Figure 2a). When the authors exclude lever pulls that were preceded by another lever pull in the lockout window, was it required that the preceding lever pull also exceed the 15{degree sign} threshold? Or was another threshold used? On pg. 27 it says that rewarded and non-rewarded lever pulls were counted as lever pulls in the lockout period. What is meant by unrewarded here: unrewarded because the lever pulls did not meet the minimum threshold, the minimum hold duration for the pull, not hitting the maximum value (as described in Methods)?

The concern we are getting at is that if the minimum 15{degree sign} threshold was used to determine lever pulls in the lockout window, the authors may be not counting smaller (and/or less purposeful) unrewarded lever pulls in the lockout window. Should these pulls also modulate cortical activity, this would pose a problem with regards to Points 1 and 2 above. That is, it seems to us that mice should be as quiescent as possible (excepting perhaps licking / grooming) during the lockout window, to be certain that EDT estimates are not inflated by smaller lever pulls (actions) that may be missed in the lockout period.

4. A response to these concerns would greatly benefit from deeper behavioral analyses. The authors should consider whether pre-pull behaviors are associated with EDTs. We have several suggestions on this point:

Might the authors be able to use a classifier on their videos, to see whether pre-pull behaviors (e.g. subthreshold pulls, paw movements unassociated with the task, licking, grooming, random body motion, rewarded pulls, etc.) can predict future behavior? In other words, the authors could analyze whether classifier output is related to EDT. For example, when an EDT is -5 sec in a given session, are there animal behaviors occurring at -5 sec which might be predictive of the upcoming pull?

The authors have a heterogeneous mouse population. Mouse M1 seems to have a smaller time horizon in their EDTs (as noted in Point 2 above) in Figure 6g. The authors should compare this mouse's behavior to other mice. For example, in the pre-pull period, does Mouse 1 engage in more/less behaviors that might explain why their EDTs differ from other mice.

Are past behaviors predictive of future lever pulls? That is, do mice perform movement sequences of lever pulls (both rewarded/unrewarded)? Do the periodicity of movement sequences relate to EDTs? When sequences are more common in a session, does this appear related to EDT? Do actions such as licking, grooming, body motion, etc. predict upcoming lever presses? If so, does the relative timing between actions relate to EDTs on a given session?

Providing video data in example sessions across the EDT spectrum (e.g. some sessions where EDT was -10 sec, and others where it was -5 sec), could be helpful to the reader in interpreting changes in animal behavior/state that might occur near the EDT.

*Reviewer #3 (Recommendations for the authors):*

• The authors need to make a stronger case regarding the novelty of their study as compared to Murakami et al., 2014, 2017 and da Silva et al., 2018.

• By definition, voluntary action should be reason-responsive. Do the authors have any data to show lever-pull actions are goal-directed?

• Panels a,b,f in Figure 1 do not seem to contain any new information and could simply be mentioned in the text.

• Please report the exact p-values.

• I am not an SVM expert but is it surprising to see decoding accuracy increasing with number of trials? What is the significance of this finding? This also refers to decoding data from multiple cortical regions rather than a specific region. Isn't it expected to see better decoding accuracy with more data?

• There are multiple interesting findings throughout the paper, but the authors failed to explain their significance. For example: (A) they found a higher bias of phases: i.e., behaviours were more likely to occur during a specific phase in mice than in humans. (B) There was a high diversity of phase preference between animals. (C) EDT improved longitudinally. (D) Limb and motor cortex oscillations had most power during pre-movement neural activity [but each animal showed a different pattern]. These are interesting findings but what are their significance and how do they contribute to addressing the research question?

• The statistics are reported in quite an unusual way. Sometimes it is just described qualitatively without any inference test. Sometimes it is reported separately for each animal and sometime across the group. It is quite hard to make any conclusion about the findings when animals show different and in some cases opposite behaviour. Maybe the authors could perform mixed-effect modelling with by-subject random slope and intercept to investigate whether any of the findings is consistent across the animals and significant at the population level.

• The authors conclude that "Overall, these results link human voluntary action studies with rodent self-initiated behavior studies and suggest mice as an adequate model for studying the neural correlates of self-initiated action". This conclusion is not justified without a control condition contrasting self-initiated with externally-triggered action.

[Editors' note: further revisions were suggested prior to acceptance, as described below.]

Thank you for resubmitting your work entitled "Mesoscale cortex-wide neural dynamics predict self-initiated actions in mice several seconds prior to movement" for further consideration by *eLife*. Your revised article has been evaluated by Christian Büchel (Senior Editor) and a Reviewing Editor.

The manuscript has been improved but there are some remaining issues that need to be addressed, as outlined below. In particular, the reviewers would like the authors to respond to the following comments, providing edits to the manuscript as needed (please see the full reviews for further elaboration).

1) It seems the most likely interpretation is that EDTs relate to complete, or at least partial, pauses in behaviors (see Reviewer #1's recommendations for details).

2) In Figure 3, where the authors discuss their finding that movement initiation tends to coincide with peaks in a slow neural oscillation, this interpretation seems potentially problematic, as this oscillation could more simply be a decrease in neural activity during the pause in motor behaviors, and then the subsequent increase during the active motor period (so basically, an ON-OFF-ON transition in neural activity that is correlated to distinct motor events rather than some latent oscillation). An analysis that compares the duration of "quiescent periods" (nicely highlighted in Figure 1-Supp. 5c,d) to EDT is necessary to understand how to interpret these results (see Reviewer #1's recommendations for details).

3) The revised results are not as strong as in the original manuscript. More rigorous analysis of the EDTs revealed that the presence of sequential lever pulls had artificially lengthened the EDT, as anticipated by the reviewers. Additionally, the authors now acknowledge that some of the results are not as reliable as previously reported (i.e., large variability across animals in neural activity phases and longitudinal changes in neural dynamics), weakening the main conclusions of the paper.

4) The reviewers would like the authors to comment on the claim that they have identified a specific neural signature of self-initiated voluntary action. The authors show that pre-movement neural activity in mice contains structures that are not present in random neural activity. This observation is well supported by the data. However, to claim that this structure – neural dynamics becoming increasingly stereotyped prior to movement – is specific to self-initiated actions one needs to show that pre-self-initiated movement (uncued) neural activity in mice contains structures that are not present in externally-triggered movement (cued) neural activity. This comparison could also rule out other possible explanations such as motor confounds associated with the lever pulls or other related micro-movements. Comparison with random time during the task is not sufficient to make any conclusion regarding the self-initiated nature of the behavior.

*Reviewer #1 (Recommendations for the authors):*

I commend the authors on the care they have taken to address reviewer concerns. The added attention to 15-second lockout data is very useful. I think the focus on the 15-second lockout periods largely addresses the issue that sequences (or even random periods of movement) bias EDT estimates. Figure b (rebuttal Page 3) aligns with the updated EDTs in Figure g on Page 4. That is, neural activity changes around 3-4 seconds prior to the lever pull event in Figure b (Page 3), and EDTs seem largely capped at 3-4 seconds in Figure g (Page 4). In addition, there is evidence that mice are sensitive to the 3-second experimental lockout, given the drop in movement probability as shown in Figure c (Page 6). Thus, there does appear to be a 'cognitive decision' associated with obtaining the reward.

But the clear analyses presented by the authors, highlight a new issue that has potential implications for the way they interpret the data. An interpretation that seems consistent with the analyses detailed on Pages 2-6, is that changes in neural activity related to EDTs are less so about pulling the lever, and more so about suppressing actions. For example, in Figure 2-Supplement 2, we see a gradual decline in activity in many brain regions. It appears that the onset of this decline aligns with EDT estimates via SVM. This decline appears to align with the movement data shown in Figure 1-Supp. 5c and 5d, which shows that overall movements start to decline about 3-5 seconds prior to the lever press event. And again, EDTs in the 15-second lockout datasets, appear to saturate around 3-5 seconds. And neural variability decreases in that window (Figure 5). Thus, it seems to be that this decline in spontaneous behaviors is related to the decline in neural activity, and the EDT estimates obtained by SVM. On the other hand, neural activity exhibits a sharp spike 500 ms prior to lever press (Figure 2-Supplement 2) which peaks at the same time as a sudden burst of all sorts of movements (not just lever presses) as seen in Figure 1-Supplement 5.

It seems the most likely interpretation is that EDTs relate to complete, or at least partial, pauses in behaviors. The mouse has learned to suppress movement, and these movement suppressions are needed to acquire the reward. These movement suppressions begin at about 3-5 seconds because this is indeed the experimentally imposed lockout period. In other words, mouse behavior seems to represent a sequence of actions: pause for 3 seconds, then move.

At the end of the day, this does still relate to volition. Without a cue, the mouse chooses to start a sequence, pause then move. But the idea that mice have learned a sequence, pause then move, does not seem to align with many critical points the authors make: e.g., "structured multi-second neural dynamics preceding self-initiated action" as claimed for example in the abstract, or the paper's title. Rather, if the mice are performing a sequence, pause then move, then the initial changes in neural activity (which are detected in EDT) are not about lever press, but about the pause that starts the sequence. In other words, it seems if the mice have a state change from movement to quiescence (and then lever press), then the EDTs should be detected not relative to the lever press but the movement quiescence that begins the "sequence".

Another related issue might be the interpretation in Figure 3, where the authors discuss their finding that movement initiation tends to coincide with peaks in a slow neural oscillation. This interpretation seems potentially problematic, as this oscillation could more simply be a decrease in neural activity during the pause in motor behaviors, and then the subsequent increase during the active motor period (so basically, an ON-OFF-ON transition in neural activity that is correlated to distinct motor events rather than some latent oscillation).

Thus, while overall the mice do clearly show that they are able to volitionally control themselves for at least 3-5 seconds to obtain the reward (i.e., stop moving), this should not be taken to mean that the neural activity during this period is necessarily important to the upcoming lever press. Stated very succinctly, it is problematic to assign EDT-related neural activity to the lever press, when there is a large coincident change in the behavioral state of the animal that precedes the lever press (the complete or partial suppression of behavior). It seems more analysis on this quiescence is very important to the paper's interpretation of their results.

Data that are needed on this point would be an analysis that compares the duration of "quiescent periods" (nicely highlighted in Figure 1-Supp. 5c,d) to EDT. For example, I anticipate that occasions where the animal pauses body movements for a longer period prior to the lever press will relate to longer EDTs. On this note, it would be very helpful for the authors to provide an analogue of Figure 1-Supplement 5 (Panels c and d) for the 15-second lockout data specifically, and for all their mice as opposed to solely mouse M1. Do 15-second lockout data also show the 3-5 second quiescent periods prior to lever pulls? Another suggestion on this point: the error bars in Figure 1-Supp. 5d are large: both in the "pause" prior to time point 0, and the peak in activity at time point 0. This implies some heterogeneity in the extent to which animals pause, and the action they take after a pause. The authors could take advantage of this. If the authors binned trials based on (a) complete pause, (b) partial pause, and (c) no pause, do these groups show different neural activity patterns in the -5 to 0-sec range in Figure 2-Supp. 2 (a similar binning analysis could be done based on the length of the pause.)? If so, again, these analyses would be a reason to suspect that much of the neural signal change detected in the EDT has to do with a pause in activity prior to lever press, as opposed to the lever press itself. Trial-to-trial analyses could also be conducted on this issue, relating the pause duration (or perhaps pause magnitude, e.g., the extent to which behaviors stop prior to lever press) relates to neural activity on that trial (to this end, GLMs may be useful to parse overall activity into component behaviors).

*Reviewer #2 (Recommendations for the authors):*

This is the second review of the manuscript "Mesoscale cortex-wide neural dynamics predict self-initiated actions in mice several seconds prior to movement".

Overall, the authors have addressed in detail the concerns raised by the previous round of reviews.

On the one hand:

1) The authors have appropriately toned down claims about "voluntary" movements.

2) The authors have also performed new analyses to address the concerns about sequences of stereotyped movements but did not address the concerns of micro-movements due to a lack of proper video data. This limitation was acknowledged in the manuscript and I have no additional concerns in this respect.

On the other hand:

3) The revised results are not as strong as in the original manuscript. More rigorous analysis of the EDTs revealed that the presence of sequential lever pulls had artificially lengthened the EDT, as anticipated by the reviewers. Additionally, the authors now acknowledge that some of the results are not as reliable as previously reported (i.e., large variability across animals in neural activity phases and longitudinal changes in neural dynamics), weakening the main conclusions of the paper.

4) There are a few typos in the manuscripts, for instance, on page 11, line 224: double "also".

*Reviewer #3 (Recommendations for the authors):*

The authors have done a great job addressing the comments. The new analyses and figures have significantly improved the readability and quality of the manuscript.

I am, however, still not completely convinced by the authors' claim that they have identified a specific neural signature of self-initiated voluntary action. The authors show that pre-movement neural activity in mice contains structures that are not present in random neural activity. This observation is well supported by the data. However, to claim that this structure – neural dynamics becoming increasingly stereotyped prior to movement – is specific to self-initiated actions one needs to show that pre-self-initiated movement (uncued) neural activity in mice contains structures that are not present in externally-triggered movement (cued) neural activity. This comparison could also rule out other possible explanations such as motor confounds associated with the lever pulls or other related micro-movements. Comparison with random time during the task is not sufficient to make any conclusion regarding the self-initiated nature of the behavior.

---

## [Author Response]

Essential revisions:1) There was confusion amongst the reviewers as to the precise definitions of what the authors called "voluntary" movements. Providing a more precise definition (and the appropriate controls, as described below) will be important.

We agree with the reviewer that the term “voluntary” was not adequately defined. In fact, given the comments of the reviewers we prefer to use the term “self-initiated” as it better describes our paradigm. Thus, when referring to mouse behaviors studied in our paradigm we replaced the word ‘voluntary’ with ‘self-initiated’.

2) One of the goals of the authors was to study the neural mechanisms underlying voluntary movements. While they acknowledge (in the discussion) that they do not have evidence that actions are "intentional", they make the assumption that mice do "form the intent to act near the lever pull time".

We thank the reviewers for their comments and agree that we need to clarify the focus of study. We do not intend to solve the issue of intent in human or nonhuman animals in our study and we hope that the revisions made to the main manuscript make it increasingly clear of the findings and limitations of our work. We have now removed most all mentions of the term “intent” when referring to results from our study and changed the language to refer only to “self-initiated” behaviors in mice.

To back up this assumption, the authors should at least present some evidence that the action of interest (i.e., the rewarded lever-pull) is not just a random jerky movement that happens to be rewarded once in a while. In fact, mice seemed to pull the lever very frequently and impulsively (the majority of inter-pull intervals were way below 3 s in Supplementary Figure 1.2) even for the last sessions of the training. Therefore, it is not readily apparent that mice apply any control to their lever-pull actions. Providing evidence that the action is goal-directed is important if the goal of the paper is to study neural signatures of the intention to act.

We view habitual or innate behavior (or even “jerky movements”) as those relating to spontaneous grooming or periodic whisking, and as such could amount to instinctual, non-task-oriented behavior.

In contrast, our task is of sufficient complexity to elicit goal-directed behavior:

(a) The mice must seek out a lever in space (it is not touching their body and their right paw is not attached to it). That is, the task is not a natural or trivial one as mice must grab and grip the lever (i.e. it is not attached to their paws) and pull it in a specific direction. Rewards cannot be generated by random body movements or jerks of the paws or limbs of the mice as the lever is located to the side of the mice (see Figure 1—figure supplement 1) and does not prevent or block random limb movements.

(b) The mice must pull the lever and hold it for at least 0.1 sec (and we note many lever pulls failed this requirement).

(c) The mice must also learn to wait a minimum of 3 sec before attempting a new lever pull in order to get a water reward. Mice appear to have learned the 3sec reward lockout condition. The peaks in inter-lever-pull histograms (Figure 1—figure supplement 2), also shows that four of six mice had peaks in the inter-lever-pull histogram at ~ 3 seconds indicating that mice internalized, or “learned”, the lockout period in their behaviors and carried out many lever pulls approximately 3 secs after a previous lever pull.

(d) Moreover, we now cite previous work by the lab from Silasi et al., 2018 (paragraph from the main text of that study is included below), where mice were able to employ advanced forms of the lever pulling task described in our manuscript here during home cage training: e.g. increasing the hold duration and decreasing the goal range. As evidence that the mice were learning after training, in that study we relaxed and then escalated task difficulty (a second time) and mice were able to reach criterion significantly faster, indicating learned behaviour.

“Previously, we have evaluated the extent to which the lever pulling task reflects learned behaviour. This was assessed by finding that mice were able to adjust to increasing task difficulty: mice were able to hold the lever longer, or over a narrow rewarded goal range. These actions reflected learning and not random behavior since when criteria were relaxed and then reinstated mice progressed more quickly when trained a second time.” (Silasi et al. 2018).

(e) We also find evidence of increased lever-pull activity within each session. In response to the reviewers’ comment, we have also generated Figure 1—figure supplement 4 which shows every rewarded lever pull across all animals and sessions. This figure shows that the average number of lever pulls across each session in 4 mice (M3-M6) increases across time. That is, within each ~20minute session, these 4 mice perform increasingly more water rewarded lever pulls across the session (on average). This supports the view that the mice are aware of being in a paradigm where they can receive water rewards and are seeking to perform a behavior that results in water.

Taken together, these results support that mice learned the association between 3-sec lockout lever pulls and water rewards and engaged in lever pulls as a learned goal-directed behavior.

A somewhat compelling analysis could be to compare rewarded lever-pulls with "spontaneous" movements, provided that these two types of movement can be convincingly characterized as goal-directed vs. incidental. In contrast, throughout the manuscript, the neural activity aligned to rewarded lever-pull events (which are assumed to be "voluntary" actions) is compared to the neural activity aligned to random times during the task (whether or not it involved movements), which may not be the most convincing control.

One of our study’s findings is that upcoming self-initiated behaviors (implicitly covering “goal-oriented” and “non-goal-oriented behaviors”) do not arise from random processes, but that they have non-random neural correlates and they can be decoded prior to movement. From our original manuscript abstract:

“Several human studies have found peaks in neural activity preceding voluntary actions, e.g. the readiness potential (RP), and some have shown upcoming actions could be decoded even before awareness. Others propose that random processes underlie and explain pre-movement neural activity. Here we seek to address these issues by evaluating whether pre-movement neural activity in mice contains structure beyond that present in random neural activity. [….] Our findings support the presence of structured multi-second neural dynamics preceding voluntary action beyond that expected from random processes.”

In support of our conclusion, we compared neural activity from both self-initiated lever-pulls and other body movements to random periods of time. This paradigm replicates the human studies on this subject: our study’s paradigm is not new (or intended to be new) as both “self-initiated” and “spontaneous” behaviors have been studied in this field (e.g. the original Kornhuber and Deckee studies in the 1960s; Libet et al. 1983). Our analysis is nearly identical to that of human fMRI studies (see Soon et al. 2008, 2011; Bode et al. 2013). There is nothing inherently controversial or new in our paradigm nor in the method we implemented for computing EDTs.

Additionally, we point out that we did not design our experimental paradigm to enable decoding of self-initiated rewarded lever pulls from other body movements such as paw movements. In our study, limb movement initiations often occurred near simultaneously to the lever pulls and could not be disambiguated easily, thus making the reviewers’ suggestion difficult to implement. To show this, we generated Author response image 1 showing high correlation peaks between body movements and lever pulls:

**Author response image 1. sa2fig1:** Cross-correlation between lever pull and body movement initiations. (a) Visualizating body movement times (colored lines = boolean values) and lever pull locations (red dots) in a ~50sec window shows that many body movements occur in groups, are preceded by seconds of non-movement and often temporally coincide with lever pulls (example data was taken from mouse M1). (b) Zoomed in section from (a). (c) The cross-correlation between lever pull initiation time and the left paw, right paw and licking bout initiations (for session data in (a)). (d) The average (normalized) cross correlation (continuous lines) and standard deviation (shading) for all sessions in mouse M1 between lever pull and left paw, right and tongue movement invitations. (note: we obtained similar results for mice M2-M6).

In sum, panels a-b show that lever pulls generally occur after quiescence periods, i.e. there is a decrease in body movements and licking activity in the few seconds prior to a lever pull. Panel c shows no strong peaks in body movements preceding the lever pull on the scale multiple seconds. Panel d shows that in the period of approx. -5 to approx 0 seconds prior to a lever pull there is a decrease in correlated activity between body movements and licking.

These results suggest that: (i) many body movements coincide with the lever pull time (making them more difficult to disambiguate); and (ii) that mice tend to decrease their overall body movements a few seconds prior to lever pull. Taken together, these results support that neural signals in the preceding few seconds prior to lever pulls likely represent ongoing neural dynamics of behavior preparation rather than performance (see below also for further analysis on this point).

We have made changes to the manuscript to address the reviewers’ comments on this particular issue to be more consistent with the paradigm and findings. For example, we added this paragraph to

“We developed an analogous self-initiated behavior paradigm in six mice (M1-M6) to characterize pre-movement neural activity while recording widefield [Ca++] activity from cortex (Figure 1c, d, Figure 1—figure supplements 1,2; see also Methods). Mice were head fixed and trained to perform a self-initiated lever-pull to receive a water reward without sensory cues or stimuli. Four of six mice learned the lever lockout period of 3 sec (Sup Figure 1.2 shows peaks at 3-sec in the inter-lever-pull intervals in mice M3-M6) and four of six mice learned to pull increasingly more often towards the end of each ~20 minute session (Figure 1—figure supplement 4, mice ). Mice tended to decrease their body movements prior to a lever pull and we did not find evidence of stereotyped behaviors prior to lever pull (Figure 1—figure supplement 5; see also Methods section on detecting stereotyped movements).”

3) The learning trajectory of mice is also not well characterized (e.g. changes in inter-pull intervals are not quantified, nor the relative increase in rewarded actions across training sessions, etc.). Yet, several claims in the paper are directly based on the fact that mice have learned to pull the lever after 3 s interval to receive water rewards (which relates to point 1).

We agree with the reviewers' concerns regarding the lack of characterization of task learning in our original manuscript. We have addressed this in Essential Revision Point 2 (see comments above). There, we pointed to the design of the task including the presence of lockouts, duration of lever hold and systematic increase in lever pulls across a session. We also pointed to a recent paper in our lab using a similar lever pull paradigm with increasingly challenging parameters that mice also performed well on (Silasi et al. 2018). While lever-pulls were central to our study, learned advancement of lever difficulty was not required and we focused on the cortical activity patterns around each action.

In particular, one important assumption in the paper is that as mice learn, the lever-pull movements become more stereotyped, but this has not been shown explicitly. It would be helpful, for example, to see how analog traces of lever-pulling change throughout the learning stages and how the variance of the movement across trials decreases in late sessions.

We regret this confusion potentially caused by our manuscript’s original language (e.g. Figure 4), but our claim with respect to longitudinal performance is primarily that neural activity becomes more stereotyped – not paw trajectories.

While it is not central to our study we agree with the reviewer that it is possible that trajectories of the right paw could become more stereotyped over time (e.g. see Kawai et al. 2015). Although not directly affecting our main findings, we have generated Author response image 2 to evaluate any stereotyped structure in the movements of the right paw in mouse M6 (which had by far the most sessions and recording hrs: 70 sessions and > 13 hours of video recordings).

**Author response image 2. sa2fig2:** Evaluation of systematic longitudinal changes path in right paw movement trajectories. Panel (a) shows UMAP applied on 1-sec long right paw movement segments from all sessions in mouse M6. For clarity, we took the DeepLabCut detected locations of the right paw, split them into continuous, non-overlapping 1 sec segments (and centered them, i.e. removed any offset as we were interested in relative changes in the 1 sec period), and fed it into UMAP (i.e. each example had dimensions of: [n_time points, n_spatial_locations] = [15, 2] in this panel). Panel (b) is the same as (a) but for 5 sec long movement snippets.

In sum, we do not find clusters of stereotyped activity nor correlations between the UMAP distribution shape and session time.

4) The central claim of the paper is that rewarded lever-pulls can be predicted from pre-movement neural activity several seconds (even up to 10 s) prior to the action. However, obvious motor confounds and other alternative explanations have not been convincingly ruled out. In fact, the action of lever pulling may require a series of complex movements (like changing posture, extending the forelimb, reaching the lever, grabbing the lever, etc.). The authors themselves mentioned that they found strong correlations between lever pulls and body movements in all mice, but the data is not used nor shown in the paper. The motor commands preceding but related to lever-pull could unfold at least a few hundreds of milliseconds prior to the detection of lever-pull in the task, and thus be reflected in the neural activity that is predictive of the lever pull. Moreover, if this series of movements is highly stereotyped, and in turn leads to stereotyped neural activity (like the slow oscillations observed before the lever-pulls), it could explain why the detection of lever-pulling actions always occurs at a given phase of the neural oscillation.

We agree with the reviewers that analyzing the presence and effect of pre-lever pull movements is important to our study’s findings relating to the absolute values of EDTs. We provide the following evidence that such stereotyped series of movements do not exist in the behaviors of our mice:

In sum, this figure shows that in the few seconds prior to lever pulls, mice tend to decrease their average number of body movements, with some sessions (as the example session shown in panel c) having very few behaviors prior to lever pulls. Additionally, we used PCA to search for correlated sequences of activity and were not able to identify any (see Methods for more details on this analysis).While we agree with the reviewer that motor command preparation of lever pull behaviors likely takes O(100ms) or more and is present in pre-movement neural activity, we did not find evidence that stereotyped behavior sequences lasting a few to several seconds prior to lever pulls were present in our data. Our data suggests that the 3sec lockout has the effect of causing mice to enter into increasingly quiescent states prior to attempting a lever pull for water reward.

Such observations that stereotyped movements occur way before the lever-pull detection could partially rule out the fully "cognitive" explanation proposed in the paper, but would concur with recent findings that showed that ramping neural activity can be, for the most part, explained by movement-related activity (Musall et al., 2019).

As shown above, we do not find evidence for stereotyped movements occurring “way before” lever pulls.

With respect to Musall et al. 2019: that study was more focused on decoding contemporaneous behaviors whereas our study focused on detection of future behaviors. The behavior paradigm in Musall et al. 2019 contains short periods of behavior, e.g. holding levers for 1 sec, processing stimuli for 0.6 sec, a pause of 0.5 sec, a stimulus for 0.6sec, and a lick movement after a 1 sec window. The findings of that study suggest that while performing learned tasks during these different task segments, mice can also engage in non-task related behaviors that have neural correlates which are detectable in widefield [Ca] recordings.

The paradigm we study in this manuscript is different from that studied by Musall et al. 2019 due to its stimulus-free nature where the only constraint was a minimum of 3 sec of task lockout (or 15 seconds of post-hoc lockout). Importantly, while spontaneous behaviors (and their neural correlates) can occur while mice are head fixed – there is no evidence in Musall et al. 2019 that there are sequences of behaviors preceding rewarded behavior.

As additional evidence that mice do not transition between multiple behavioral states during the period of approx -5 sec to – 3sec, we also generated Figure 2—figure supplement 2 which shows that there is only one neural motif occuring during this period.

The panel shows an increase in inhibition across forelimb (red), hindlimb (purple) and motor (blue) areas lasting a few seconds and then a rapid increase in neural activity in the 1-2 seconds prior to movement. There is no evidence for multiple phases or oscillations in this motif. This single phase of inhibitory dynamics is inconsistent with those supporting body movements (which generally require activation of cortex; though the time course of the [ca] reporter may affect the dynamics). This time course is also inconsistent with the presence of multiple sequential movements.

In sum, we did not find evidence for the presence of multiple behavioral motifs as identified in Musall et al. 2019. We find evidence only for a single neural motif which involves distributed inhibition over many areas during a behavior preparation period of several seconds prior to lever pull followed by a return to baseline prior to lever pull and an increase in activity past t=0 sec (shown in other panels, e.g. Figure 1).

5) Toward the end of the result section (Figure 6), the authors briefly begin to address the issue about whether pre-movement activity can really be considered movement free. Here, "lockouts", i.e. periods where other movements (like licking, or previous lever-pulls) did not occur, were introduced in the analysis. The lockouts altered the earliest-decoding-time (EDT) of the lever-pull (in some mice EDT was even divided by half: from -4 s to -2 s). However, the effects of "micro-movements" like facial movements or changes in body posture may not be taken into account with the lockout approach. Such micro-movements have been shown to explain a large variance of the neural activity (see Stringer et al. 2019 and Musall et al. 2019). Therefore, to fully control for movement confounds, the effect of high dimensional/micro-movements extracted from video recordings should be removed from the neural activity. These analyses could yield a much shorter EDT (e.g., -0.15 s), more consistent with previous reports.

We thank the reviewers for this concern regarding micro-movement effects on neural activity. Given our video recording and experimental paradigm, we were not in a position to track or regress out micro-movements including those occurring in facial muscles. It is indeed possible that multiple forms of “micro-movements” could occur during the period prior to a level pull.

However, we do not believe that such micro-movements – even if present in our study – would affect EDTs obtained in our study for reasons outlined above and summarized here.

(a)Neither Musall et al. 2018 nor Stringer et al. 2019 reveal the presence of sequences of body movements (or micro-movements) before rewarded behavior, nor of such body movement sequences lasting a few to several seconds.(b)We did not find evidence for stereotyped sequences of movements prior to a lever pull (Essential Revision Point 3).(c)We did not find evidence for the presence of a series of movements in the neural data (Essential Revision Point 4).(d)The average neural dynamics occurring in the period of approx. -5 to 0 seconds prior to lever pulls is inhibitory and is inconsistent with the presence of multiple movements which generally require increases in neural activity – not inhibition (Essential Revision Point 4).(e)The average neural dynamics occurring in the period of approx. -5 to 0 seconds prior to lever pull has a single multi-second (inhibitory) phase which is unlikely to support multiple complex movements or micro-movements.

Based on the reviewers’ suggestion, we have now added the following paragraph in the discussion on this topic.

“The effect of confounding movements on the decoding of future rewarded action. Our study does not directly address the effects of stimuli or other perturbations on the neural dynamics preceding self-initiated decisions, for example, as in some human choice paradigms that consider decision choice in the presence of novel information (e.g. Maoz et al. 2019). Additionally, there is evidence that micro-movements can contribute to ongoing neural activity between behavior bouts (Stringer et al. 2019 and Musall et al. 2019). While it is possible that in preparing for a self-initiated rewarded behavior mice undergo a series of physical movements, we did not see any evidence for this in our recordings. In particular, we found a tendency for mice to decrease their body movements in the 3-5 seconds prior to self-initiated action and we did not find evidence for sequences of paw or licking movements. We suggest future studies focusing on the micro-movements underlying self-initiated action could address this issue using high-frame and high-resolution video recordings.”

6) SVM decoding accuracy in Figures 1h, 2b, 2d, etc. shows in some cases that the earliest decoding time can precede a lever pull by 10-15 seconds. But how are we to know that SVM predictions in a given 1-second window are predicting an upcoming movement 10-15 seconds in the future, versus a separate pulling event that occurs within the SVM window on some subset of trials in a session? The increase in SVM accuracy as the movement approaches may be (as in A) due to changes in probability and temporal coupling in pulling events. The data are aligned to lever pull n, hence maximizing SVM accuracy at that time point. It is unclear whether the authors can address such concerns with the existing data. One analysis that would be telling would be something akin to GLM. For example, in Figure 1e, are the oscillations due to multiple movements occurring in succession? The authors could fit a Gaussian to the last peak (from about -2 sec to 2 sec) in Figure 1e. Then the authors can take this Gaussian and convolve it with each lever pull event, and then add all these together into a composite trace that represents what neural activity should look like if it is only determined by lever pull events that follow some Gaussian distribution. Figure 1e could be reconstructed from this idealized timeseries as if it were the actual raw data. Should this control also show oscillations with increasing amplitude, this suggests that the neural activity in Figures 1e (and the SVM analyses by extension) is due to multiple lever pulls – not volitional activity that predicts an upcoming lever pull.

We agree with the reviewer’s concerns that sequential lever pulls could affect the EDT decoding times. Our original manuscript raised this concern as noted in the introduction to Figure 6:

“For example, in animals that perseverate and pull the lever frequently it is not known whether decoding methods leverage dynamics from multiple lever pulls or just the lever pull occurring at t=0sec.”

The issue raised here in Essential Revision Point 6 (whether simulations provide insight into EDTs and oscillations), Essential Revision Point 8 (interpreting the effect of lever-pull lockout) and Essential Revision Point 9 (effect of downsampling non-lever pull lockout data) are all related to the effect of sequential lever pulls on decoding. We largely agree with the reviewers’ concerns and generated additional figures and analyses to address these points (discussed below). Briefly, our findings are that:

(a)We find evidence that sequential lever pulls can generate stereotyped oscillatory-like activity that likely decrease EDTs (Essential Revision Point 6).(b)While “locking out” previous lever pulls for periods of 15 sec (to remove the effect of sequential level pulls) shortens EDTs averages by 1-2 seconds (as shown in original manuscript Figure 6), filtering the neural dynamics prior to EDT computation results in EDTs that are lengthened by 1-3 seconds. This filtering step results in 15 sec lockout EDTs averages falling between -3 to -7 seconds (see updated manuscript and below). (Essential Revision Point 8).(c)Subsampling non-lockout data to match lockout data results in average EDTs shortening by 1-2 seconds, similar to locking-out effects. This supports that pooling across longitudinal data decreases the ability to decode behavior (Essential Revision Point 9).

We proceed to address the specific issue in Essential Revision Point 6, i.e. whether simulated data provides insight into the sequential lever pull issue (we leave the analysis for Essential Revision Points 8 and 9 below). Based on the reviewers’ comments we generated the Author response image 3. Consistent with the reviewers suggestions, we find that in some sessions simulated neural data based on lever pull times can generate oscillations similar to those observed in the neural data. Additionally, oscillations in locked-out neural data seem to decrease for some animals.

**Author response image 3. sa2fig3:** Generating synthetic “neural” data by convolving lever pull times and gaussians. (a) A synthetic neural signal obtained by convolving each rewarded lever pull time within a single session with a gaussian kernel note: we used 1.5sec std gaussian, which gives about a 3.5-4.5sec wide signal; 1 sec or 2 sec std gaussians gave similar results; random data was generated by shifting the times by a random value drawn from U(-25sec,25sec). (b) Average synthetic neural data (blue) centered on the lever pulls and the random control neural signal (red). (c) Four example sessions from different mice showing synthetic data (note: M4 had many trials and shuffling -/+25 secs still yielded a “bump” in the random average; a higher shift value removes this effect). (d) The real (not simulated as in a-c) average of motor cortex activity for no lockout (blue) vs. 21 sec lockout (red; note: we used the 21 sec as the longest window still yielding sufficient data) conditions shows that some animals continue to have small oscillations even when locking out previous lever pulls.

In sum, this figure shows that simulated neural data based on the number and times of level pulls can generate oscillations observed in real neural data (Panels a,b,c). Consistent with the reviewer’s concerns, some of oscillations observed in real neural data might thus not be related to preparation of the decoded lever pull but a previous lever pull. Oscillations are often decreased in real neural data when locking out sequential lever pulls (Panel d: decrease in lockout curve (red) oscillations compared to all trials (blue) in mice M1, M4 and M6). However, we also observe that even after lockout real neural data small oscillations can be present in some animals (e.g. note small red oscillations in Panel (d) M1 and M3). Overall, we agree with the reviewers that multiple lever pulls can have a causal effect on neural activity. We added the following text to the main manuscript (along with several changes as described below):

“We find that oscillations observed in the neural data are likely enforced by repetitive and stereotyped recent lever pulls and that EDT analysis requires exclusion of trials that occur too soon after a previous lever pull (here we chose a lockout of 15 seconds).”

Similarly, as a control, the authors should also re-plot neural activity in Figures 1e, 5a, etc., using the longer post-hoc lockouts employed in Figure 6. That is, the authors should re-plot activity but exclude all trials where a previous lever pull occurred within x seconds, where x is 6, 9, 12, etc. as in Figure 6g.

We agree with the reviewers’ comments and have amended the manuscript as follows:

(a)Figure 1e was replaced with a panel generated from trials where a 21sec lever-pull lockout was enforced (we chose 21sec to reflect the time range of the original panel).(b)We replaced Figures 1h with decoding curves generated following a 15 sec lockout:

In response to this Essential Revision Point (and others below), we made several additions and changes to our manuscript including the following statements (in multiple sections of the manuscript):

“To disambiguate the effects of multiple sequential lever pulls, we considered only lever pulls that were preceded by at least 15 seconds of no-lever pull activity (see further results below and Methods on lever lockout analysis).”

“Given the strong dependence of EDT on the # of trials, we sought to re-evaluate EDTs using only trials that were not preceded by another lever pull (either rewarded or non rewarded). We find that oscillations observed in the neural data are likely enforced by repetitive and stereotyped recent lever pulls and that EDT analysis requires exclusion of trial that occur too soon after a previous lever pull (here we chose a lockout of 15 seconds; note: this approach significantly decreased the number of trials available for analysis as mice only rarely went without pulling the lever for 15 seconds; we thus pooled trials from across sessions into a minimum of 50 to a maximum of 200 trial hybrid-sessions; see Methods). We found that after lockout the neural data had a single negative (i.e. inhibitory) phase preceding self-initiated rewarded lever pulls that comenced ~5 seconds prior (similar to Figure 1e; see Figure 2—figure supplement 2). We additionally found that EDTs decoded from lockout trials were shorter (Figure 2g: average EDT in seconds: mouse 1 (M1): -1.93; M2: -3.14; M3: -2.27; M4: -1.87; M5: -1.64; M6: -2.49). However, low pass filtering the neural time series (at 0.3hz) (as a type of feature engineering based on power-analysis results in Figure 7 below) resulted in EDTs more similar to the initial results (Figure 2h; average EDTs of causal filtered neural data in seconds: M1: -3.5; M2: -4.85; M3: -6.95; M4: -4.31; M5: -3.0; M6: 3.7). The improvement in EDT was qualitatively observable in decoding accuracy curves (Figure 2i) and was present even for non-lock out trials (see Figure 2—figure supplement 4; see also Methods).

The initial loss of EDT (without the filtering step) suggests that sequential lever pulls might have a causal role in lengthening EDTs by generating stereotyped neural time series which represents preceding – not just the current – rewarded lever pulls. However, we also found that pooling trials from sessions far apart in time (days or weeks) as required by the lockout method also also shortened EDT values (i.e. closer to 0 sec; Figure 2—figure supplement 3). This suggests that higher data variance (due to learning, [Ca] bleaching, implant degradation etc.) might also have a causal role in shortening EDTs. Overall, these findings show that self-initiated water rewarded lever pulls in mice can have neural correlates that are present up to several seconds prior to lever pull and can be decoded several seconds prior to level pulls, but that such analysis must appropriately take into account previous behaviors and the effects of data variance over longitudinal studies.”

“…While sequential stereotyped pulls have the effect of lengthening EDTs, longitudinal changes in the neural recordings have the effect of shortening EDTs.”

“… Thus, surviving trials used for analysis came from sessions that were increasingly further apart (e.g. multiple days or even a week). Pooling trials from separate days or weeks provides an additional source of noise due to changes in animal behavior, [Ca++] indicator properties, and longitudinal network changes observed in our cohorts (see Figure 4). As evidence for this, we found that subsampling the number of non-lockout trials to match the number of trials following 15sec lockout had the effect of shortening most EDTs (Figure 2—figure supplement 3).

We also recomputed locked-out EDTs for each animal (as in (g)) but following low-pass filtering the neural time series (filter set to 0.3Hz; see Methods) as described above (Figure 6i). We found that average EDTs detected were longer than using non-filtered data (Figure 6j; e.g, 15 sec lock out data means: M1: -3.50; M2: -4.85; M3: -6.95; M4: -5.01; M5: -3.0-; M6: -3.70).

Taken all these factors into account, the results suggest that while EDTs increased in some animals when using only lockout trials, the cause of the increase is due to: (i) removing stereotyped sequential lever pulls which could artificially bias the neural signal; and (ii) increased variance in the longitudinal data caused by [Ca++] state changes, systematic neural network restructuring due to longitudinal performance and other unknown factors”.

7) Another related concern seems to be the convex hull analysis in Figures 4d-g. For example, in Figure 4e it is shown that the convex hull decreases in size as the movement (t=0 sec) approaches. The authors conclude that neural activity looks more random prior to the lever pull. However, given the concerns above, it seems this "randomness" could actually be due to other lever pulls (and other spontaneous behaviors) that are not exactly temporally aligned, and not certain to occur in each sample. Thus, we would also ask the authors to repeat this analysis using longer post-hoc lockouts to prevent contamination from earlier lever pulls.

We agree with the reviewers’ concern and updated figures as follows:

(a)Figure 4b was updated to show two examples of decoding accuracy curves from 15 sec lockout (and smoothed) neural time series.(b)Figure 4c was updated to show the trends from 15 sec lockout, filtered time series.(c)Figure 4g was updated using 15 sec lockout data.

We updated the text to reflect these changes.

“…We carried out this analysis using only 15 sec lockout data grouped in sessions of up to 200 trials (similar to carried out above to exclude any possible trends arising from increased intra-session lever pulls or effects of sequential lever pulls; see Methods). We found that, as in the EDT longitudinal trends, 2 mice (M1, M6) that had decreasing EDTs (i.e. poorer decoding over time) also had an increased similarity (i.e. increased AUC values) between lever pull dynamics and random neural states. Considering only the statistically significant results, one explanation may be that cortical dynamics may return to natural, i.e. pre-learning, patterns and look increasingly the same as random neural states (occurring near or far from behaviors) as subcortical-cortical circuits increasingly facilitate and “take over” self-initiated behavior preparation during automaticity processes.”

(Note: Figure 4d only shows the last second of neural dynamics prior to a lever pull and was thus not affected by introducing longer lockouts; Figure 4e was generated from a session with few short inter-lever-pull intervals; and Figure 4f is a qualitative sketch demonstrating our method; we noted in the Figure legend these changes and notes).

Additionally, we made several changes to the main manuscript including the following statements:

“Longitudinal cortical network dynamics support shortening of EDTs. Considering the number of rewarded lever pulls per session, we found that between the first and last days of the experiment three of the mice (M2,M3 and M6) increased the number of rewarded lever pulls while one additional mouse (M5) also had a positive trend (with pval of 0.13); one of the mice (M1) decreased its number of pulls per day and the remaining mouse did not have statistically significant changes (Figure 4a; pearson correlation values provided in figure insets; see Methods). We labeled the 4 mice with either a strong or a trend in positive correlation over time as the “performer” group (M2,M3,M5 and M6) and the remaining mice (M1, M2) as “non-performers” as they did not increase their pulls over time. Given the potential confounds identified in Figure 2 between sequential lever pulls and decoding time, further analysis in this section focused primarily on 15 sec lockout trials only (see Figure 2). We found that SVM decoding accuracy curves over the weeks or months of behavior revealed potential trends over time (Figure 4b for examples from 2 mice). In particular, we found that EDTs shortened (i.e. were closer to the lever pull time) over time in 2 mice (M1 and M6); a similar trend was present in another mouse (M5; pval 0.06); while the remaining three mice did not show statistically significant trends. Although only statistically significant in two mice, shortening of EDT decoding time trends may be explained by automaticity findings in other studies: i.e. that following learning and repetitive behavior performance, the control of behavior is transferred from cortical structures (which we had access to during widefield [ca] imaging) to subcortical structures (that we could not access in our paradigm; another explanation could be that implants slowly degraded) (see e.g. Ashby et al. 2010; see also Discussion). ”

“Lastly, we evaluated the area under the ratio curve (AUC) longitudinally to evaluate whether there are systematic changes in the neural activity convex hull over weeks of behavior performance (Figure 4g). We carried out this analysis using only 15 sec lockout data grouped in sessions of up to 200 trials (similar to carried out above to exclude any possible trends arising from increased intra-session lever pulls or effects of sequential lever pulls; see Methods). We found that, as in the EDT longitudinal trends, 2 mice (M1, M6) that had decreasing EDTs (i.e. poorer decoding over time) also had an increased similarity (i.e. increased AUC values) between lever pull dynamics and random neural states. Considering only these two mice (as they were the only statistically significant results), one explanation may be that cortical dynamics may return to pre-lever pull learning patterns and look increasingly the same as random neural states (occurring near or far from behaviors) because subcortical circuits increasingly facilitate and “take over” self-initiated behavior preparation during automaticity processes. ”

“…While 5 of the 6 mice had increasing convex hulls, the trends were statistically significant in only 2 of the mice, which again, were mice M1 and M6 (although mouse M3 also showed a similar trend, p value: 0.14) (Figure 4i). Interestingly, mouse M4 showed a decrease in overlap of lever dynamics with all dynamics. These mixed results suggest that different mouse-specific mesoscale neural representations may be involved during learning and performing of a task.

We also found mixed results with respect to the intersection between the right paw convex hull, i.e. the paw used to pull the lever, and the lever pull convex hulls: the overlap decreased with time in 2 of the mice (M4 and M6) and showed a similar trend in another mouse (M3; pval 0.09); and it increased with time in mouse M1. These results suggest that in some mice (M4, M6 and possibly M3) lever pull neural dynamics increasingly specialize or differentiate from non-lever pull right paw movements neural dynamics (despite the right paw being used for the lever pull task). In contrast, in one mouse the similarity between right paw movements and lever pulls increased (e.g. mouse M1), but this could be explained by this mouse being a behavior outlier as the only mouse with decreased number of rewarded lever pulls over time..

In sum, EDTs shortened longitudinally in some mice suggesting neural dynamics underlying self-initiated behavior might be transferred from cortical to subcortical circuits decreasing the power of cortical-based decoding methods. The convex hull of the neural activity prior to self-initiated lever pulls also increased over time in some mice with a potential explanation that cortical dynamics return to pre-learning similarity over time. Lastly, right paw and lever dynamics appeared to become increasingly dissimilar in a few mice, with one mouse showing the opposite trend. These findings suggest that learning or mere longitudinal performance of a task restructures the neural dynamics underlying self-initiated action but that the effects could be subject specific, drawing attention to the need for intra-animal analyses (rather than cohort) in future studies.“

8) It is excellent that the authors considered the issue described in Point (1), in their control analysis in Figure 6. But the reviewers disagree with their conclusions. Figures 6g and 6h clearly show that EDT drops considerably as the lever lockout window increases. For example, the 9, 12, and 15 second lever lockout periods appear to mostly eliminate all sessions with EDTs greater than 5 seconds. This is a drastic reduction in the time horizon (10-15 seconds) argued by the authors given their earlier results.The authors might argue that these results are mixed, given that they were not statistically significant in M1 and M2 (Figure 6g). But this is misleading. In M1, there are only 2 or so EDTs greater than 5 seconds in the 3 second lockout condition. There are none in the 6 second lockout condition. Thus, in this mouse, the lack of statistical significance is simply because there are little to no EDTs to eliminate by gradually increasing the lever lockout period. This analysis is not useful in a mouse that exhibited little to no EDTs in the 10-15 second range argued by the authors. For M2, it is surprising that no statistically significant effect was detected given that all EDTs greater than 5 seconds are eliminated in the 12 and 15 second lockout conditions.

We thank the reviewers for their comments on Figure 6. In response to this point we generated 2 additional Supplementary Figures and made two additional panels to Figure 2, as described below. Briefly, our results show that an additional preprocessing step (low pass filtering) results in average EDTs of approx -3 seconds to approx. -7 seconds even for 15 sec lockout data.

Figure 2—figure supplement 4 shows that low-pass filtering (0.3Hz) the neural time series as a type of feature engineering results in EDTs (for non-lockout data) that decrease by a few seconds in most sessions.

We also added an additional panel to Figure 2 to recompute EDTs using low-bandpass filtered neural components (i.e. the same analysis as Sup Figure 2.4 but on 15 sec locked out data).

This figure shows that lowpass filtering the neural time series of locked out trials also yields lower EDTs (we also note that filtering to 0.3Hz makes the signal closer to fMRI time dynamics studied in humans, e.g. Soon et al. 2008).

Following the reviewers comments in Point 8 and the results above, we added a new panel to Figure 6 which shows EDTs following lockout pre- and post-low bandpass filtering.

These results show that although excluding sequential lever pulls decreases decoding power (i.e. shortens EDT times), EDTs obtained from low-pass filtered data can still range between -3sec to -7sec (average EDTs for 15 sec lock out filtered data: M1: -3.50; M2: -4.85; M3: -6.95; M4: -5.01; M5: -3.0; M6: -3.70).

There is a clear trend here despite the statistical results, which calls into question the statistical technique used to assess the relationships in Figure 6g. Could the authors report their statistical technique? As far as we can tell this is not described in the paper.

We apologize for this oversight. For Figure 6 we used the same test as in Figure 2f and g (and can be found in the original manuscript), namely, a 2-sample KS test with asterisks indicating: * <0.05; **<0.01; ***<0.001; ****<0.0001; *****<0.00001. We have also added exact p-values throughout the manuscript in response to this issue.

Furthermore, is there a way the authors could analyze outlying mice behavior (M1 and M2) to understand why their results differ? M1 in particular may be very telling, because they have very few "large" EDTs to begin with as shown in Figure 6g? Can this be attributed to some behavior that M1 did (or did not do) as compared with other mice?

We agree with the reviewers that mouse M1’s behavior was an outlier. Mouse M1 seemed the least “motivated” mouse and had the least # of pulls overall, including a decrease over time in the number of pulls it did in each session and the number of lever pulls it carried out across each individual session (Figure 4a shows decrease in # of rewarded lever pulls over time). Figure 1—figure supplement 2 also suggests that neither M1 or M2 acquired the strong 3-sec peak in the inter-lever-pull interval histogram. It is possible M1 did not require as much water as other mice. It is not possible to know whether mouse M1 – unlike other mice – required less intention or preparation time to pull the lever or experience less stress towards getting the water reward. However, the 15 sec filtered lockout results from Figure 6 show that after filtering, EDT distributions of mouse M1 for 3sec lockout look more similar to other mice. The 15 sec behavior lockout yielded an average EDT of -3.5, which is comparable to other mice.

We have also added text to the main manuscript

“… EDTs computed when excluding previous lever pulls for a period of up to 15 seconds prior shortened EDTs in most animals, and average EDTs for band-passed neural data ranged from approximately -3sec to -7sec. We conclude that it is critical to exclude sequential lever pulls to the computation of EDT as well as track and control for longitudinal changes in neural dynamics caused by learning or implant related causes.”

9) Overall, it is surprising that the observed clear and large reductions in EDTs in Figures 6g and 6h are dismissed or downplayed by the authors at several points. For example, the authors state "Taking all these factors into account, the results suggest that while EDTs increased slightly in some animals, the cause of the increase could not be disambiguated between [Ca++] state changes, animal behavior, systematic neural network restructuring due to longitudinal performance…". The title of this section reads "Lever pull EDTs are similar or slightly higher with increasing lever-pull lockout duration". The Discussion again states that "Self-caused movement confounds have only minor effects on the decoding of future rewarded action." First, the increase in EDTs is substantial (for example, in Figure 6h, the mean EDT across mice seems to increase from about -4.5 to about -2.5 sec, almost a 50% change) and calls into question the primary results.

We agree with the reviewers' concerns and regret the cursory way in which we tackled this issue in the original manuscript. We have generated additional analysis and text (see Essential Revision Points 6 and 8 above).

And second, while we can appreciate that many factors can worsen EDT detection as the lockout period increases (e.g., fewer sessions that occurred further apart), it is not sufficient to explain away the clear and concerning trends by listing several limitations. Especially when it seems that there may be ways to test this. For example, the authors could downsample lever pull events in the 3 sec and 6 sec lockout groups, etc., in order to match the sample size across the 5 conditions, and then repeat their analysis. Additionally, if their concern is that sessions are further apart in the longer lockout conditions, they could resample trials in the smaller lockout conditions in such a way to match the "timing" of samples across the 5 conditions, and then repeat their analysis.

We thank the reviewers for the suggested analysis and have generated Figure 2—figure supplement 3 to address this point. This figure compares EDTs from non-lockout trials (i.e. all trials) vs. lockout trials (15 sec) vs non-lockout trials that are uniformly subsampled to match the # of trials in the 15 sec lockout condition.

Taking the reviewers’ comments and these results into consideration, we modified our main manuscript to indicate that sequential lever pulls tend to decrease EDTs and should be excluded from analysis, although this leads to a fewer number of trials that has the effect of shortening EDTs.

Reviewer #1 (Recommendations for the authors):Recommendations for the author1. Our main concern with the authors' study is the lack of quiescence prior to lever pulling events. The inter-lever-pull distributions in Supplementary Figure 1.2 show that many pulls will occur within the 15-20 second window prior to a given lever pulling event. This is problematic. Suppose we analyze pull n. How are we to know that neural activity prior to lever pull n relates to that pull, when lever pulls n-3, n-2, n-1, etc., occur within the analysis window?

We agree with the reviewer’s concern and have addressed these issues in depth in the Essential Revision.

It seems very possible that several key results may be altered by this confound, as outlined below:A. The elongated and oscillatory neural activity profiles in Figure 1e, Figure 3a, Figure 5a, and Figure 7a seem to us to reflect a sequence of lever pulling movements. For example, in Figure 1e, there are oscillations in Ca++ activity, with 4-5 peaks. The authors' attribute the change in peak amplitude to an increase in stereotypy as the movement approaches. An alternate hypothesis here is that the change in amplitude is related to the probability and alignment of successive pulling events. For example, there are two large peaks because two pulling events are likely to occur together. The smaller earlier peaks reflect that a third, fourth, and fifth pulling event can occur, but with a smaller likelihood and a less temporally precise coupling to pull n.

We agree and this issue has been discussed in the Essential Revision resulting in additional panels and changes to the main text.

B. SVM decoding accuracy in Figures 1h, 2b, 2d, etc. shows in some cases that the earliest decoding time can precede a lever pull by 10-15 seconds. But again, how are we to know that SVM predictions in a given 1-second window are predicting an upcoming movement 10-15 seconds in the future, versus a separate pulling event that occurs within the SVM window on some subset of trials in a session? The increase in SVM accuracy as the movement approaches may be (as in A) due to changes in probability and temporal coupling in pulling events. The data are aligned to lever pull n, hence maximizing SVM accuracy at that time point.

We agree and have adjusted our results and panels accordingly.

It is unclear whether the authors can address such concerns with the existing data. One analysis that would be telling would be something akin to GLM. For example, in Figure 1e, are the oscillations due to multiple movements occurring in succession? The authors could fit a Gaussian to the last peak (from about -2 sec to 2 sec) in Figure 1e. Then the authors can take this Gaussian and convolve it with each lever pull event, and then add all these together into a composite trace that represents what neural activity should look like if it is only determined by lever pull events that follow some Gaussian distribution. Figure 1e could be reconstructed from this idealized timeseries as if it were the actual raw data. Should this control also show oscillations with increasing amplitude, this suggests that the neural activity in Figures 1e (and the SVM analyses by extension) is due to multiple lever pulls – not volitional activity that predicts an upcoming lever pull.

We agree and have adjusted our results and panels accordingly.

Similarly, as a control, the authors should also re-plot neural activity in Figures 1e, 5a, etc., using the longer post-hoc lockouts employed in Figure 6. That is, the authors should re-plot activity but exclude all trials where a previous lever pull occurred within x seconds, where x is 6, 9, 12, etc. as in Figure 6g.

We agree and have adjusted our results and panels accordingly.

Finally, another related concern seems to be the convex hull analysis in Figures 4d-g. For example, in Figure 4e it is shown that the convex hull decreases in size as the movement (t=0 sec) approaches. The authors conclude that neural activity looks more random prior to the lever pull. However, given the concerns above, it seems this "randomness" could actually be due to other lever pulls (and other spontaneous behaviors) that are not exactly temporally aligned, and not certain to occur in each sample. Thus, we would also ask the authors to repeat this analysis using longer post-hoc lockouts to prevent contamination from earlier lever pulls.

We agree and have adjusted our results and panels accordingly.

2. It is excellent that the authors considered the issue described in Point (1), in their control analysis in Figure 6. But we disagree with their conclusions. Figures 6g and 6h clearly show that EDT drops considerably as the lever lockout window increases. For example, the 9, 12, and 15 second lever lockout periods appear to mostly eliminate all sessions with EDTs greater than 5 seconds. This is a drastic reduction in the time horizon (10-15 seconds) argued by the authors given their earlier results.

We agree and have adjusted our results and panels accordingly. We provide explanations and figures showing evidence for our original comments (including that pooling data longitudinally affects EDTs significantly).

The authors might argue that these results are mixed, given that they were not statistically significant in M1 and M2 (Figure 6g). But this is misleading. In M1, there are only 2 or so EDTs greater than 5 seconds in the 3 second lockout condition. There are none in the 6 second lockout condition. Thus, in this mouse, the lack of statistical significance is simply because there are little to no EDTs to eliminate by gradually increasing the lever lockout period. This analysis is not useful in a mouse that exhibited little to no EDTs in the 10-15 second range argued by the authors. For M2, it is surprising that no statistically significant effect was detected given that all EDTs greater than 5 seconds are eliminated in the 12 and 15 second lockout conditions. There is a clear trend here despite the statistical results, which calls into question the statistical technique used to assess the relationships in Figure 6g. Could the authors report their statistical technique? As far as we can tell this is not described in the paper. Furthermore, is there a way the authors could analyze outlying mice behavior (M1 and M2) to understand why their results differ? M1 in particular may be very telling, because they have very few "large" EDTs to begin with as shown in Figure 6g? Can this be attributed to some behavior that M1 did (or did not do) as compared with other mice?

We agree and have adjusted our results and panels accordingly.

Overall, it is surprising to us that the observed clear and large reductions in EDTs in Figures 6g and 6h are dismissed or downplayed by the authors at several points. For example, the authors state "Taking all these factors into account, the results suggest that while EDTs increased slightly in some animals, the cause of the increase could not be disambiguated between [Ca++] state changes, animal behavior, systematic neural network restructuring due to longitudinal performance…". The title of this section reads "Lever pull EDTs are similar or slightly higher with increasing lever-pull lockout duration". The Discussion again states that "Self-caused movement confounds have only minor effects on the decoding of future rewarded action."

We agree and have adjusted our results and panels accordingly.

First, the increase in EDTs is substantial (for example, in Figure 6h, the mean EDT across mice seems to increase from about -4.5 to about -2.5 sec, almost a 50% change) and calls into question the primary results. And second, while we can appreciate that many factors can worsen EDT detection as the lockout period increases (e.g., fewer sessions that occurred further apart), it is not sufficient to explain away the clear and concerning trends by listing several limitations. Especially when it seems that there may be ways to test this. For example, the authors could downsample lever pull events in the 3 sec and 6 sec lockout groups, etc., in order to match the sample size across the 5 conditions, and then repeat their analysis. Additionally, if their concern is that sessions are further apart in the longer lockout conditions, they could resample trials in the smaller lockout conditions in such a way to match the "timing" of samples across the 5 conditions, and then repeat their analysis.

We agree with this suggestion and have carried out this analysis. Subsampling shows a significant shortening of EDTs.

3. There is another point to check here, related to these matters. If we understand correctly, the authors used a post-hoc lockout of 3 seconds in their primary analyses (e.g., Figures 2b and 2d). Other lockout periods are tested in the controls in Figure 6. We are curious what constitutes a lever pull in these post-hoc lockout periods? In other words, there is a threshold beyond which the lever needs to be pulled to be rewarded (e.g., the 15{degree sign} angle illustrated in Figure 2a). When the authors exclude lever pulls that were preceded by another lever pull in the lockout window, was it required that the preceding lever pull also exceed the 15{degree sign} threshold? Or was another threshold used? On pg. 27 it says that rewarded and non-rewarded lever pulls were counted as lever pulls in the lockout period. What is meant by unrewarded here: unrewarded because the lever pulls did not meet the minimum threshold, the minimum hold duration for the pull, not hitting the maximum value (as described in Methods)?

We apologize for the confusion. All unrewarded pulls were locked out regardless of the failure mode.

The concern we are getting at is that if the minimum 15{degree sign} threshold was used to determine lever pulls in the lockout window, the authors may be not counting smaller (and/or less purposeful) unrewarded lever pulls in the lockout window. Should these pulls also modulate cortical activity, this would pose a problem with regards to Points 1 and 2 above. That is, it seems to us that mice should be as quiescent as possible (excepting perhaps licking / grooming) during the lockout window, to be certain that EDT estimates are not inflated by smaller lever pulls (actions) that may be missed in the lockout period.

We agree and the original analysis was indeed as suggested by the authors, namely all lever pulls were excluded for the lockout periods.

4. A response to these concerns would greatly benefit from deeper behavioral analyses. The authors should consider whether pre-pull behaviors are associated with EDTs. We have several suggestions on this point:Might the authors be able to use a classifier on their videos, to see whether pre-pull behaviors (e.g. subthreshold pulls, paw movements unassociated with the task, licking, grooming, random body motion, rewarded pulls, etc.) can predict future behavior? In other words, the authors could analyze whether classifier output is related to EDT. For example, when an EDT is -5 sec in a given session, are there animal behaviors occurring at -5 sec which might be predictive of the upcoming pull?

We have carried out analysis seeking to quantify the occurrence of different spontaneous behaviors and have found few spontaneous body movements occurring prior to lever pulls and no stereotyped structure (e.g. sequences) in movements.

The authors have a heterogeneous mouse population. Mouse M1 seems to have a smaller time horizon in their EDTs (as noted in Point 2 above) in Figure 6g. The authors should compare this mouse's behavior to other mice. For example, in the pre-pull period, does Mouse 1 engage in more/less behaviors that might explain why their EDTs differ from other mice.

We agree and have made some comments in the summary above. Mouse M1 behaved somewhat differently then other mice including decreased # of lever pulls over time and not showing peak in inter-lever-pull intervals at 3sec as most of the other mice did.

Are past behaviors predictive of future lever pulls? That is, do mice perform movement sequences of lever pulls (both rewarded/unrewarded)? Do the periodicity of movement sequences relate to EDTs? When sequences are more common in a session, does this appear related to EDT? Do actions such as licking, grooming, body motion, etc. predict upcoming lever presses? If so, does the relative timing between actions relate to EDTs on a given session?

We have carried out analysis (see above and main manuscript) and have not found evidence of movement sequences in our data.

Providing video data in example sessions across the EDT spectrum (e.g. some sessions where EDT was -10 sec, and others where it was -5 sec), could be helpful to the reader in interpreting changes in animal behavior/state that might occur near the EDT.

We believe the additional provided lever-pull autocorrelation and lever-pull vs. body movement cross correlation analysis to support our findings. Additionally, EDTs beyond 10 sec were largely removed. (we note that each EDT time was obtained from dozens to hundreds of trials and it is not practical to visualize such information in video).

Reviewer #3 (Recommendations for the authors):• The authors need to make a stronger case regarding the novelty of their study as compared to Murakami et al., 2014, 2017 and da Silva et al., 2018.

We had reviewed these studies prior to submitting our original manuscript and concluded they do not directly relate to the main findings of our study: that slow changes in neural activity lasting multiple seconds underlie self-initiated behavior; that future behaviors could be decode between 3 sec to 7 sec prior to movement; that the self-initiated behavior neural code is distributed across multiples regions not just motor cortex; that phases of neural oscillations across several cortical regions are highly stereotyped at the time of behavior; and that the variance of neural dynamics begins to change several seconds prior to behavior initiation. We provide additional comments on these studies as follows:

i. da Silva et al. 2018 is an in-depth study of dopamine neurons (which we have not targeted in our study) and showed that dopamine cell activity increases (naturally or optogenetically induced) in dopamine subcortical centers in the approximately 1sec prior to a behavior to facilitate movement. There is no specific discussion on decoding of future behavior nor timelines for decoding of such, though dopamine centers are certainly candidates in the search for the neural correlates of volitional action.

ii. Muarakami et al. 2014 is a study of secondary motor cortex neurons in a self-started task paradigm with the first component of the study having some similarities to the paradigm we implemented. The study finds that M2 neurons ramp to a “constant threshold at rates proportional to waiting time, strongly resembling integrator output”. The study focuses on M2 single neuron activity (whereas we targeted multiple cortical areas at the mesoscale activity level) and found that M2 neurons indeed ramped slowly prior to “poke in” behaviors on the scale of ~1 sec (or less) and “poke out” behaviors on the scale of 1 to 2 sec (we note that statistically significant differences of firing rates occurred generally only +/- 0.5 sec related to a poke in; e.g. Figure 5c of that paper). Additionally, mice carried out “poke-ins” quickly following the end of previous trials with little differences in firing rates in the pre-poke-in stage vs. duration of poke (e.g. Figure 3a of that paper). There is no analysis on behavior preparation. Additionally, there is no decoding analysis of the self-initiated components of the study. The study’s integrator model (Figure 8 of that paper) is focused largely on the hold condition of the study (not the self-initiated pre-poke stage), though such a model is also consistent with our data (and we see no novelty or controversy in that model being applicable to our data). We note that the task structure is different from our study. First, there are up to two stimuli as part of the task (the first tone is required for small water reward, the second for larger water reward). Second, abandoning waiting early (after the first water reward) still resulted in a smaller water reward – whereas performing a lever pull early in our paradigm yielded no water reward. Lastly, nose poking during a session could be viewed as actively sampling the nose port with their whiskers – which is a type of continuous stimulus sampling that was less present in our task (we agree that removing all sensory stimuli was not possible in our task and paradigm).

iii. Lastly, Murakami et al. 2017 is another interesting study on the neural dynamics preceding a nose-poke initiated task. It shows that “waiting time” biases are encoded by the secondary motor cortex at the single-trial level, while the medial prefrontal cortex only represents biases in action timing. These biases occurred on the ~1 sec time scale. There is no discussion or analysis of behavior preparation or decoding of the earliest neural correlates of future action preparation.

• By definition, voluntary action should be reason-responsive. Do the authors have any data to show lever-pull actions are goal-directed?

We thank the reviewer for their suggestion and have addressed the issue of learning at length above (see for example Essential Revision Point 2).

• Panels a,b,f in Figure 1 do not seem to contain any new information and could simply be mentioned in the text.

We thank the reviewer for this comment. We consider that some readers would benefit from a brief sketch of human studies paradigms.

• Please report the exact p-values.

We have added p-values in several missing sections.

• I am not an SVM expert but is it surprising to see decoding accuracy increasing with number of trials? What is the significance of this finding? This also refers to decoding data from multiple cortical regions rather than a specific region. Isn't it expected to see better decoding accuracy with more data?

Re: the correlation between decoding accuracy increases and number of trials is discussed at more length in the Essential Revisions Points 6,8 and 9. Briefly, while increasing the number of trials for a decoding paradigm usually improved decoding due to a decrease in the overall variance, in our case the increase in trials reflected stereotyped lever pulls which appear to affect EDT decoding. We have now made several changes including using only lockout data (i.e. only trials that were not preceded by lever pulls on the order of many seconds, e.g. 12sec or 15 sec, see Figure 2g,h).

• There are multiple interesting findings throughout the paper, but the authors failed to explain their significance. For example: (A) they found a higher bias of phases: i.e., behaviours were more likely to occur during a specific phase in mice than in humans. (B) There was a high diversity of phase preference between animals. (C) EDT improved longitudinally. (D) Limb and motor cortex oscillations had most power during pre-movement neural activity [but each animal showed a different pattern]. These are interesting findings but what are their significance and how do they contribute to addressing the research question?

We thank the reviewer for this issue, and we have added more interpretation in the Discussion of our study.

• The statistics are reported in quite an unusual way. Sometimes it is just described qualitatively without any inference test. Sometimes it is reported separately for each animal and sometime across the group. It is quite hard to make any conclusion about the findings when animals show different and in some cases opposite behaviour. Maybe the authors could perform mixed-effect modelling with by-subject random slope and intercept to investigate whether any of the findings is consistent across the animals and significant at the population level.

We thank the reviewer for these citations. We acknowledge that some of our results are described qualitatively and some quantitatively. In response to this concern (and concerns from other reviewers) we report additional statistical results in more detail in our study. However, we also note part of our findings is that we find inter-animal differences across several parameters including behaviors (e.g. # of rewarded lever pulls per session, trends in # of rewarded lever pulls per session) and neural dynamics (phase of neural areas activated during lever pulls vary significantly between animals, see Figure 3). As such, pooling animals into single categories and carrying out group statistics was not always feasible.

While we agree that mixed-effect modeling could be useful for longitudinal studies such as ours where some of the data is missing, we leave such analysis for a future study.

• The authors conclude that "Overall, these results link human voluntary action studies with rodent self-initiated behavior studies and suggest mice as an adequate model for studying the neural correlates of self-initiated action". This conclusion is not justified without a control condition contrasting self-initiated with externally-triggered action.

We thank the reviewer for their comment. Our conclusion refers only to the study of self-initiated behavior studies in humans and in mice. We have not carried out analysis of stimulus-triggered actions in mice nor have references any human studies that focus on this.

[Editors' note: further revisions were suggested prior to acceptance, as described below.]

The manuscript has been improved but there are some remaining issues that need to be addressed, as outlined below. In particular, the reviewers would like the authors to respond to the following comments, providing edits to the manuscript as needed (please see the full reviews for further elaboration).1) It seems the most likely interpretation is that EDTs relate to complete, or at least partial, pauses in behaviors (see Reviewer #1's recommendations for details).

Below we provide a detailed analysis in response to this Reviewer's specific recommendations.

2) In Figure 3, where the authors discuss their finding that movement initiation tends to coincide with peaks in a slow neural oscillation, this interpretation seems potentially problematic, as this oscillation could more simply be a decrease in neural activity during the pause in motor behaviors, and then the subsequent increase during the active motor period (so basically, an ON-OFF-ON transition in neural activity that is correlated to distinct motor events rather than some latent oscillation). An analysis that compares the duration of "quiescent periods" (nicely highlighted in Figure 1-Supp. 5c,d) to EDT is necessary to understand how to interpret these results (see Reviewer #1's recommendations for details).

We respond in detail below and have added Figure 1, Supplementary Figure 6 and Figure 1, Supplementary Figure 7.

3) The revised results are not as strong as in the original manuscript. More rigorous analysis of the EDTs revealed that the presence of sequential lever pulls had artificially lengthened the EDT, as anticipated by the reviewers. Additionally, the authors now acknowledge that some of the results are not as reliable as previously reported (i.e., large variability across animals in neural activity phases and longitudinal changes in neural dynamics), weakening the main conclusions of the paper.

Our response is as follows:

Re: sequential lever pulls artificially lengthening EDTs, this was an issue addressed in our initial submission where we carried out analysis excluding repeating lever pulls (e.g. Figure 6). On Revision #1, we added analysis showing that filtering data in Figure 6 resulted in EDTs ranging from -7sec to -3sec prior to lever pulls sufficient to support our conclusions re: EDT decoding.

Re: the longitudinal results being “unreliable”, we disagree (and this is not a term we used). We pointed out the original manuscript and Revision #1 that part of our results were the presence of diversity in mice in learning, performance and the underlying [ca] dynamics and decoded EDTs. We chose to leave all mice in the study rather than designing exclusionary thresholds (based on learning rates, decreased engagement etc.) to document this inherent variance.

Re: weakening the main conclusions of the paper, we believe the analysis added in Revision #1 and Revision #2 have strengthened and clarified our conclusions. In particular, we add Figure 8 which contains an analysis of [ca] dynamics preceding limb movements that are completely locked out from lever pulls and not preceded by any other movements by at least 5 seconds. This figure shows that for non-water rewarded random left paw movements, the average motor cortex [ca] signal over all mice begins to increase between -2sec to -1sec prior to body movement which is the dynamics and time scale observed in the EEG literature on the RP in humans. In contrast, for water rewarded lever pulls, motor cortex [ca] activity has a negative phase prior to movement which is part of the novel findings of our study.

In sum, these results show that when removing all confounding preceding movements, self-initiated left forelimb movements in most mice contain an excitatory motor and limb cortex signal that has a time course consistent with that observed in human EEG studies of the readiness potential. In contrast, self-initiated water rewarded lever pulls contain an inhibitory signal that starts earlier and has an inhibitory dynamic. Additionally, the dynamics underlying lever pulls are significantly less noisy than spontaneous actions and enable the decoding of upcoming behaviors using SVM methods (as shown in Figures 1,2 and 6), whereas the dynamics representing limb movements were too noisy to support SVM classification (see full explanation below; see also Methods).

In light of these results, we made the following changes:

Modified Fig1e to show [ca] averages for both left paw movements and rewarded lever pulls (and removed the panels on decoding of paw movements):

Results

Figure 1e

… (e) Average over all mice of pre-movement motor [ca] dynamics of left paw movements (black traces) and water-rewarded lever pulls (blue traces) (shading is standard error; see also Methods).

The Figure 8 is also added to our Main Results as follows:

Results (Figure 8)

The widefield [ca] activity correlates of self-initiated limb movement vs goal-oriented actions.

Given the results in Figures 1 – 7, namely, that prior to rewarded lever pulls there are systematic changes in cortex, e.g. inhibition of motor activity, on the order of several seconds prior to movement we sought to relate our results to human EEG studies of volitional action. In human volitional studies the motor cortex EEG signal preceding spontaneous body movements (e.g. flicking a wrist, pressing a button) generally shows an increase in activity commencing between -2sec to -1sec prior to the body movement. To compare our results to human studies we evaluated the [ca] activity preceding isolated random left forelimb movements, i.e. movements that were not related to lever pulls nor preceded by other movements for several seconds.

Identifying left paw movements isolated from other movements. We first computed the locations of all lever pulls relative to every left paw movement in windows of 20sec with 15sec of pre-paw movement and 5sec of post-paw movement (Figure 8a). Across all video recorded sessions we identified between 13,757 and 124,397 left paw movement bouts in individual mice (i.e. times where the left paw moved; see Methods; number of movements in all mice: M1: 54016; M2: 16241; M3: 18186; M4: 20487; M5: 13757; M6: 124397). We then ranked every left paw movement bout by the longest period of non-lever pull activity, i.e. we ranked paw movement bouts by the amount of lever pulls occurring in the previous 15sec or following 5sec (Figure 8a). We next extracted the trials with a complete lever-pull lockout (i.e. no lever pulls 15sec before or 5sec after) and added the locations of left and right paw movements and licking events (Figure 8b). We then realigned the surviving bouts by the longest period of body movement quiescence, i.e. we reranked the remaining paw movement again by quiescence (Figure 8c). Finally, we looked for left paw movements that were not preceded by any movements for at least 5sec. (Figure 8d). This approach revealed between 96 to 557 left paw movements that had at least 5sec of no preceding body movements and were completely isolated from lever pulls (number of movements per mouse: M1: 416, M2: 96, M3: 133, M4: 167, M5: 177, M6: 557).

Neural activity increases one to two seconds prior to left-limb movement. We next computed the average neural dynamic for the right hemisphere motor, upper-limb and retrosplenial cortex activity and found that both motor and upper-limb cortex exhibited a slow rising time course commencing at approximately -2sec prior to paw movement, with the upper-limb cortex signal being larger than motor cortex (Figure 8e for example from mouse M1). We computed the upper-limb cortex average signal for all animals and found that in 4 mice (M1, M3, M4, M5) this signal showed a significant increase prior to movement (i.e. rising above 3 x the standard deviation of the preceding neural activity) commencing approximately -2sec to -1sec relative to paw movement time (Figure 8f). Mouse M2 had an average [ca] signal which was noisy, likely due to the lower number of trials. Mouse M6 also had a noisy signal that showed an increase above 3 x std prior to movement, but not as pronounced as the other 4 mice. We note that mouse M6 had more than 5 months of recording and the pooled trials likely reflected significantly more variance due to pooling bouts from longitudinal data (see Figure 2, figure supplement 3 and Methods for a detailed discussion on the challenges of pooling widefield [ca] data across long periods of time). We also sought to decode upcoming behavior-locked out paw movements but found that SVM-based methods did no better than chance (not shown). This was likely due to higher dynamics occurring during non-goal oriented paw movements, but we also could not rule out the effect of pooling only a few hundred trials from tens of thousands of trials across many months of recording (see also Discussion).

Computing the average neural signals in motor, upper-limb and retrosplenial cortex from the means in all mice we found that both motor and upper-limb cortex showed a significant increase in the average neural signal beginning at approximately 2sec prior to paw movement (Figure 8g). In contrast, the average neural signal in these areas prior to water rewarded (locked-out) lever pulls showed a decrease commencing as early as -5sec prior to lever pull (Figure 8h).

In sum, we show that prior to non-lever pull related and isolated left paw movements, the [ca] neural activity in mouse motor cortex begins to increase at approximately -2sec to -1sec prior to paw movement – consistent with human EEG studies on the readiness potential dynamics occurring prior to spontaneous finger or hand movement. In contrast, self-initiated water rewarded lever pulls contain an inhibitory signal that starts earlier, at approximately -5 to -4 seconds prior to lever pull and contains stereotyped neural patterns that can be decoded to predict upcoming level pull timing (see Figures 1,2 and 6).

We removed Figure 2k, l, m panels which described decoding of left, right and licking behaviors but without controlling for confounding lever pulls. We now clarified those results:

Figure 2 Results:

… Importantly, with a few exceptions, licking or paw movements EDT distributions were not statistically different from lever pull time ETDs, however, due to the high correlation between paw movements and licking to lever pull times – we do not view them as completely independent analyses. Importantly, as we show below (see Figure 8) when considering only body movements that are isolated from lever pulls (i.e. not preceded by a lever pull in the previous 15sec, or following 5sec) and also not preceded by other body movements for at least 5sec, we find that [ca] averages show an increase in motor cortex (and other areas) and that EDTs are near 0sec as the signal is too noisy to enable decoding (see also Methods and Discussion).

4) The reviewers would like the authors to comment on the claim that they have identified a specific neural signature of self-initiated voluntary action. The authors show that pre-movement neural activity in mice contains structures that are not present in random neural activity. This observation is well supported by the data. However, to claim that this structure – neural dynamics becoming increasingly stereotyped prior to movement – is specific to self-initiated actions one needs to show that pre-self-initiated movement (uncued) neural activity in mice contains structures that are not present in externally-triggered movement (cued) neural activity. This comparison could also rule out other possible explanations such as motor confounds associated with the lever pulls or other related micro-movements. Comparison with random time during the task is not sufficient to make any conclusion regarding the self-initiated nature of the behavior.

The Reviewers suggest that our findings that “that pre-movement neural activity in mice contains structures that are not present in random neural activity” is not sufficiently established without evaluation whether similar dynamics are “present in externally-triggered movement (cued) neural activity.”

We respond as follows:

Our paradigm used self-initiated behavior invitations as trigger points for all analysis. And our findings are that there are stereotyped changes in [ca] dynamics prior to such behavior initiations sufficient to predict the timing of such initiations. However, we acknowledge this concern and have added the following to our discussion.

Discussion

Comparison of our study with cued action studies. In our experience, mice following external cues such as a light or sound triggers initiate their behaviors immediately and decoding the timing of future behavior on the scale of seconds is not possible. However, it is possible that in cue-triggered experiments – expert mice exhibit motor cortex inhibitory signals while they are waiting for a cue (e.g. see Schurger et al. 2012). However, such neural states would be conditioned on the eventual arrival of an external cue towards a future action and would not constitute self-initiated volitional action. Another possible paradigm would be to train mice to withhold action following a cue for several seconds prior to movement. In such a paradigm the neural correlates of action inhibition would be the focus rather than preparation of future action. Lastly, optogenetic approaches to activate or inhibit motor and forelimb cortex of mice after they have acquired the task could also be implemented, especially if mice learn a self-initiated rather than cued task (the latter being more often the case). Overall, we view these and other similar paradigms as useful complementary studies to our work and suggest them for future studies.

Reviewer #1 (Recommendations for the authors):I commend the authors on the care they have taken to address reviewer concerns. The added attention to 15-second lockout data is very useful. I think the focus on the 15-second lockout periods largely addresses the issue that sequences (or even random periods of movement) bias EDT estimates. Figure b (rebuttal Page 3) aligns with the updated EDTs in Figure g on Page 4. That is, neural activity changes around 3-4 seconds prior to the lever pull event in Figure b (Page 3), and EDTs seem largely capped at 3-4 seconds in Figure g (Page 4). In addition, there is evidence that mice are sensitive to the 3-second experimental lockout, given the drop in movement probability as shown in Figure c (Page 6). Thus, there does appear to be a 'cognitive decision' associated with obtaining the reward.But the clear analyses presented by the authors, highlight a new issue that has potential implications for the way they interpret the data. An interpretation that seems consistent with the analyses detailed on Pages 2-6, is that changes in neural activity related to EDTs are less so about pulling the lever, and more so about suppressing actions. For example, in Figure 2-Supplement 2, we see a gradual decline in activity in many brain regions. It appears that the onset of this decline aligns with EDT estimates via SVM. This decline appears to align with the movement data shown in Figure 1-Supp. 5c and 5d, which shows that overall movements start to decline about 3-5 seconds prior to the lever press event. And again, EDTs in the 15-second lockout datasets, appear to saturate around 3-5 seconds. And neural variability decreases in that window (Figure 5). Thus, it seems to be that this decline in spontaneous behaviors is related to the decline in neural activity, and the EDT estimates obtained by SVM. On the other hand, neural activity exhibits a sharp spike 500 ms prior to lever press (Figure 2-Supplement 2) which peaks at the same time as a sudden burst of all sorts of movements (not just lever presses) as seen in Figure 1-Supplement 5.It seems the most likely interpretation is that EDTs relate to complete, or at least partial, pauses in behaviors. The mouse has learned to suppress movement, and these movement suppressions are needed to acquire the reward. These movement suppressions begin at about 3-5 seconds because this is indeed the experimentally imposed lockout period. In other words, mouse behavior seems to represent a sequence of actions: pause for 3 seconds, then move.At the end of the day, this does still relate to volition. Without a cue, the mouse chooses to start a sequence, pause then move. But the idea that mice have learned a sequence, pause then move, does not seem to align with many critical points the authors make: e.g., "structured multi-second neural dynamics preceding self-initiated action" as claimed for example in the abstract, or the paper's title. Rather, if the mice are performing a sequence, pause then move, then the initial changes in neural activity (which are detected in EDT) are not about lever press, but about the pause that starts the sequence. In other words, it seems if the mice have a state change from movement to quiescence (and then lever press), then the EDTs should be detected not relative to the lever press but the movement quiescence that begins the "sequence".

The Reviewer suggests that “the most likely interpretation is that EDTs relate to complete, or at least partial, pauses in behaviors”. The Reviewer suggests that the results in Figure 3 showing differential firing on specific phases of widefield [ca] activity might be interpreted as neural activity phases, such as OFF-ON, occurring potentially due to a “pause in motor behaviors” during periods prior to lever pulls rather than reflecting the preparation of a future behavior. The Reviewer suggests comparing “quiescent period” duration to EDT might help to interpret the results.

In response to this comment we respond as follows:

We responded in part above in the summary, and Figure 8 also provides context to show that isolated left-paw movements do not exhibit motor cortex inhibition. That is, lever pulls towards a water reward might be a separate class of behavior with its own neural correlates. And we agree that inhibition of cortical activity may play a role in the preparation of such actions. But, as explained in the Discussion above, cortex inhibition as a component of preparation of future action may have confounds which lead to decreased activity. We view the issue as interesting and largely unexplored in humans and suggest that it be further addressed in future studies.

With respect to evaluating the role of quiescent periods to neural activity and EDTs, the Reviewer has suggested (i) splitting the 15sec lockout trials (or perhaps all trials) by the amount of movement occurring and (ii) then determining the relationship of pre-movement quiescence (e.g. duration) with EDT (or putatively [ca] dynamics). In response, we generated Figure 1, figure supplement 6. In sum, the figure shows that the vast majority of water rewarded lever pulls (i.e. 59% to 90% across all mice) contain several movements amounting to at least 1 second of motion in the period of -5sec to 0sec.

In sum, Figure 1—figure supplement 6 shows that:

1. As previously shown in Figure 1, figure supplement 5, mice tended to decrease their body movements in the period of -5sec to 0sec prior to a lever pull. However, the number of body movements prior to a lever pull lies on a continuum and there are few completely quiescent pre-lever pull periods.

2. In the vast majority of trials (59% to 90% across all mice) mice move for at least 1 sec (4 x 0.250sec time bins) in the period of -5sec to 0sec prior to lever pulls.

Taken together, these findings suggest that most of the time mice do not pause their motor behaviors in the period of -5sec to 0sec prior to a lever pull. However, given the above concerns regarding interpretation, we added additional content to the Discussion as shown above.

Another related issue might be the interpretation in Figure 3, where the authors discuss their finding that movement initiation tends to coincide with peaks in a slow neural oscillation. This interpretation seems potentially problematic, as this oscillation could more simply be a decrease in neural activity during the pause in motor behaviors, and then the subsequent increase during the active motor period (so basically, an ON-OFF-ON transition in neural activity that is correlated to distinct motor events rather than some latent oscillation).Thus, while overall the mice do clearly show that they are able to volitionally control themselves for at least 3-5 seconds to obtain the reward (i.e., stop moving), this should not be taken to mean that the neural activity during this period is necessarily important to the upcoming lever press. Stated very succinctly, it is problematic to assign EDT-related neural activity to the lever press, when there is a large coincident change in the behavioral state of the animal that precedes the lever press (the complete or partial suppression of behavior). It seems more analysis on this quiescence is very important to the paper's interpretation of their results.Data that are needed on this point would be an analysis that compares the duration of "quiescent periods" (nicely highlighted in Figure 1-Supp. 5c,d) to EDT. For example, I anticipate that occasions where the animal pauses body movements for a longer period prior to the lever press will relate to longer EDTs. On this note, it would be very helpful for the authors to provide an analogue of Figure 1-Supplement 5 (Panels c and d) for the 15-second lockout data specifically, and for all their mice as opposed to solely mouse M1. Do 15-second lockout data also show the 3-5 second quiescent periods prior to lever pulls? Another suggestion on this point: the error bars in Figure 1-Supp. 5d are large: both in the "pause" prior to time point 0, and the peak in activity at time point 0. This implies some heterogeneity in the extent to which animals pause, and the action they take after a pause. The authors could take advantage of this. If the authors binned trials based on (a) complete pause, (b) partial pause, and (c) no pause, do these groups show different neural activity patterns in the -5 to 0-sec range in Figure 2-Supp. 2 (a similar binning analysis could be done based on the length of the pause.)? If so, again, these analyses would be a reason to suspect that much of the neural signal change detected in the EDT has to do with a pause in activity prior to lever press, as opposed to the lever press itself. Trial-to-trial analyses could also be conducted on this issue, relating the pause duration (or perhaps pause magnitude, e.g., the extent to which behaviors stop prior to lever press) relates to neural activity on that trial (to this end, GLMs may be useful to parse overall activity into component behaviors).

The Reviewer suggests, as in the previous segment, that splitting the data based on a range of behavior amounts from least movement to partial movements to most movements may reveal the systematic relationship between pauses and EDT or [ca] activity.

We respond as follows:

We followed the Reviewers suggestions and carried out an “analysis that compares the duration of "quiescent periods"” and binned “trials based on (a) complete pause, (b) partial pause, and (c) no pause”. We provide Figure 1, Supplementary Figure 7.

The above analysis shows that:

In some mice, the average neural activity traces from the least active trials (red traces) appear to commence slightly earlier than the most active trials (black trials), for example mouse M1, M3 and M4 (panel f). However, the averages are noisy and the standard-error-of-the-mean shading overlaps between these two groups in all cases. In contrast, traces from mouse M2, M5 and M6 seem to show that either mid-activity trials (blue) or high-activity trials (black) begin to decrease or peak lower than the least active trials (red). The average of the averages of all mice (panel (g)) shows that the most active trials are largely within 3 standard deviations of the man while the mid-active and least-active averages show similar curves.

Taken together, these results suggest that during the most active trials motor cortex receives the most inhibition whereas in other circumstances the dynamics are likely to be similar.

We have additionally added the Method for this figure to our Manuscript and referenced it in Figure 1.

Methods

"Data processing for Figure 1, figure supplement 7. Pooling data across weeks or months of longitudinal recordings adds substantial noise and other confounds to our datasets (see Figure 2, figure supplement 3, and Main manuscript for Figure 6 for a detailed explanation). For the analysis here we pooled lever lockout trials (as in Figure 6) but were further limited to only those sessions where video was available. This reduced our overall trials by approximately 50% (or more) in all mice compared to Figure 6 analysis. The trials were further split into 3 groups representing least to most body movements. Additionally, trials pooled into these subgroups were ordered by movement amount – as opposed to ordered by day of acquisition time. This also has the effect of increasing the variance of data means due to time between individual trials in the subgroup. Given these issues, our analysis in this panel was limited to characterizing [ca] time courses only (i.e. it was not possible to also carry out EDT decoding on trials contained in these reduced datasets – as the time series were too noisy).”

Reviewer #2 (Recommendations for the authors):This is the second review of the manuscript "Mesoscale cortex-wide neural dynamics predict self-initiated actions in mice several seconds prior to movement".Overall, the authors have addressed in detail the concerns raised by the previous round of reviews.On the one hand:1) The authors have appropriately toned down claims about "voluntary" movements.2) The authors have also performed new analyses to address the concerns about sequences of stereotyped movements but did not address the concerns of micro-movements due to a lack of proper video data. This limitation was acknowledged in the manuscript and I have no additional concerns in this respect.On the other hand:3) The revised results are not as strong as in the original manuscript. More rigorous analysis of the EDTs revealed that the presence of sequential lever pulls had artificially lengthened the EDT, as anticipated by the reviewers. Additionally, the authors now acknowledge that some of the results are not as reliable as previously reported (i.e., large variability across animals in neural activity phases and longitudinal changes in neural dynamics), weakening the main conclusions of the paper.

We thank the reviewer for the comment, we have addressed it above in the Main Concerns discussed above.

Reviewer #3 (Recommendations for the authors):The authors have done a great job addressing the comments. The new analyses and figures have significantly improved the readability and quality of the manuscript.I am, however, still not completely convinced by the authors' claim that they have identified a specific neural signature of self-initiated voluntary action. The authors show that pre-movement neural activity in mice contains structures that are not present in random neural activity. This observation is well supported by the data. However, to claim that this structure – neural dynamics becoming increasingly stereotyped prior to movement – is specific to self-initiated actions one needs to show that pre-self-initiated movement (uncued) neural activity in mice contains structures that are not present in externally-triggered movement (cued) neural activity. This comparison could also rule out other possible explanations such as motor confounds associated with the lever pulls or other related micro-movements. Comparison with random time during the task is not sufficient to make any conclusion regarding the self-initiated nature of the behavior.

We have responded to this comment above and added a section to our Discussion directly to address it.